


# An inter-comparison of tropospheric ozone reanalysis products from CAMS, CAMS-Interim, TCR-1 and TCR-2

Vincent Huijnen[1], Kazuyuki Miyazaki[2], Johannes Flemming[3], Antje Inness[3], Takashi Sekiya[4], Martin G. Schultz[5]

[1] Royal Netherlands Meteorological Institute, De Bilt, the Netherlands
[2] Jet Propulsion Laboratory, California Institute of Technology, Pasadena, CA 91109, USA
[3] ECMWF, Shinfield Park, Reading, RG2 9AX, UK
[4] Research Institute for Global Change (RIGC), Japan Agency for Marine–Earth Science and Technology (JAMSTEC), Yokohama 2360001, Japan
[5] Jülich Supercomputing Centre, Forschungszentrum Jülich, Jülich, Germany

*Correspondence to*: V. Huijnen (Vincent.Huijnen@knmi.nl)

**Abstract.** Global tropospheric ozone reanalyses constructed using different state-of-the-art satellite data assimilation systems, prepared as part of the Copernicus Atmosphere Monitoring Service (CAMS-iRean and CAMS-Rean) as well as two fully independent Tropospheric Chemistry Reanalyses (TCR-1 and TCR-2), have been inter-compared and evaluated for the past decade. The updated reanalyses (CAMS-Rean and TCR-2) generally show substantially improved agreements with independent ground and ozonesonde observations over their predecessor versions (CAMS-iRean and TCR-1) for the diurnal, synoptical, seasonal, and decadal variability. The improved performance can be attributed to a mixture of various upgrades, such as revisions in the chemical data assimilation, including the assimilated measurements, and the forecast model performance. The updated chemical reanalyses agree well with each other for most cases, which highlights the usefulness of the current chemical reanalyses in a variety of studies. Meanwhile, significant temporal changes in the reanalysis quality in all the systems can be attributed to discontinuities in the observing systems. To improve the temporal consistency, a careful assessment of changes in the assimilation configuration, such as a detailed assessment of biases between various retrieval products, is needed. Even though the assimilation of multi-species data influences the representation of the trace gases in all the systems and also the precursors' emissions in the TCR reanalyses, the influence of persistent model errors remains a concern, especially for the lower troposphere. Our comparison suggests that improving the observational constraints, including the continued development of satellite observing systems, together with the optimization of model parameterisations, such as deposition and chemical reactions, will lead to increasingly consistent long-term reanalyses in the future.

## 1 Introduction

The global distribution of present-day tropospheric ozone, together with its interannual variability and trends, plays an important role when describing the impact of human activity and natural processes on air quality and climate. Amongst





other factors, increments in surface ozone concentrations contribute to changes in air quality (e.g., Im et al., 2018), human health (Liang et al., 2018), and agriculture (van Dingenen et al., 2009). Owing to its radiative effects, tropospheric ozone is

an important driver in climate-change (Checa-Garcia et al., 2018), even if no improvement in long-range weather forecasts has been detected so far (Cheung et al., 2014). Considering its lifetime of a few weeks, tropospheric ozone can be controlled by both local and remote pollution sources through atmospheric chemical processes and long-range transport (Jonson et al., 2018), as well as stratospheric influx (e.g. Knowland et al., 2017). In addition to anthropogenic sources, natural processes such as El Niño–Southern Oscillation (ENSO) conditions affect tropospheric ozone production and loss terms, through

changes in upwelling, convection, solar irradiance, humidity, and biomass burning emissions (e.g., Ziemke and Chandra, 2003, Inness et al., 2015). Other processes that potentially influence tropospheric ozone, but are generally considered of minor importance, are the quasi-biennial oscillation (Neu et al., 2014) and the North Atlantic Oscillation (Thouret et al., 2006).

Various types of datasets have been compiled to allow the analysis of the current state of tropospheric ozone and its changes

over time. Tropospheric ozone is reasonably well monitored through in-situ networks measuring surface concentrations (Global Atmosphere Watch (GAW), Global Monitoring Division (GMD), European Monitoring and Evaluation Programme (EMEP), AirNow), as collected and homogenised by the Tropospheric Ozone Assessment Report (TOAR, Schultz et al., 2017). Above the ground ozone is monitored through ozone sondes, collected by World Ozone and Ultraviolet Radiation Data Centre (WOUDC; https://woudc.org/), and aircraft (In-Service Aircraft for a Global Observing System (IAGOS),

Nédélec et al., 2015). These observations are complemented with (combined) satellite observations of the instruments Global Ozone Monitoring Experiment–2 (GOME-2), Ozone Monitoring Instrument (OMI), Microwave Limb Sounder (MLS), Infrared Atmospheric Sounding Interferometer (IASI), Tropospheric Emission Spectrometer (TES). Here each retrieval product comes with its specific (vertical) sensitivity, which allows the derivation of tropospheric ozone columns as listed in Gaudel et al. (2018).

The multitude of observational datasets have led to observationally constrained assessments of the current state and trends in tropospheric ozone, for instance documented as part of TOAR (Schultz et al., 2017; Gaudel et al., 2018; Fleming et al., 2018; Tarasick et al., 2019). Recent studies have also shown decadal-scale changes in global tropospheric ozone using various observations, such as a shift in the seasonal cycle at northern hemisphere (NH) mid-latitudes and long-term trends over many regions (e.g., Parrish et al., 2014; Cooper et al., 2014; Gaudel et al., 2018; Fleming et al., 2018). Based on a combination of

multiple ozone retrieval products, Ziemke et al. (2019) have inferred positive trends in tropospheric ozone trends, particularly in the 2005-2016 time period.

Simultaneously international modelling initiatives have been established, for instance to analyse the contribution of ozone on air quality (AQMEII), the impact of long-range transport on air quality (HTAP), and the impact of composition changes on climate change (CCMI). (e.g., Young et al., 2013; Morgenstern et al., 2018; Liang et al., 2018)

Following the concept of meteorological reanalyses such as ERA5 (Hersbach et al., 2018), observationally constrained reanalyses of the atmospheric chemical composition have been developed to provide time series of tropospheric and



stratospheric ozone. A reanalysis is a systematic approach to create long-term data assimilation products by combining a series of observational datasets with a model. Advanced data assimilation, such as four-dimensional variational data assimilation (4D-VAR) and ensemble Kalman filter (EnKF), allows the propagation of observational information in time and
space, and from a limited number of observed species, to an analysis of a wide range of chemical components. This can be used in reanalyses to provide consistent global fields that are in agreement with individual observations (Lahoz and Schneider, 2014; Bocquet et al., 2015). A reanalysis hence provides an instantaneous global image of atmospheric composition, together with its change over time and therefore serves in principle to analyse the mean state of the atmosphere, together with its variability and trends.

Applications of chemical reanalyses include comprehensive spatiotemporal evaluation of independent models, such as those developed in the framework of ACCMIP (Young et al., 2013) and CCMI (Morgenstern et al., 2018). This was shown useful as comparisons using individual measurements suffer from significant sampling biases (Miyazaki and Bowman, 2017). In their study the ACCMIP ensemble ozone simulations were evaluated using a chemical reanalysis, complementing the use of individual measurements for such purpose. The chemical reanalyses can also be used as an input to meteorological
reanalyses, e.g. for radiation calculations (Dragani et al., 2018), and they can provide boundary conditions to regional-scale models and to analyse particular constellations of pollution (e.g., those associated with heat waves or large-scale forest fires; Ordóñez et al., (2010), Huijnen et al., (2012, 2016)). Finally, they can be used as a reference to identify to what extent particular periods and regions deviate from climatology, as provided by the reanalysis, as for instance also discussed in the series of the State of the Climate (Flemming and Inness, 2018).

However, all of these applications presume that the reanalysis is sufficiently accurate, which, despite many years of research and the range of observations assimilated into the system, is not ensured. Issues are multiple, and depend on the availability of observations, and on the modelling and data-assimilation framework with respect to the species and location under consideration. For tropospheric ozone reanalyses, state-of-the-art global analysis systems have been used to assimilate satellite-based observations, where satellite measurements have limited information on vertical profiles. In particular, the
small measurement sensitivities to the lower troposphere makes it difficult to correct near-surface ozone. Advanced satellite retrievals provide improved vertical resolution to the troposphere (Cuesta et al., 2018; Fu et al., 2018), but the temporal coverage and vertical resolution of these retrievals is still limited, and their application in data assimilation remains a challenge (Miyazaki et al., 2019a). This also implies that constraints on other parts of the system (other trace gases, aerosol, their emissions, as well as meteorology, driving the tracer transport and its removal) will strongly affect the quality of the
reanalysis. Simultaneous optimization of concentrations and precursors' emissions seems thus important in improving the analysis of lower tropospheric ozone (Miyazaki et al., 2012b). Furthermore, providing consistent time series over a decadal time-scales is challenging. The observational data from satellite instruments available for assimilation evolve over time with new instruments becoming available while others cease to exist, and different satellite retrieval products typically showing biases with respect to ground-based observations as well as with respect to each other.



In the framework of the Copernicus Atmosphere Monitoring Service (CAMS, https://atmosphere.copernicus.eu), ECMWF's Integrated Forecasting System (IFS) has been extended to include modules for atmospheric chemistry, aerosols and greenhouse gases. Using this system, three recent reanalyses have been released: the Monitoring Atmospheric Composition and Climate (MACC) reanalysis for the years 2003-2012 (Inness et al., 2015), the 'CAMS Interim Reanalysis' for the years 2003-2018 (Flemming et al., 2017) and recently the 'CAMS Reanalysis' for the years 2003-present (Inness et al., 2019).

Miyazaki et al. (2015) simultaneously estimated concentrations and emissions for an 8-year tropospheric chemistry reanalysis (TCR-1) for the years 2005–2012 obtained from an assimilation of multi-constituent satellite measurements using an ensemble Kalman filter (EnKF). TCR-1 has been used to provide comprehensive information on atmospheric composition variability and elucidate variations in precursor emissions, and to evaluate bottom-up emission inventories (Miyazaki et al., 2012, 2014, 2015, 2017; Ding et al., 2017; Jiang et al., 2019; Tang et al., 2019). A second version of the EnKF-based

reanalysis (TCR-2) has been recently produced using an updated model and satellite retrievals for the years 2005–2018 (Kanaya et al., 2019; Miyazaki et al., 2019a; Thompson et al., 2019). For stratospheric constituents, several studies have been conducted to produce and compare stratospheric chemical reanalysis products (Davis et al., 2017; Errera et al., 2019).

Here we evaluate the ability of the two CAMS and two TCR atmospheric composition reanalysis data sets to constrain tropospheric ozone variability. We do not evaluate the MACC reanalysis here, because it has been extensively documented

in the past (Inness et al. 2013; Flemming et al., 2017; Bennouna et al., 2019) and only covered the 2003-2012 time period. Furthermore it has been shown to suffer from significant spurious drifts in tropospheric ozone due to a bias-correction issue, which makes it less useful to assess its multi-annual mean and inter-annual variability. In particular, Katragkou et al. (2015) discusses the ozone in the MACC reanalysis; while Inness et al., (2019) reports how CAMS-REAN compares to CAMS-iREAN and MACC reanalysis.

To assess the quality of these reanalysis products, and in particular to evaluate their fitness for purpose for the various types of application described above, this study evaluates tropospheric ozone for a range of independent in-situ observations: ozone sondes from various networks (WOUDC, NOAA Earth System Research Laboratory (ESRL), and Southern Hemisphere Additional Ozonesondes (SHADOZ), monthly mean gridded surface ozone as collected within TOAR, and individual surface ozone observations from the EMEP network.

In this study, we limit ourselves to tropospheric ozone in the reanalysis products, and only refer, where relevant, to interactions with other components in the reanalysis systems, such as NOx and CO, and aerosols. This intercomparison aims to reveal to what extend the reanalysis products agree, depending on region and time periods. Temporal consistency is an important aspect when assessing long-term time series and intercomparing individual years. At the same time this is a challenge because of the changing constellation of satellite observations used to constrain the reanalysis products over the

course of a decade or more, all having different retrieval specifications (see also Gaudel et al., 2018).

In the next sections we describe the various reanalysis products used in this paper (Sect. 2) and the observational data used for evaluation (Sect. 3). Evaluations against ozone sondes are presented in Sect. 4, and against TOAR gridded surface ozone and EMEP surface observations in Sect. 5, and Sect. 6, respectively. We continue describing the reanalysis products through



assessment of their tropospheric column time series (Sect 7) and average concentrations (Sect 8), to assess the spread. We

end with discussions and conclusions in Sect 9.

## 2. Chemical reanalysis products

The global atmospheric chemistry reanalysis products evaluated in this paper are listed in Table 1. The general configuration of the various data assimilation systems, together with details specific to tropospheric ozone analysis, are provided in the following subsections. For more detailed information on the specifications of the various reanalysis products the reader is

referred to the references.

**Table 1. Overview of recent reanalysis products**

| Name (reference) | Time period | Altitude range and horizontal resolution | Forecast model | Data assimilation scheme | Assimilated components |
|---|---|---|---|---|---|
| CAMS-iREAN (Flemming et al., 2017) | 2003-2018 | Up to 0.1hPa T159/L60 | IFS(CB05) CY40R2 | 4D-VAR | CO, $O_3$, AOD |
| CAMS-REAN (Inness et al., 2019) | 2003-present | Up to 0.1hPa T255/L60 | IFS(CB05) CY42R1 | 4D-VAR | CO, $O_3$, $NO_2$, AOD |
| TCR-1 (Miyazaki et al., 2015; Miyazaki and Bowman, 2017) | 2005-2014 | Up to 4.4 hPa T42/L32 | MIROC-Chem Nudged to ERA-Interim | EnKF | CO, $O_3$, $NO_2$, $HNO_3$ |
| TCR-2 (Miyazaki et al., 2019a, Kanaya et al., 2019) | 2005-2018 | Up to 4.4 hPa T106/L32 | MIROC-Chem Nudged to ERA-Interim | EnKF | CO, $O_3$, $NO_2$, $HNO_3$, $SO_2$ |





## 2.1 The CAMS Interim reanalysis

In CAMS, the data assimilation capabilities in IFS for trace gases and aerosols relies on the four-dimensional variational (4D-VAR) technique, developed for the analysis of meteorological fields. The CAMS interim Reanalysis (CAMS-iREAN, Flemming et al., 2017) has been the intermediate reanalysis between the widely used MACC Reanalysis (Inness et al., 2015) and the recently produced CAMS reanalysis (Inness et al., 2019). The chemistry module as adopted in CAMS-iREAN is described and evaluated in Flemming et al. (2015). It relies on the modified CB05 tropospheric chemistry mechanism as originating from TM5 (Huijnen et al., 2010; Williams et al., 2013) which contains 52 species and 130 (gas-phase + photolytic) reactions; stratospheric ozone is modelled through the Cariolle parameterization (Cariolle and Teyssèdre, 2007). Anthropogenic emissions originate essentially from the MACCity inventory (Granier et al., 2011) with enhanced wintertime CO emissions over Europe and US (Stein et al., 2014). Monthly specific biogenic emissions originate from MEGAN-MACC (Sindelarova et al., 2014), but using monthly climatological values from 2011 onwards. Daily biomass burning emissions originate from GFASv1.2 (Kaiser et al., 2013). The meteorological model is adopts IFS CY40R2.

In terms of ozone, observations from the following set of satellite instruments assimilated: SBUV-2, OMI, MLS, GOME-2, SCIAMACHY, GOME and MIPAS, see also Table 2.

Limb observations are instrumental to discriminate between the tropospheric and stratospheric contribution of the total column observations. CAMS-iREAN uses observations from the MIPAS instrument for the period February 2005 - March 2012. MLS data on Aura have been used from August 2004 onwards, based on version 2 observations during August 2004-Dec 2012, and V3.4 from January 2013 onwards. V3.4 has a different specification of the vertical levels and observation errors compared to V2 (Schwartz et al., 2015). Finally, note that in CAMS-iREAN no observations of $NO_2$ have been assimilated. CO has been constrained through assimilation of Measurement of Pollution in the Troposphere (MOPITT) total columns.

**Table 2: Observations of ozone used in the CAMS-iREAN assimilation system**

| Instrument (satellite) | Product | Data provider/version | Period |
|---|---|---|---|
| SCIAMACHY (Envisat) | TC | ESA, CCI (BIRA) | 2003-01-01 to 2012-04-08 |
| MIPAS (Envisat) | Prof | ESA, NRT<br>ESA, CCI(KIT) | 2005-01-27 to 2012-03-31 |
| MLS (Aura) | Prof | NASA /V2<br>NASA/V3.4 | 2004-08-03 to 2012-12-31<br>2013-01-01 to 2016-12-31 |
| OMI(Aura) | TC | KNMI /V3<br>KNMI /NRT | 2004-08-03 to 2015-05-31<br>2015-06-01 to present |
| GOME (ERS-2) | Prof | RAL | 2003-01-01 to 2003-05-31 |



| GOME-2 (Metop-A) | TC | ESA(CCI),BIRA /fv0100 | 2007-01-23 to 2012-12-31 |
| | | ESA(CCI),BIRA /fv0300 | 2013-01-01 to 2016-12-31 |
| | | NRT | 2017-01-01 to present |
| GOME-2 (Metop-B) | TC | ESA(CCI),BIRA /fv0300 | 2013-01-01 to 2016-12-31 |
| | | NRT | 2017-01-01 to present |
| SBUV/2 (NOAA-14 – NOAA-19) | PC | NASA / v8.6 13L | 2003-01-01 to 2012-12-31 |
| | | NRT 21L | 2013-01-01 to present |

170

**Table 3: Observations of ozone used in the CAMS-REAN assimilation system**

| Instrument (satellite) | Product | Data provider/version | Period |
|---|---|---|---|
| SCIAMACHY (Envisat) | TC | ESA, CCI (BIRA) | 2003-01-01 to 2012-04-08 |
| MIPAS (Envisat) | Prof | ESA, NRT<br>ESA, CCI(KIT) | 2003-01-27 to 2004-03-26 and<br>2005-01-27 to 2012-03-31 |
| MLS (Aura) | Prof | NASA /V4 | 2004-08-03 to 2016-12-31 |
| OMI(Aura) | TC | KNMI /V3<br>KNMI /NRT | 2004-08-03 to 2015-05-31<br>2015-06-01 to present |
| GOME-2 (Metop-A) | TC | ESA(CCI),BIRA /fv0100<br>ESA(CCI),BIRA /fv0300<br>NRT | 2007-01-23 to 2012-12-31<br>2013-01-01 to 2016-12-31<br>2017-01-01 to present |
| GOME-2 (Metop-B) | TC | ESA(CCI),BIRA /fv0300<br>NRT | 2013-01-01 to 2016-12-31<br>2017-01-01 to present |
| SBUV/2 (NOAA-14 – NOAA-19) | PC | NASA / v8.6 13L<br>NRT 21L | 2003-01-01 to 2013-07-07<br>2013-07-08 to present |

175





## 2.2 The CAMS Reanalysis

The CAMS Reanalysis (CAMS-REAN; Inness et al., 2019) is the successor of the CAMS-iREAN. Compared to CAMS-iREAN, the horizontal resolution has increased to ~80 km (T255), while meteorology is now based on CY42R1. Emissions are largely similar to CAMS-iREAN, except that the monthly varying biogenic emissions have been used for the full time period. With respect to the CB05-based chemistry module, heterogeneous chemistry on clouds and aerosol has been switched on, as well as the modification of photolysis rates due to aerosol scattering and absorption (Huijnen et al., 2014).

As for assimilated ozone observations, data from a very similar set of instruments have been used as for CAMS-iREAN: SCIAMACHY, MIPAS, OMI, MLS, GOME-2, and SBUV/2, see Table 3. However, note that the CAMS-Interim Reanalysis additionally assimilated GOME observations during the first 5 months of 2003, which have not been assimilated in CAMS-REAN as it was found to lead to a degradation in the $O_3$ analysis. Different to CAMS-iREAN, CAMS-REAN also assimilated observations from the MIPAS instrument during 2003 and early 2004. Also frequently newer versions of the data have been adopted in CAMS-REAN compared to CAMS-iREAN, particularly for MLS observations the reprocessed version 4 has been applied throughout the full time period.

In CAMS-REAN also tropospheric $NO_2$ columns are assimilated, using observations from the SCIAMACHY (2003-2012), OMI (from October 2004 onwards) and GOME-2 (from April 2007 onwards) instruments. A variational bias correction (VarBC) scheme was applied to OMI, SCIAMACHY and GOME-2 retrievals of total ozone columns to ensure optimal consistency of all information used in the analysis. SBUV/2 and also profile retrievals from MLS and MIPAS were assimilated without correction. Inness et al. (2019) provide an extended overview of the biases of various assimilated observations against the reanalysis. For ozone assimilation in particular, the following findings are most noteworthy for this study (see also Appendix C of Inness et al., 2019):

• Larger biases for SCIAMACHY observations in 2003 and early 2004, associated to issues with the early SCIAMACHY $O_3$ retrievals in this time period

• Larger departures for MIPAS data during 2003-2004 than after 2005, where CCI data was used

• Different behaviour of OMI data between 2009 and 2012, associated to a deterioration in the OMI row anomalies (Schenkeveld et al., 2017) which could not be filtered out in the CAMS assimilation procedure.

• An increasing bias correction for GOME-2A especially after January 2013, associated to a version change of the SBUV/2 data.

CAMS-iREAN and CAMS-REAN surface and tropospheric ozone are archived with a three-hourly output frequency.



### 2.3 Tropospheric Chemistry Reanalysis (TCR-1)

The TCR-1 data assimilation system is constructed using an EnKF approach. A revised version of the TCR-1 data is used in this study. A major update from the original TCR-1 system (Miyazaki et al., 2015) to the system used here and in Miyazaki et al. (2017) is the replacement of the forecast model from CHASER (Sudo et al., 2002) to MIROC-Chem (Watanabe et al.,
2011), which caused substantial changes in the a priori field and thus the data assimilation results of various species. MIROC-Chem (Watanabe et al., 2011), considers detailed photochemistry in the troposphere and stratosphere by simulating tracer transport, wet and dry deposition, and emissions, and calculates the concentrations of 92 chemical species and 262 chemical reactions. The MIROC-Chem model used in TCR-1 has a T42 horizontal resolution (~2.8°) with 32 vertical levels from the surface to 4.4 hPa. It is coupled to the atmospheric general circulation model MIROC-AGCM version 4 (Watanabe
et al., 2011). The simulated meteorological fields were nudged toward the 6-hourly ERA-Interim (Dee et al., 2011) to reproduce past meteorological fields.

The a priori anthropogenic $NO_x$ and CO emissions were obtained from the Emission Database for Global Atmospheric Research (EDGAR) version 4.2 (EC-JRC, 2011). Emissions from biomass burning were based on the monthly Global Fire Emissions Database (GFED) version 3.1 (van der Werf et al., 2010). Emissions from soils were based on monthly mean
Global Emissions Inventory Activity (GEIA; Graedel et al., 1993).

The data assimilation used is based upon on an EnKF approach (Hunt et al., 2007) that uses an ensemble forecast to estimate the background error covariance matrix and generates an analysis ensemble mean and covariance that satisfy the Kalman filter equations for linear models. The concentrations and emission fields of various species are simultaneously optimized using the EnKF data assimilation, see also Table 4.

For data assimilation of tropospheric $NO_2$ column retrievals, the version 2 Dutch OMI $NO_2$ (DOMINO) data product (Boersma et al., 2011) and version 2.3 TM4NO2A data products for SCIAMACHY and GOME-2 (Boersma et al., 2004) were used, obtained through the TEMIS website (http://www.temis.nl). The TES ozone data and observation operators used are version 5 level 2 nadir data obtained from the global survey mode (Bowman et al., 2006; Herman and Kulawik, 2013). Note that the availability of TES measurements is strongly reduced after 2010, which led to a degradation of the reanalysis
performance, as demonstrated by Miyazaki et al. (2015). The MLS data used are the version 4.2 ozone and $HNO_3$ level 2 products (Livesey et al., 2011). Data for pressures of less than 215 hPa for ozone and 150 hPa for $HNO_3$ were used. The MOPITT CO data used are version 6 level 2 TIR products (Deeter et al., 2013). A superobservation approach was employed to produce representative data with a horizontal resolution of the forecast model $NO_2$ and CO observations, following the approach of Miyazaki et al. (2012). No bias correction was applied to the assimilated measurements.






## 2.4 Updated Tropospheric Chemistry Reanalysis (TCR-2)

An updated CTM and satellite retrievals are used in TCR-2 (Kanaya et al., 2019; Miyazaki et al., 2019a, 2019b; Thompson et al., 2019). A high-resolution version of the MIROC-Chem model with a horizontal resolution of T106 (1.1° x 1.1°) was used. Sekiya et al. (2018) demonstrated the improved model performance on tropospheric ozone and its precursors by

increasing the model resolution from 2.8° x 2.8° to 1.1° x 1.1°. A priori anthropogenic emissions of $NO_x$ and CO were obtained from the HTAP version 2 inventory for 2008 and 2010 (Janssens-Maenhout et al., 2015). Emissions from biomass burning are based on the monthly GFED version 4.2 inventory (Randerson et al., 2018) for NOx and CO, while those from soils are based on the monthly GEIA inventory (Graedel et al., 1993) for NOx. Emission data for other compounds are taken from the HTAP version 2 and GFED version 4 inventories.

The satellite products used in TCR-2 are more recent than those used in TCR-1, see Table 4. Tropospheric $NO_2$ column retrievals used are the QA4ECV version 1.1 L2 product for OMI (Boersma et al., 2017a) and GOME-2 (Boersma et al., 2017b). The MLS data used are the version 4.2 ozone and $HNO_3$ L2 products (Livesey et al., 2011). The MOPITT total column CO data used were the version 7L2 TIR/NIR product (Deeter et al., 2017). OMI $SO_2$ data of the planetary boundary layer vertical column L2 product were used as produced with the principal component analysis algorithm (Krotkov et al.,

2016; Li et al., 2013). As in TCR-1, a super-observation approach to produce representative data with a horizontal resolution of the forecast model (1.1° × 1.1°) for $NO_2$ and CO observations was applied.

TCR-2 data was used to study the processes controlling air quality in East Asia during the KORUS-AQ aircraft campaign (Miyazaki et al., 2019a). Kanaya et al. (2019) demonstrated the TCR-2 ozone and CO performance using research vessel observations over open oceans. Thompson et al. (2019) used the TCR-2 data to help understanding of near surface $NO_2$

pollutions observed during the KORUS-OC campaign. Both for TCR-1 and TCR-2 the reanalysis data is archived on a two-hourly output frequency.

**Table 4: Observations used for ozone assimilation in TCR-1, and in brackets changes for TCR-2.**

| Instrument (satellite) | Species | Product | Data provider/version | Period |
|---|---|---|---|---|
| OMI (Aura) | $NO_2$ | TrC for $NO_2$ | DOMINO v2 [QA4ECV v1.1] for $NO_2$ | 2005-01-01 to present |
| | [+ $SO_2$] | [PBL for $SO_2$] | [PCA v3 for $SO_2$] | |
| SCIAMACHY (Envisat) | $NO_2$ | TrC | DOMINO v2 [QA4ECV] | 2005-01-01 to 2012-03-29 |
| GOME-2 (Metop-A) | $NO_2$ | TrC | DOMINO v2 [QA4ECV] | 2007-01-01 to present |
| TES (Aura) | $O_3$ | Profile | v5 [v6] | 2005-01-01 to 2011-06-04 |





| MLS (Aura) | O$_3$, HNO$_3$ | Profile | v3.3 [v4.2] | 2005-01-01 to present |
|---|---|---|---|---|
| MOPITT (Terra) | CO | Profile | v6 NIR [v7 TIR/NIR] | 2005-01-01 to present |

## 3. Ozone observations used for evaluation

### 3.1 Ozone sondes

For evaluation of free tropospheric ozone data from the global network of ozone sondes, as collected by the WOUDC, is used, expanded with observations available from SHADOZ (Thompson et al., 2017; Witte et al., 2017) and ESRL. The observation error of the sondes is about 7–17% below 200 hPa and ±5% in the range between 200 and 10 hPa (Beekmann et al., 1994, Komhyr et al., 1995 and Steinbrecht et al., 1996). Typically, the sondes are launched once a week, but in certain periods, such as during ozone hole conditions, launches can be more frequent. Sonde launches are mostly carried out between 9:00 and 12:00 local time.

The ozone sonde network provides critical independent validation of the reanalysis products. Although the number of soundings varied for the different stations, the global distribution of the launch sites is expected to be sufficient to allow meaningful monthly to seasonal averages over larger areas. However, because of the sparseness of the ozone sonde network, we are aware that the evaluation based on ozone sonde observations can introduce large biases in regional and seasonal reanalysis performance (Miyazaki and Bowman, 2017).

The model data has been collocated with observations through interpolation in time and space. Individual intercomparisons have been aggregated on a monthly and seasonal basis. The number of stations contributing to the monthly and regional means varies over the course of the reanalysis products, and is additionally reported as this is naturally an important consideration when assessing interannual variability of ozone biases. While we present time series from 2003 onwards in our figures, where CAMS starts to provide reanalysis products, for any of the statistics we only base this on the 2005-2016 time period (unless explicitly mentioned), to allow fair intercomparison between CAMS and TCR.





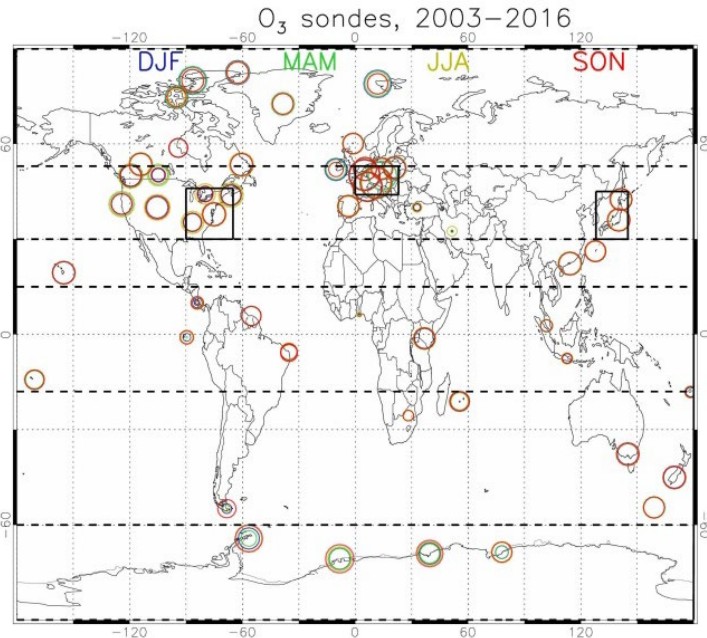

**Figure 1: Location of ozone sondes contributing to the various regions, and indicated by the black boxes and the dashed lines. The size of circles indicates the relative number of observations contributing to the statistics for December–February (blue), March–May (green), June–August (yellow), and September–November (red).**

For spatial aggregation the choice is more difficult, depending on the characteristics of the species and availability of observations. Tilmes et al. (2012) defined an aggregation approach for ozonesonde locations based on the characteristics of the observed ozone profiles. We follow in part their aggregation approach, by adopting the European, Eastern US, Japan, and Antarctic regions. For several regions, the number of measurements could be insufficient to construct meaningful aggregates. Instead we define regions for the northern hemisphere (NH) subtropics (15°N to 32°N), the tropics (18°S to 15°N), southern hemisphere (SH) mid latitudes (60°S to 18°S) and Antarctic (90°S to 60°S), and combine the NH Polar regions to a single region (60°N to 90°N), see also Figure 1.

**3.2 Surface ozone**

We evaluate surface ozone against the TOAR database (Schultz et al., 2017), which provides a globally consistent, gridded, long-term dataset with ozone observation statistics on a monthly mean basis. The TOAR database has been produced with particular attention to quality control, and representativeness of the in-situ observations, in order to establish consistent, long-term time records of observations. TOAR provides a disaggregation of rural and urban stations. For our study we use the 2°×2° gridded monthly mean dataset representative for rural stations for the 1990-2014 time period. This allows easy intercomparison with monthly mean results from the various reanalysis products.



Note that in these comparisons we used rural observations only, because any of the reanalysis model resolutions is

considered too coarse to resolve local concentration changes over highly polluted urban areas. Therefore the rural observations can be considered as more representative data for grid averaged concentrations. Nevertheless, neglecting urban observations could lead to biased evaluations particularly in cases where large fractions of the grid cells are associated to urban conditions, e.g. in megacities.

This TOAR dataset has a good global coverage, including stations over East Asia, and provides overall a constant, and good

quality controlled data record up to 2014. Nevertheless, the number of records in this database decreases significantly for various regions on the globe after 2012. Therefore in our evaluation statistics we focus on the period before 2012, considering that the reduction in available observations afterwards hampers the intercomparison of model performance between different years. Similar to the evaluation against ozone sonde observations, the statistics is computed for data from 2005 onwards.

**3.3 EMEP observations**

In order to assess the ability of the reanalysis products to represent spatial and temporal variability on a sub-seasonal and on regional scales, we additionally evaluate the reanalyses against ground-based observations from the EMEP network (obtained from http://ebas.nilu.no/) for the year 2006. Although EMEP data are also included in the TOAR data product, this analysis allows for a complementary approach, in particular the assessment of pollution events during heat waves, but also

evaluation of the diurnal cycles and spatial variability in the various products. The summer period of 2006 over Europe was characterized by a heat wave event (Struzewska and Kaminski, 2008). For this evaluation, we collocate the model output spatially and temporally to the observations. Considering the comparatively coarse horizontal resolution, which is not generally able to represent the local orography at the location of the individual observations, we match the model level with the same (average) pressure level at the location of the observations. Here we note that the CAMS reanalyses use a higher

vertical resolution than TCR. This implies that for high-altitude stations also different (higher) model levels are sampled in the CAMS reanalyses compared with TCR. After this collocation procedure, we compute temporal correlation coefficients on a seasonal basis, using temporally collocated 3-hourly mode and observational data.

**4. Evaluation against ozonesondes**

**4.1 Annually and regionally averaged profiles**

Figure 2 provides an overview of the multi annual mean ozone for the four reanalyses for the 2005–2016 time period. All reanalyses capture the observed vertical profiles of ozone from the lower troposphere to the lower stratosphere, with a regional mean bias of typically less than 8 ppb throughout the troposphere. Corresponding mean biases at 850, 650 hPa and 350 hPa are given in Tables 5-7. These multi-annual, regional mean biases are below 3.7 ppb (~7%) at 850 hPa and 4 ppb (~7%) at 650 hPa. Generally, the CAMS reanalysis shows improvement against the CAMS interim reanalysis at 650 hPa and




also 850 hPa, particularly for regions over the NH high- and mid-latitudes, as well as the SH-mid latitudes, but at the cost of
a degradation (an emerging positive bias) towards the surface. TCR-2 shows a more mixed picture in this respect. Biases
between TCR and CAMS are within a similar order of magnitude, but are not correlated in any way in sign or magnitude.
For most of the major polluted areas in the lower troposphere, the biases are lower in the CAMS reanalysis than in the TCR
reanalyses, probably due to its higher model resolution and a better chemical forecast model performance. The annual mean
ozone biases in TCR are relatively large in the tropics and SH high latitudes. After 2011, no TES tropospheric ozone
measurements were assimilated, which could lead to enhanced ozone biases, as demonstrated by Miyazaki et al. (2015).
Assimilation of MLS measurements does not noticeably influence the tropospheric ozone analysis in the tropics. Note, in the
NH subtropics and tropics, the ozonesonde network is sparse, while the spatial and temporal variability of ozone is large,
which could limit our understanding of the generalized reanalysis performance (Miyazaki and Bowman, 2017). At high
latitudes, the large diversity in the reanalysis ozone could be associated with the lack of direct tropospheric ozone
measurements in all of the systems. In the TCR systems, TES ozone data was excluded poleward of 72 degree because of the
small retrieval sensitivity, limiting data assimilation adjustments at high latitudes in the troposphere.

Overall, this evaluation shows that the biases from these reanalysis products are much smaller than those reported from
recent CTM simulations. E.g. Young et al. (2013) present median biases across ACCMIP model versions at 700 (500) hPa
up to 10 (15)%, depending on the region. This demonstrates that the reanalysis of tropospheric ozone fields is generally well
constrained by assimilated measurements for the globe.

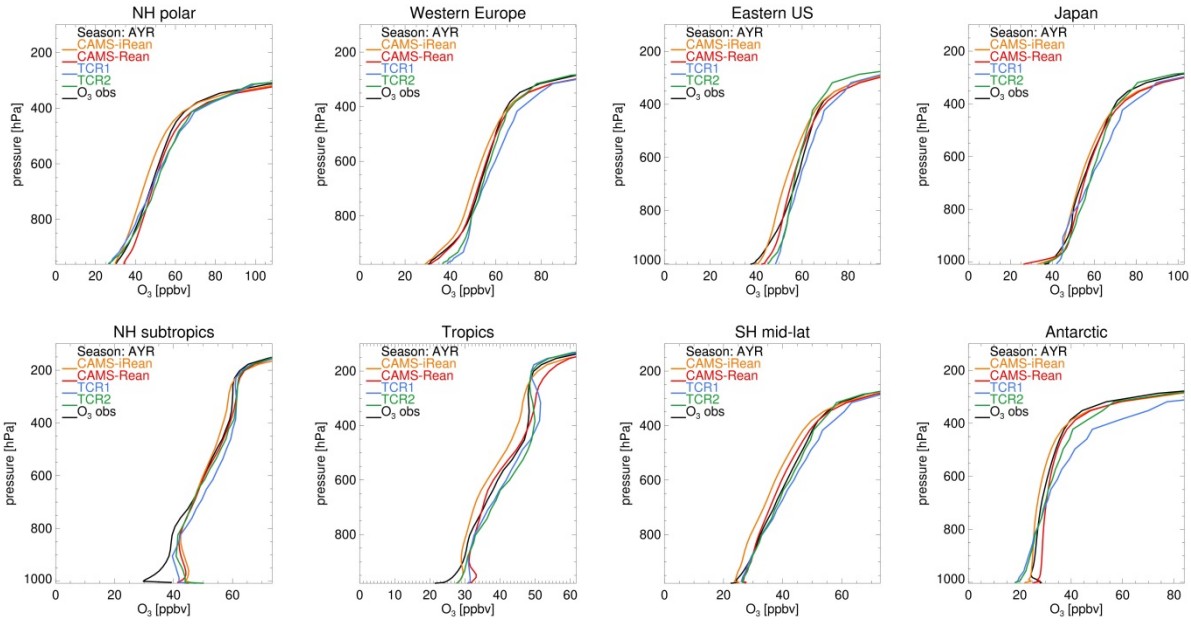




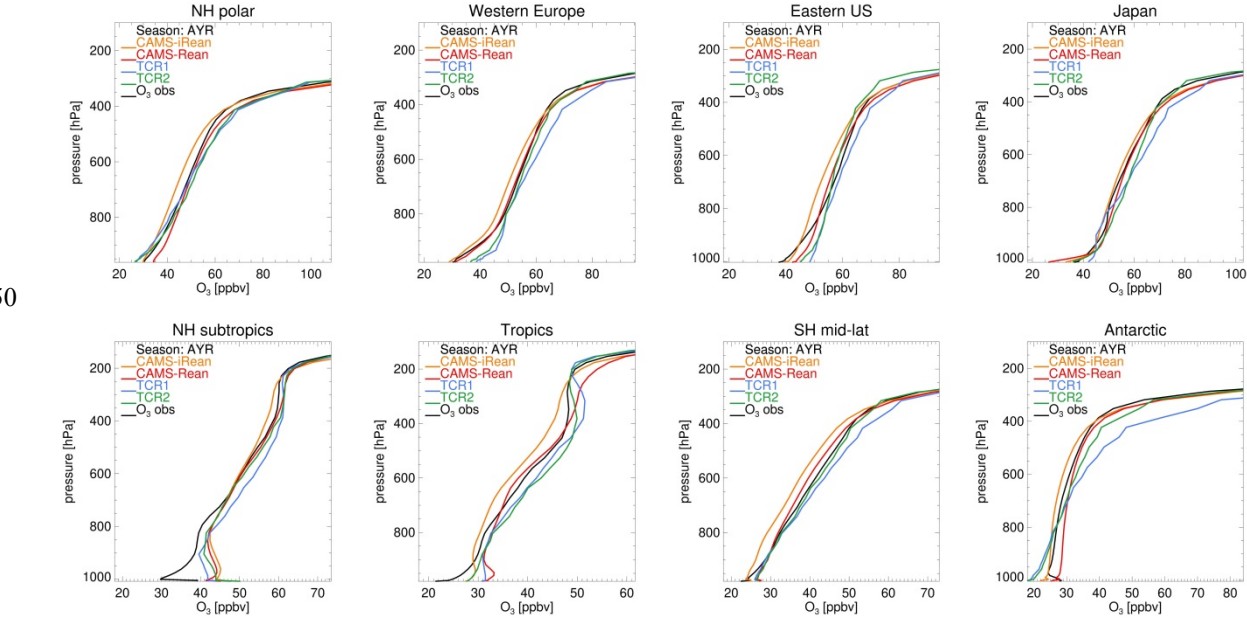

**Figure 2: Evaluation of regional mean multi-annual mean O$_3$ profiles against ozone sondes, averaged over the 2005–2016 time period.**

**Table 5. Evaluation of ozone mean bias, RMSE (both in ppb) and temporal correlation R for the four reanalysis products at 850 hPa against sondes, computed for various regions, for the 2005-2016 time series.**

| **MB** | NH polar | Western Europe | Eastern US | Japan | NH subtropics | Tropics | SH mid lat | Antarctic |
|---|---|---|---|---|---|---|---|---|
| CAMS-iRean | -1.8 | -1.9 | -2.3 | -1.3 | 3.7 | -0.2 | -3.1 | -1.3 |
| CAMS-Rean | 2.7 | 0.4 | 0.6 | 1.2 | 3.2 | 1.9 | -0.0 | 2.0 |
| TCR-1 | -1.6 | 3.3 | 2.4 | -1.3 | 2.5 | 1.7 | 0.6 | -1.4 |
| TCR-2 | 0.4 | 2.7 | 2.6 | 2.2 | 2.3 | 2.3 | 0.6 | -0.9 |
| **RMSE** | NH polar | Western Europe | Eastern US | Japan | NH subtropics | Tropics | SH mid lat | Antarctic |
| CAMS-iRean | 3.7 | 3.3 | 4.2 | 5.1 | 5.1 | 3.6 | 3.8 | 4.2 |
| CAMS-Rean | 4.1 | 2.3 | 3.0 | 5.1 | 4.8 | 4.4 | 2.3 | 3.7 |
| TCR-1 | 3.2 | 4.7 | 4.7 | 9.1 | 5.1 | 4.0 | 2.5 | 4.2 |
| TCR-2 | 3.6 | 3.7 | 4.7 | 6.8 | 4.5 | 4.4 | 2.2 | 4.7 |



| R | NH polar | Western Europe | Eastern US | Japan | NH subtropics | Tropics | SH mid lat | Antarctic |
|---|---|---|---|---|---|---|---|---|
| CAMS-iRean | 0.70 | 0.92 | 0.86 | 0.86 | 0.92 | 0.66 | 0.90 | 0.84 |
| CAMS-Rean | 0.79 | 0.94 | 0.90 | 0.86 | 0.91 | 0.61 | 0.92 | 0.91 |
| TCR-1 | 0.75 | 0.88 | 0.80 | 0.53 | 0.89 | 0.64 | 0.92 | 0.81 |
| TCR-2 | 0.57 | 0.93 | 0.84 | 0.75 | 0.89 | 0.62 | 0.91 | 0.72 |

**Table 6. Same as Table 5 but for 650 hPa.**

| MB | NH polar | Western Europe | Eastern US | Japan | NH subtropics | Tropics | SH mid lat | Antarctic |
|---|---|---|---|---|---|---|---|---|
| CAMS-iRean | -3.1 | -2.7 | -4.0 | -1.3 | 0.0 | -1.6 | -3.4 | -1.4 |
| CAMS-Rean | 0.9 | -0.5 | -1.6 | 0.8 | 0.5 | 0.5 | -1.2 | 1.5 |
| TCR-1 | 0.9 | 2.2 | 1.7 | 4.2 | 3.0 | 3.1 | 2.1 | 2.4 |
| TCR-2 | 2.4 | 1.3 | -0.5 | 3.3 | 0.6 | 3.8 | 1.2 | 1.7 |
| **RMSE** | NH polar | Western Europe | Eastern US | Japan | NH subtropics | Tropics | SH mid lat | Antarctic |
| CAMS-iRean | 4.5 | 3.3 | 4.9 | 4.7 | 4.3 | 3.8 | 4.1 | 4.2 |
| CAMS-Rean | 3.0 | 2.2 | 3.2 | 5.0 | 4.7 | 3.2 | 2.7 | 3.4 |
| TCR-1 | 3.9 | 5.0 | 5.0 | 8.6 | 6.1 | 4.6 | 3.6 | 4.4 |
| TCR-2 | 3.9 | 2.7 | 3.4 | 6.9 | 4.6 | 5.2 | 2.7 | 4.8 |
| **R** | NH polar | Western Europe | Eastern US | Japan | NH subtropics | Tropics | SH mid lat | Antarctic |
| CAMS-iRean | 0.81 | 0.95 | 0.90 | 0.88 | 0.90 | 0.71 | 0.91 | 0.81 |
| CAMS-Rean | 0.88 | 0.94 | 0.91 | 0.85 | 0.88 | 0.75 | 0.90 | 0.89 |
| TCR-1 | 0.68 | 0.74 | 0.73 | 0.64 | 0.84 | 0.71 | 0.90 | 0.82 |
| TCR-2 | 0.81 | 0.93 | 0.87 | 0.76 | 0.89 | 0.68 | 0.90 | 0.71 |





**Table 7. Same as Table 5 but for 350 hPa.**

| MB | NH polar | Western Europe | Eastern US | Japan | NH subtropics | Tropics | SH mid lat | Antarctic |
|---|---|---|---|---|---|---|---|---|
| CAMS-iRean | 2.9 | 1.6 | 1.3 | 4.4 | -1.3 | -1.4 | -1.7 | 3.6 |
| CAMS-Rean | 6.1 | 2.3 | 2.6 | 5.4 | 1.2 | 2.5 | 0.2 | 3.5 |
| TCR-1 | 4.1 | 6.0 | 3.7 | 6.7 | 1.6 | 4.3 | 4.3 | 22.3 |
| TCR-2 | 2.9 | 1.1 | -4.0 | -1.1 | 1.6 | 0.6 | 0.6 | 5.0 |
| **RMSE** | NH polar | Western Europe | Eastern US | Japan | NH subtropics | Tropics | SH mid lat | Antarctic |
| CAMS-iRean | 9.3 | 6.2 | 10.6 | 12.1 | 6.5 | 4.4 | 3.8 | 6.8 |
| CAMS-Rean | 8.5 | 4.7 | 9.0 | 11.8 | 7.1 | 4.8 | 3.4 | 5.8 |
| TCR-1 | 8.7 | 9.1 | 11.7 | 12.4 | 7.0 | 6.6 | 6.3 | 23.6 |
| TCR-2 | 8.1 | 5.0 | 9.9 | 8.7 | 7.6 | 5.7 | 3.8 | 7.3 |
| **R** | NH polar | Western Europe | Eastern US | Japan | NH subtropics | Tropics | SH mid lat | Antarctic |
| CAMS-iRean | 0.85 | 0.88 | 0.76 | 0.79 | 0.82 | 0.76 | 0.89 | 0.76 |
| CAMS-Rean | 0.92 | 0.94 | 0.83 | 0.79 | 0.77 | 0.78 | 0.89 | 0.83 |
| TCR-1 | 0.85 | 0.81 | 0.73 | 0.78 | 0.78 | 0.67 | 0.81 | 0.59 |
| TCR-2 | 0.87 | 0.89 | 0.80 | 0.82 | 0.73 | 0.68 | 0.86 | 0.74 |





**Table 8. Same as Table 5 but for tropospheric columns in units DU.**

| MB | NH polar | NH mid latitudes | Tropics | SH mid latitudes | Antarctic |
|---|---|---|---|---|---|
| CAMS-iRean | -1.0 | -1.3 | -0.5 | -1.2 | 0.5 |
| CAMS-Rean | 0.9 | 0.4 | 0.5 | -0.2 | 0.6 |
| TCR-1 | 0.3 | 1.7 | 1.2 | 1.4 | 2.1 |
| TCR-2 | 1.0 | 0.9 | 1.1 | 0.8 | 0.8 |
| **RMSE** | NH polar | NH mid latitudes | Tropics | SH mid latitudes | Antarctic |
| CAMS-iRean | 2.1 | 1.6 | 1.1 | 1.5 | 1.7 |
| CAMS-Rean | 1.7 | 0.8 | 1.1 | 0.9 | 1.5 |
| TCR-1 | 1.5 | 2.1 | 1.5 | 1.8 | 2.6 |
| TCR-2 | 1.7 | 1.2 | 1.5 | 1.3 | 1.9 |
| **R** | NH polar | NH mid latitudes | Tropics | SH mid latitudes | Antarctic |
| CAMS-iRean | 0.81 | 0.95 | 0.77 | 0.84 | 0.76 |
| CAMS-Rean | 0.89 | 0.97 | 0.74 | 0.89 | 0.84 |
| TCR-1 | 0.82 | 0.90 | 0.75 | 0.85 | 0.76 |
| TCR-2 | 0.88 | 0.96 | 0.76 | 0.78 | 0.66 |


## 4.2 Time series of zonally averaged O₃ tropospheric columns

Collocated tropospheric columns have been compared to tropospheric columns derived from the sonde observations. An intercomparison of the monthly and zonally mean tropospheric columns sampled at the observations is given in Figure 3. Note that the figures also contain information on the number of sonde stations that are included in the evaluation for individual months. Here the tropopause has been defined as the altitude where ozone exceeds 150 ppb for each of the individual products.




Outside the polar regions all reanalyses capture the magnitude of the zonal mean tropospheric column to within a root-mean-square (RMS) of 0.8–2.1 DU depending on the reanalysis product, see also Table 8. Largest uncertainties are found for the polar regions, with the RMS ranging between 1.5 (CAMS-REAN) to 2.6 (TCR-1) DU, corresponding to up to ~15% of the

average $O_3$ tropospheric column. In contrast, in the tropics the RMS is up to 1.1–1.5DU, or ~5% of the average $O_3$ tropospheric column. Except for the NH mid latitudes and Antarctic region the seasonal cycle in both model and observations is not very pronounced. The temporal correlation between modelled and observed tropospheric columns is correspondingly highest (R>0.90) for the NH mid-latitudes, but still relatively low for the Antarctic region (R<0.84) for all reanalyses. This relatively poor temporal correlation over the Antarctic, despite the strong seasonal cycle, does indicate

difficulties of the reanalyses to reproduce a consistent seasonality over the full time series, as described in more detail in the following sections.

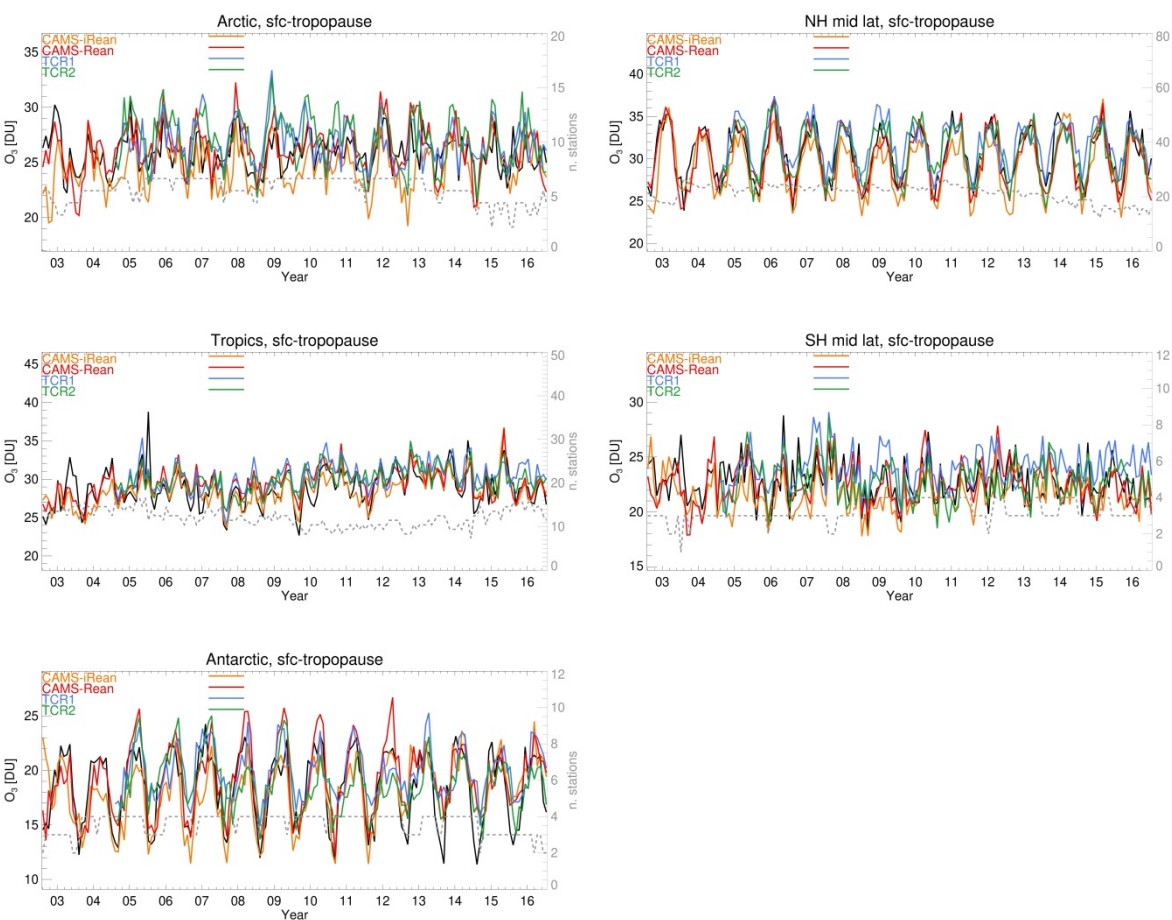

**Figure 3: Evaluation of zonally averaged monthly mean tropospheric columns against sonde observations. Observations are in black. The gray dashed line refers to the number of stations that contribute to the statistics (right vertical axis).**



### 4.3 Time series of regionally averaged O₃ biases at multiple altitudes

Figure 4 shows time series of monthly mean ozone biases against ozone sondes at three pressure levels (~850, 650, and 350

hPa), aggregated for the predefined regions. These figures give an indication of the stability of the reanalyses against sonde observations during the 2003–2016 time period. The corresponding timeseries with monthly mean concentration values, showing the seasonal cycle, is given in Figure S1 in the Supplementary material. As in the previous section, persistent changes in the number of stations may contribute to changes in biases over the course of the fourteen year time interval. The mean bias, RMSE and temporal correlation for each of these time series have been given in Tables 5-7. Based on these

evaluations we note the following: Over the **NH polar** region, CAMS-REAN shows a small positive bias in the lower troposphere (2.7 ppb at 850 hPa for the 2005-2016 multi-annual mean), particularly during the springtime (5.0 ppb when averaged over MAM). During 2003 and 2004 both CAMS reanalyses show anomalously low springtime ozone, different to the rest of the time period, particularly at ~350 hPa. CAMS-iREAN shows a large offset compared to observations and CAMS-REAN in 2003. This is attributed to the assimilation of GOME observations in CAMS-iREAN, which has been

omitted in CAMS-REAN (Inness et al., 2019). The different error statistics for 2003 over the Arctic compared to later years is furthermore attributed to the use of early SCIAMACHY and NRT MIPAS O₃ retrievals, which are of poorer quality than the OMI MLS observations which have been used from August 2004 onwards, and reprocessed MIPAS data used from January 2005 onwards.

Furthermore, before 2014 CAMS-iREAN shows lower values than CAMS-REAN, while for 2014 to 2016 the two CAMS

reanalyses are much more alike. This offset before 2014 results in a slight negative bias against observations at ~850 hPa over the Arctic, and a significant negative bias at ~650 hPa. The TCR reanalyses underestimate the lower tropospheric ozone after 2011, which could be associated with the lack of TES measurements during the recent years. At higher altitudes (650 and 350 hPa) differences between the reanalyses are relatively smaller.  On average at 650 hPa CAMS-iREAN shows a slight underestimation (-3.1 ppb), while CAMS-REAN and TCR-1 bias is below 1 ppb, and slightly larger for TCR-2 (2.4

ppb). At 350 hPa all reanalysis products perform overall similar. At this altitude a considerable inter-annual variability is visible in the observations, which appears to be well captured by the reanalysis products, with temporal correlations in the order R=0.85 (for TCR-1) to R=0.92 (CAMS-REAN). Also both the observations and reanalyses indicate an upward trend of tropospheric ozone in the UTLS, as also confirmed by Williams et al., (2019).

Over **Western Europe** the CAMS reanalyses show good correspondence to the observations at 850 hPa from 2004 onwards,

with mean biases of -1.9 (CAMS-iREAN) and 0.4 ppb (CAMS-REAN). The TCR reanalyses overestimate ozone at lower altitudes, particularly in TCR-1 before 2010, which shows positive biases at 850 hPa of up to ~15 ppb, with an average over the full time period of  3.3 ppb. Such overestimates suggest a strong influence of the forecast model performance for the boundary layer (e.g., mixing and chemistry), while the optimization of the emission precursors was not sufficient to improve the lower tropospheric ozone analysis. At ~650 and ~350 hPa, the chemical reanalyses reproduced well the observed



seasonal and interannual variations. As an exception, TCR-1 overestimates ozone for some cases, especially in winter. In contrast, the CAMS reanalyses show average (absolute) biases less than 3.3 ppb at all pressure levels.

Over the **Eastern US**, all the reanalysis products show similar RMSE values at ~850 hPa (3.0–4.7 ppb), which is associated with positive model biases, mostly during summer by 0.3–6.8 ppb. Such model biases have also been reported in other model studies (e.g., Travis et al., 2016), which could be associated with model errors, for instance, excessive vertical mixing

and net ozone production in the boundary layer. The annual mean bias for the reanalyses ranges between -2.3 and 2.6 ppb. A decrease in the observed ozone concentrations at ~850 hPa after 2014, associated to a change in the number of contributing stations in this evaluation, leads to a general and consistent over-estimate in all of the reanalyses. A similar agreement with the observations was found in the middle troposphere compared to the lower troposphere, with RMSE ranging between 3.0 and 4.9 ppb, while at ~350 hPa the RMSE ranges between 9 and 11.7 ppb.

Over **Japan**, all reanalyses on average overestimate ozone at 850 hPa and 650 hPa before 2011, with relatively large positive biases in TCR-1 and TCR-2 at 650 hPa (7.9 and 6.9 ppb, respectively, when averaged for the 2005-2010 time period). From 2011 onwards the correspondence with observations improves remarkably, despite the lack of TES measurements in TCR from June 2011 onwards.

In the **tropics**, all of the reanalysis products overestimate ozone before 2012, with large positive biases in CAMS-REAN and

TCR-1 at 850 hPa. Interestingly, both CAMS reanalyses show a strong peak in ozone at 850 hPa (and to lesser extent at 650 hPa) during the second half of 2015, with a zonally averaged overestimation of up to 20 ppb. This is associated to the strong El Niño conditions, and this particular spike was attributed to an over-estimate of ozone observed at the Kuala Lumpur station for October 2015. Here exactly the grid box affected by the extreme fire emissions in Indonesia for this period (Huijnen et al., 2016), as prescribed by the daily GFAS product, has been sampled. This peak appears much weaker in TCR,

probably owing to the lack of direct ozone measurements together with underestimated ozone production and coarser model resolution. At 650 hPa, the TCR reanalyses overestimate ozone almost throughout the reanalysis period (by 3.1–3.8 ppb on average), whereas the CAMS-Rean shows closer agreement with the observations (mean bias = 0.5 ppb, RMSE = 3.2 ppb). At ~350 hPa, the TCR-2 shows improved agreement compared with the earlier TCR-1, as confirmed by improved mean bias (from 4.3 to 0.6 ppb) and RMSE (from 6.6 to 5.7 ppb) although the temporal correlation remains relatively low.

Over the **SH mid-latitudes** a remarkably good correspondence is generally obtained for all reanalyses, but particularly CAMS-REAN and TCR-2, throughout the troposphere. This is marked by the lowest magnitudes for RMSE and highest for the temporal correlations, for any of the three altitude ranges compared to the statistics in other regions. Nevertheless, CAMS-iRean still underestimates ozone before 2012 in the lower and middle troposphere, whereas TCR-1 overestimates it particularly at 332 hPa after 2010. In contrast, large diversity among the systems is seen over the **Antarctic**. As in the Arctic

region, free tropospheric $O_3$ in the CAMS reanalyses is comparatively poorly constrained during 2003, as consequence of the use of the NRT data product from MIPAS and early SCIAMACHY data in the assimilation. Also in the period between the end of March and the beginning of August 2004 no profile data were available for assimilation, leading to a temporary degradation in the reanalysis performance.








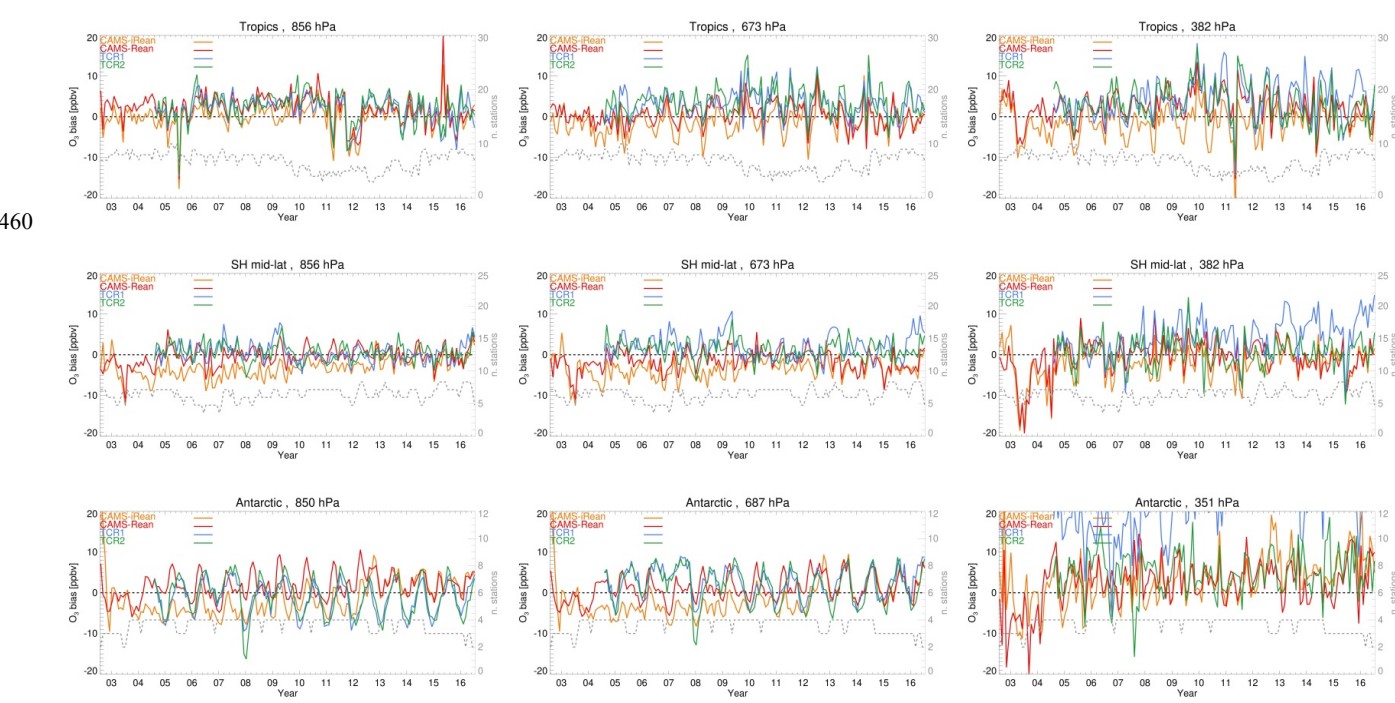

**Figure 4. Time series of regionally and monthly aggregated ozone biases at different altitudes (850, 650 and 350 hPa), sampled at ozone sonde locations, against ozone sonde observations (black).**



Before 2013, CAMS-iREAN underestimates the low ozone values in the lower and middle troposphere during austral spring, while CAMS-REAN overestimates it during austral winter. Afterwards, both systems show very similar results, also in overall better agreement with the observations, even though an overestimate during austral spring remains. Reasons for the
change in behaviour in CAMS-iREAN is a change MLS version from v2 to v3.4 after 2012. Furthermore both CAMS-iREAN and CAMS-REAN are affected by a change from 6L SBUV to 21L NRT data in January and July 2013 respectively, which appears to contribute significantly to the changes in the bias. The TCR reanalyses largely underestimate ozone during austral summer and autumn in the lower troposphere. At 332 hPa, TCR-1 substantially overestimates ozone throughout the year because of large model biases and the lack of observational constraints, which was resolved in TCR-2.

**5. Validation against TOAR surface observations**

We evaluated the reanalyses against monthly mean, gridded surface observations filtered for measurements performed at rural sites, as compiled in the TOAR project (Schultz et al., 2017). These evaluations reveal the ability of the reanalysis products to reproduce near-surface background ozone concentrations in terms of mean value and variability, both temporally, on seasonal to annual time scale, and spatially, for various regions over the globe.

**5.1 Multi-annual mean**

Figure 5 shows a map with multi-annual mean ozone observations from the TOAR database, for the 2005–2012 time period, as well as the corresponding biases in surface ozone for the reanalysis products. Detailed maps for North America, Europe and East Asia are given in Figure S2 in the Supplementary Material, while the corresponding regional mean biases are given in Table 9.
The TCR-reanalyses show significant positive biases for many regions, with multi-annual mean biases of 11.0 ppb and 6.8 ppb over the Eastern and Western US, and 6.7 ppb over Europe in TCR-2. These biases can mainly be attributed to model errors. Mean biases in the CAMS-reanalyses are generally smaller (1.5 ppb and -0.2 ppb for Eastern and Western US, respectively, -1.8 ppb for Europe), but still show substantial spatial variations, as quantified by the root-mean-square of the multi-annual mean differences across the various regions, which is 8.9 ppb and 6.1 ppb for Eastern and Western US, and 5.6
ppb over Europe for the CAMS Reanalysis (18, 11 and 11 ppb for TCR-2 for these regions). The mean bias is negative over the Arctic, Europe and the Western US and positive over East Asia and Southeast Asia in both versions of the CAMS reanalyses. The positive regional mean biases over the major polluted regions are reduced by 35 to 55% in TCR-2 as compared with TCR-1. Likewise, the negative biases over the Arctic, Europe, the Western US, and SH mid and high latitudes are reduced by more than 25% in CAMS-REAN as compared with CAMS-iREAN, illustrating overall
improvements for the newer reanalyses.



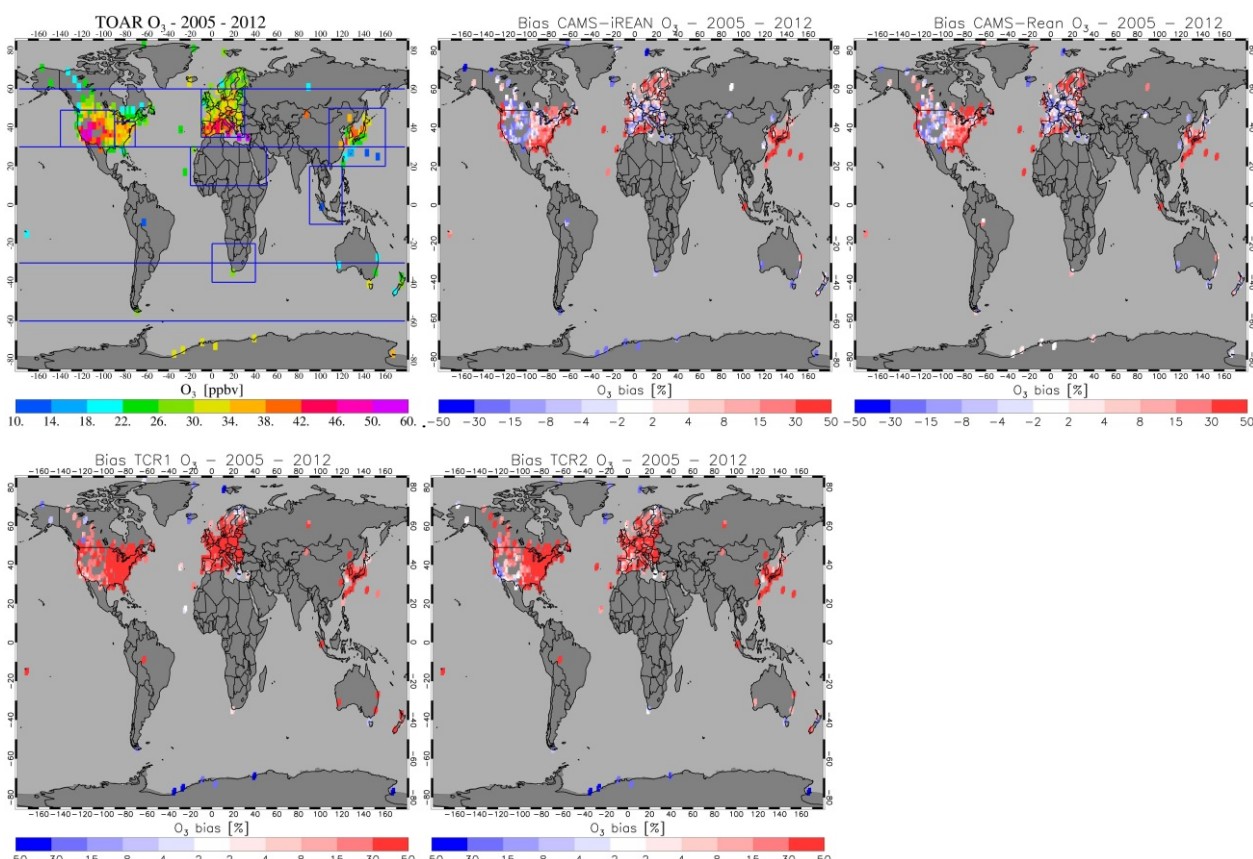

**Figure 5: Multi-annual (2005-2012) mean surface ozone from TOAR (upper left), along with corresponding relative mean bias for the reanalyses.**

**Table 9. Mean bias (ppb) for the reanalyses against TOAR monthly mean, regional mean observations for the 2005-2012 time period, as sampled for observations in the specified regions indicated in Figure 4.**

|  | Arctic | Europe | Eastern US | Western US | Southeast Asia | East Asia | SH mid-latitudes | Antarctica |
|---|---|---|---|---|---|---|---|---|
| CAMS-iREAN | -4.5 | -2.4 | 0.1 | -1.9 | 5.6 | 4.5 | -2.2 | -3.5 |
| CAMS-REAN | -1.5 | -1.8 | 1.5 | -0.2 | 6.7 | 2.7 | -0.1 | 1.1 |
| TCR-1 | -1.8 | 11.2 | 17.2 | 12.9 | 15.8 | 10.5 | 3.6 | -5.7 |
| TCR-2 | -2.3 | 6.7 | 11.0 | 6.8 | 7.4 | 7.6 | 1.5 | -5.4 |





## 5.2 Variability in regionally averaged surface ozone


Figure 6 presents the temporal variability from the ozone reanalyses against those from the TOAR observations at the surface. The corresponding time series are given in Figure S3 in the Supplementary Material. Over the Arctic, the general pattern in the seasonal variations is captured for all reanalyses (R=0.58-0.72), although they all underestimate the increased ozone values during boreal spring. A remarkable positive bias is seen in springtime 2013 in CAMS-REAN, as was also

visible in the bias plots against ozone sondes in the lower free troposphere, Figure 4.

Over Europe and the US, the CAMS reanalyses show the closest agreement with the observations (MB= -2.4 – 1.5 ppb, R>0.8), furthermore showing reduced model biases in boreal winter and spring in CAMS-REAN compared with the CAMS-iREAN. The TCR reanalyses exhibit large positive biases in boreal summer over Europe and the US regions (6.7 – 17 ppb), with significantly lower biases in TCR-2. Over East Asia, all the reanalyses show positive biases in the range of 2.7 ppb

(CAMS-REAN) to 10.5 ppb (TCR-1) and fail to reproduce the minimum concentrations in autumn. Still the temporal correlations are similar to most other regions (R=0.79 – 0.83), associated with the stable seasonal cycle in both the reanalyses and observations. Over Southeast Asia, positive biases exist throughout the period, which are largest in TCR-1. For this region the TCR-reanalyses show lower temporal correlations (R=0.39 – 0.49) compared to the CAMS reanalyses (R=0.68). Significant changes in the surface ozone biases are found in the TCR reanalyses over the SH mid latitudes, with

reduced values after 2010.

The CAMS reanalyses capture well the seasonal cycle over the SH mid latitudes and Antarctic (R=0.89 – 0.96), while CAMS-REAN shows a positive bias during austral winter (JJA), particularly during 2005-2013. The TCR reanalyses show a significant negative bias throughout the year except during Austral summer which results in lower temporal correlations (R~0.68).

The free tropospheric intercomparison at different altitudes, as reported in Tables 5 to 7, already indicated larger biases with decreasing altitude near the surface. This can be understood as near-surface ozone concentrations are less well constrained by the satellite data products used in the assimilation, and they depend strongly on local conditions such as precursor emissions, deposition, vertical mixing, and chemistry, which are difficult to parameterise at the model grid scale (Sekiya et al., 2018).

An important example of a driver for local variability is the emissions from forest fires which in the CAMS reanalyses are provided through daily-varying GFAS emissions. This has been shown to capture to a good degree the carbon monoxide and aerosol from fire plumes, although larger uncertainties exist in the NOx emissions, e.g. Bennouna et al. (2019).





**Figure 6: scatter plot of surface ozone against TOAR observations for the 2005-2012 regionally averaged, monthly mean time series. Also the mean bias and temporal correlation is given.**






### 5.3 Interannual variability of regionally averaged surface ozone

We compute the interannual variability (IAV) by subtracting the 2005-2012 multi-annual monthly, regional mean surface ozone from its corresponding instantaneous monthly, regional mean value, both for the reanalyses and for the TOAR observations, see Figure 7. By doing so, we remove the model bias, as well as the seasonal cycle. No clear long-term trends are visible in the regional mean surface ozone concentrations. Nevertheless, the observations reveal distinct deviations from the 8-year mean value, which point at temporary anomalies in meteorological conditions and/or emissions. Note that large fluctuations in the time series can also occur due to changes in the observation network. Therefore, when evaluating the temporal correlations between observed and modelled anomalies we exclude individual months with low data coverage, defined as months where the number of grid boxes with observations is less than half of its average number for the complete time series.

Overall, the reanalysis anomalies are in reasonable general agreement with those seen in the observations, with better skill for regions at low latitudes compared to those at high latitudes. Also for 2003–2004 the CAMS reanalyses mostly show larger deviations than justified from the observations, particularly the first months for CAMS-iREAN. This is attributed to the inconsistencies in the assimilated satellite retrieval products as already described. Also the observed positive anomaly associated to the 2003 heatwave period over Europe is therefore not equally seen from the CAMS reanalyses, but with an offset (see also Bennouna et al., 2019).  For later years, the magnitude of the IAV corresponds better to the observations. Over the Arctic the temporal correlation is generally low (R<0.33). For Europe CAMS-REAN shows a largest correlation (R=0.49). For the Eastern US region all reanalyses follow an extended dip during 2009, as seen from the observations, and also a second dip during 2013, particularly captured by TCR-2, also resulting in relatively good temporal correlations (R=0.4 – 0.64). Also, in the Western US the temporal correlations are acceptable (R=0.42 – 0.56). Over East Asia the correlations are relatively high (R=0.56 – 0.75), and likewise for the station in Indonesia (southeast Asia) with R=0.45 – 0.63. Here all reanalyses capture the increases in surface ozone in early 2005 and late 2006, and the decrease in 2010.

Over the SH mid latitudes and Antarctic the ozone reanalyses show overall a relatively poor temporal correlation (R<0.37), particularly for TCR (R<0.23). For these regions the TCR reanalyses show larger anomalies during 2007-2009 as compared with observations, whereas the CAMS reanalyses show larger anomalies from 2012 onwards. Figure 5 suggests that this is particularly caused by the change in system behaviour after 2012, as already described in Sec 5.2 evaluating the tropospheric ozone over the Antarctic. As was the case there, for surface ozone the CAMS reanalyses in fact show a better match to the observations from 2013 onwards.

In conclusion, the reanalyses considered here show some skill to capture IAV in monthly mean ozone surface concentrations, in particular for the tropical, sub-tropical and NH mid-latitude regions. In these regions the signal of the observed ozone variability is also larger than for the comparatively stable Arctic conditions. Here the performance is hampered due to changes in the overall model bias over time.



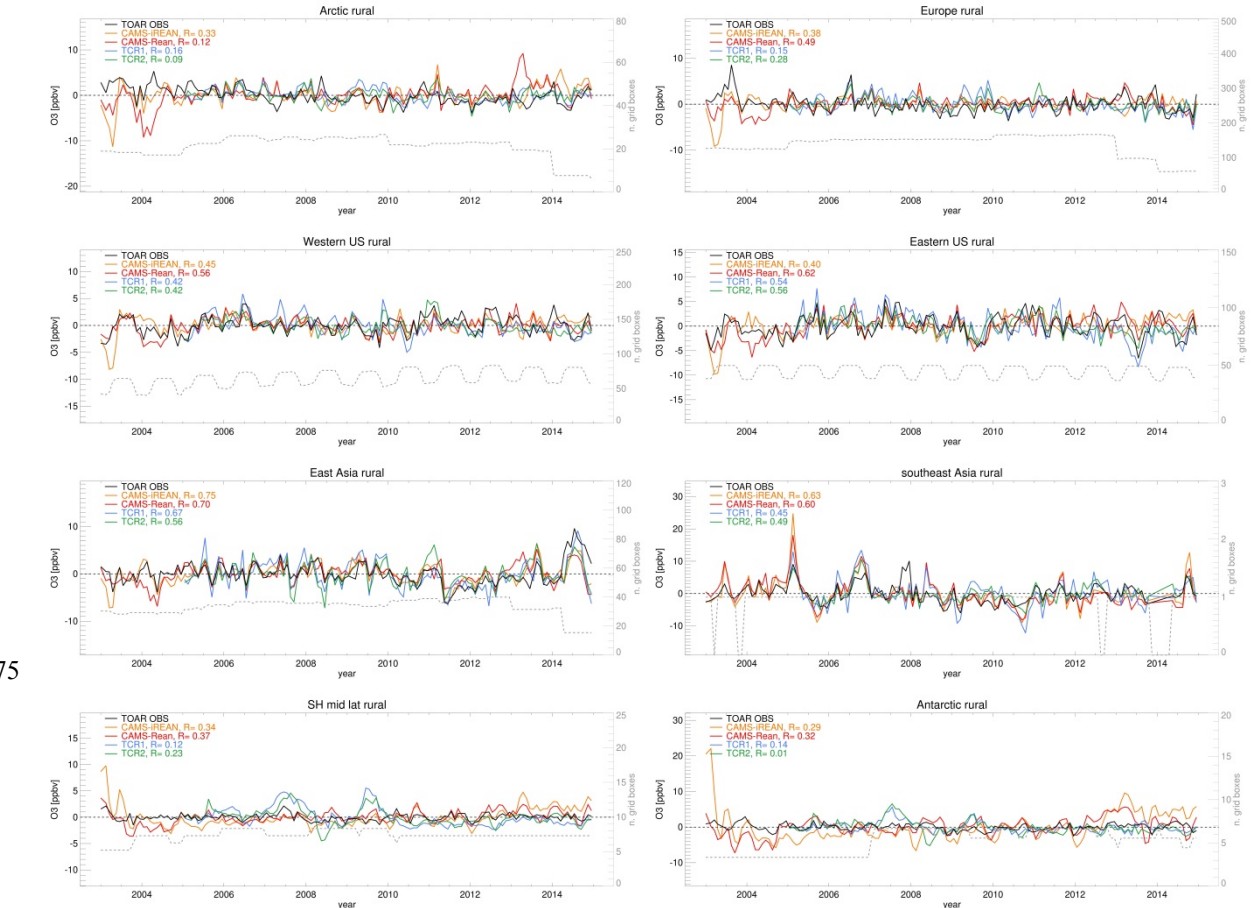

**Figure 7: Time series of regional, monthly mean ozone anomalies against those derived from the TOAR observations. The dashed line indicates the number of TOAR 2°×2° grid boxes that contribute to the statistics. Also the temporal correlation as computed for the 2005-2014 time series is given.**

## 6. Evaluation of surface ozone in 2006

To assess the ability of reanalyses to cope with local situations, and specific meteorological conditions, we analysed their performance over Europe in 2006, with a focus on the ability to capture the diurnal and synoptic variability during the heat wave event that affected large parts of Europe during July 2006 (Struzewska and Kaminski, 2008). Here we use the ground-based observations from the EMEP network. For this evaluation we note that these large-scale models do not represent local orography. Therefore we select the appropriate model level depending on its pressure level, which is representative for mean pressure at the observation site (Flemming et al., 2009). Figure 8 presents the evaluation at two EMEP stations in Great Britain, during July 2006, illustrating the general performance of the reanalyses for this situation. The Lullington Heath station (50.8° N, 0.2° E, 120 m.a.s.l.) is located in a Nature Reserve area, near the coast south of London. Great Dun Fell observatory (54.7° N, 2.4° W, 847 m.a.s.l) is located on a mountain summit, approximately 15 km north of Manchester. Both





stations show enhanced levels of ozone in the first part of July, as well as during 16-20 July. In contrast to Great Dun Fell,
Lullington Heath shows a pronounced diurnal cycle. For this evaluation, the reanalyses are sampled at different model levels
(see figure caption). Note that the TCR reanalyses have fewer model levels towards the surface than the CAMS reanalyses.
All reanalyses capture both the diurnal and synoptic variability with a significant improvement in TCR-2 compare to TCR-1,
while the CAMS reanalyses are more alike. Particularly for Lullington Heath, the CAMS reanalyses and TCR-2 show
remarkably small biases (MB < 3.6 ppb). Also at Great Dun Fell the synoptic variability is generally well captured,
particularly for the CAMS reanalyses and TCR-2.

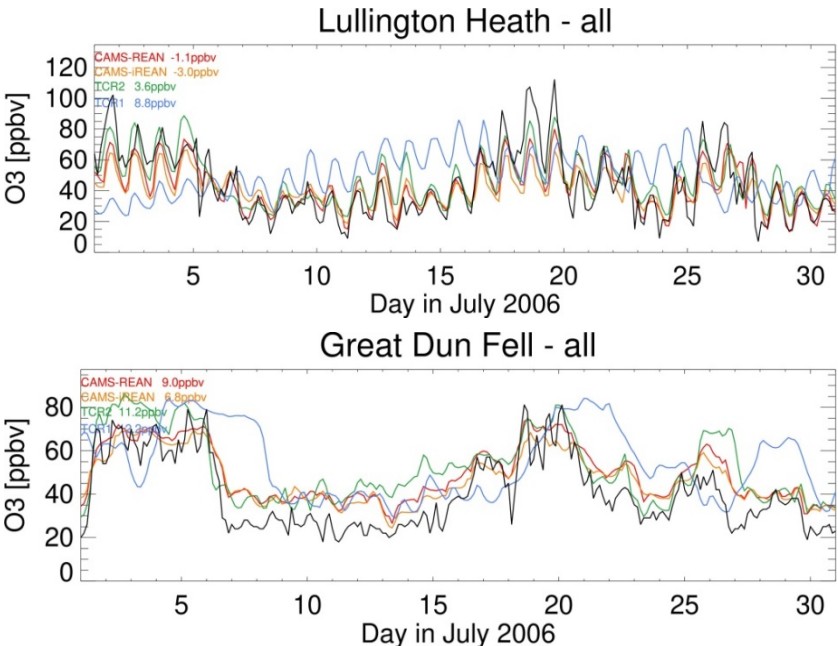

**Figure 8: Time series of reanalyses against ozone observations at two EMEP stations in Great Britain during July 2006: Lullington Heath (left, 120m a.s.l., model level 3 (CAMS) and 1 (TCR)) and Great Dun Fell (right, 847m a.s.l., model level 8 (CAMS) and 3 (TCR)). Also given are mean biases.**

A more quantitative assessment of the models' ability to capture the ozone variability is presented in Figures 9 and 10, which
show a graphical presentation of the temporal correlation coefficient at EMEP stations for December-January-February
(DJF) and June-August (JJA) 2006, computed interpolating the model and observational results onto a common 3-hourly
time frequency.

In the DJF period, regionally averaged correlation coefficients range from 0.45 (TCR-1) to 0.58 (CAMS-iREAN).
Comparatively high correlations were found over western Europe (particularly over the southern part of Britain), with R>0.8
for the CAMS reanalyses, and R>0.6 for TCR. The lower correlations over the regions in the TCR reanalyses could be
associated with its coarser model resolution.





For the summer period (JJA, Figure 10), temporal correlations are overall higher than in the winter period, most markedly by better correlation statistics over south-western, eastern and northern Europe. This is due to the more pronounced diurnal cycle during summer and results in generally consistent correlation over any of the stations across the European domain.

The average values range between R=0.61 (TCR-1) and 0.68 (CAMS-iREAN). Only stations sampling ozone around the Mediterranean are consistently poorly captured, with R<0.5.

Temporal correlations for the MAM and SON seasons are in-between those for DJF and JJA, the CAMS-REAN correlations are on average lower by ~0.02 than those of CAMS-iREAN, while TCR-2 has systematically improved temporal correlation by 0.02–0.05 over TCR-1.

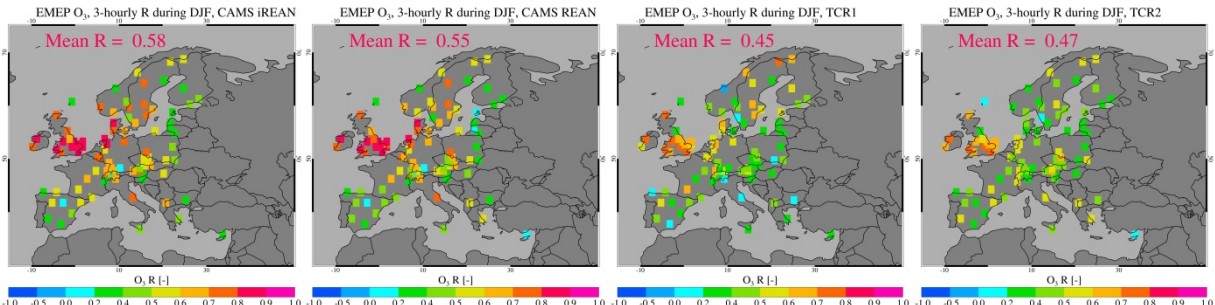


**Figure 9: Correlation coefficients computed for 3-hourly DJF 2006 at EMEP stations for the four reanalyses. The mean value, based on correlations computed for all individual stations, is given.**

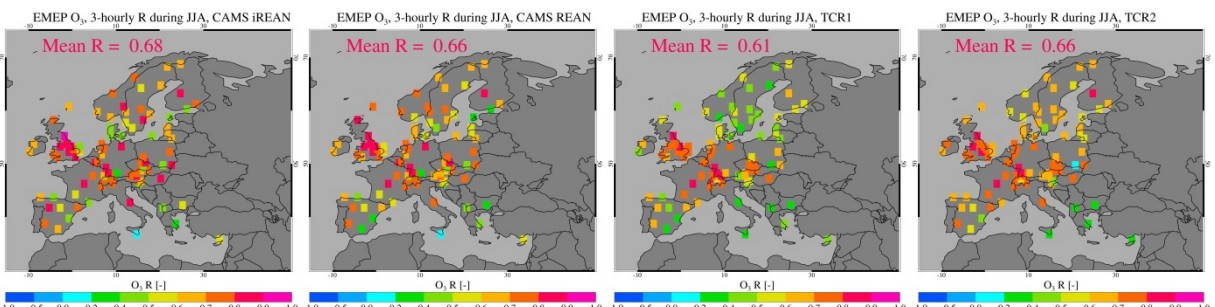

**Figure 10: As figure 9, but for JJA.**


A closer look at the diurnal cycle for different seasons and regions over Europe is given in Figure 11. In this figure the model biases have been subtracted in order to assess the model ability to capture the diurnal cycle only. All reanalyses generally capture the diurnal variability, and its variation across latitude region and season. For instance, all reanalyses show little diurnal variability for Northern European stations during DJF, although the CAMS-based reanalyses (and particularly

CAMS-REAN) show enhanced night-time $O_3$, which is not in TCR nor in the observations. Except for isoprene, no diurnal cycle in $O_3$ precursor emissions has been adopted in the CAMS reanalyses, which contributes to biases in the diurnal cycle. Note, however, that CAMS-REAN shows a comparatively large mean bias for these conditions, of -8 ppb (CAMS-iREAN bias is -6 ppb).



The diurnal cycle is generally larger for CAMS-iREAN than CAMS-REAN, overall showing better correspondence to the
observations. Particularly over middle and southern Europe during DJF the CAMS reanalyses show a larger diurnal cycle
than those obtained with TCR, also better matching to the observations. For MAM differences between the reanalyses are
rather small, while during JJA the TCR-2 and CAMS-iREAN show largest diurnal cycle across Europe, best matching again
to the observations.


**Figure 11: Plots of seasonal mean diurnal cycle against EMEP observations for 2006. Middle-Europe is here defined as the region between 35°N and 45°N, with Northern (Southern) Europe at higher (lower) latitudes. Note that the model bias has been removed. Model level selection is through matching of the model pressure with pressure level of the station sites.**



## 7. Time series of tropospheric ozone columns

Given the detailed validation results of tropospheric ozone profiles (Section 4), this section aims to demonstrate the potential value of the reanalysis products for studies of temporal changes in tropospheric ozone columns associated with changes in chemical and meteorological conditions for different regions of the world. Compared to global tropospheric column products derived directly from observations (e.g. Ziemke et al. 2019), reanalysis products have the potential to better include variations in near-surface ozone, provided that precursor emissions, deposition and chemical conversion are well constrained in the reanalysis.

Figure 11 shows time series of the annual mean partial ozone columns from the surface up to 300 hPa, for five zonal bands. From this, CAMS–iREAN shows an offset until mid 2013, followed by closer correspondence to the other reanalyses at every zonal band, in particular to the CAMS Reanalysis. The anomalously low columns in CAMS-iREAN before 2013 is due to a switch in the use of MLS data from V2 to NRT V3.4 (Flemming et al., 2017) together with the switch in the version of SBUV in 2013. While these switches implied the introduction of a positive offset in the CAMS-iREAN $O_3$ *total* columns with respect to CAMS-REAN and observations (Inness et al., 2019), the increased tropospheric columns in fact show overall a better correspondence to CAMS-REAN from this date onwards. The better consistency between CAMS-iREAN and CAMS-REAN could also be seen from the evaluation against sondes, Figures 3 and 4. Note however that from the sonde evaluations there is no overall indication that the CAMS reanalyses perform worse for the period from 2013 onwards, it can rather be characterized as a change in its error statistics.

CAMS-REAN and TCR-2 agree well over the NH extra-tropical regions, but show significant discrepancies over the tropics, with TCR-2 being 0.7 DU (2005) up to 1.8 DU (2016) larger than CAMS-REAN. Considering that tropospheric columns are already overall higher in CAMS-REAN than those derived from in-situ observations (Table 8), this suggests an overall slight over-estimate in TCR-2, particularly in the later period. Whereas CAMS-REAN is close to TCR-2 until 2009, it is closely correlated to the lower tropospheric columns in CAMS-iREAN from 2013 onwards. Nevertheless, the magnitude of the seasonal cycle is consistent for all reanalyses.

Over the SH extra-tropics, CAMS-REAN shows a good consistency with TCR-2, even though for the period before 2013 the amplitude in the seasonal cycle is a little larger. Both at mid-latitudes and high-latitudes there is a remarkable change in behaviour after 2013 in all reanalyses except TCR-1. From 2013 onwards the seasonal cycle is much weaker at mid-latitudes while essentially absent over the Antarctic. This change is largest for both CAMS reanalyses, but also visible in TCR-2, particularly over the Antarctic. Also the evaluations of tropospheric columns against sonde observations show changes in error statistics from 2013 onwards, see also Sec. 4. This shows once again the significant impact of changes in the observing system used to constrain tropospheric ozone, which may have difficulties to cope with the comparatively low magnitudes of tropospheric ozone columns over the Antarctic (~15 DU) compared to the Arctic (~ 26 DU).





680

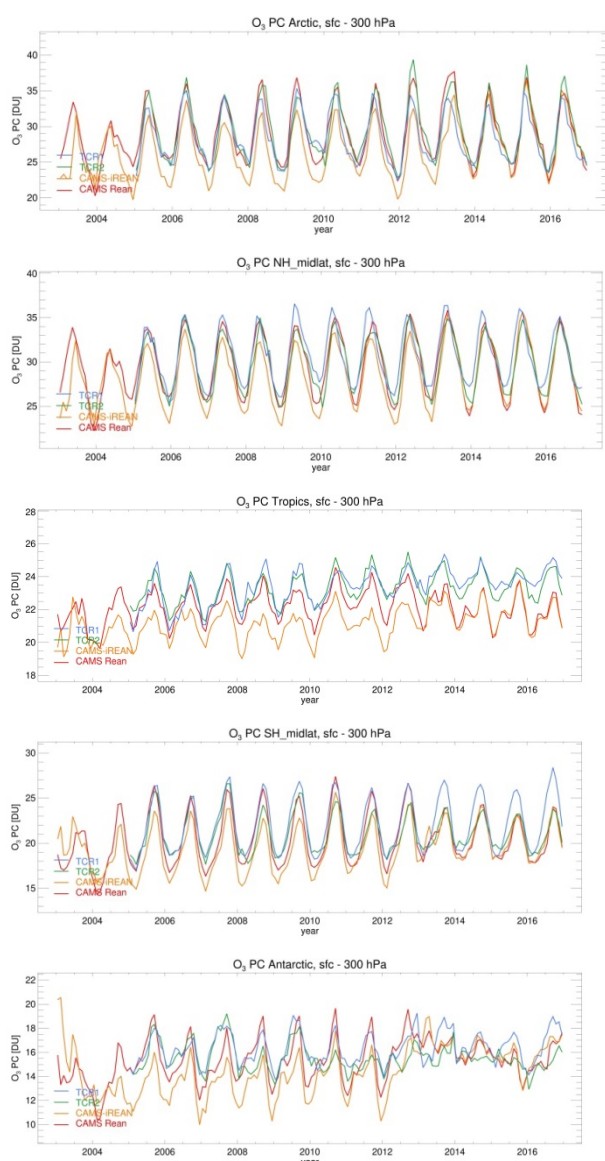

**Figure 12: Intercomparison of regionally averaged monthly mean partial columns up to 300 hPa.**





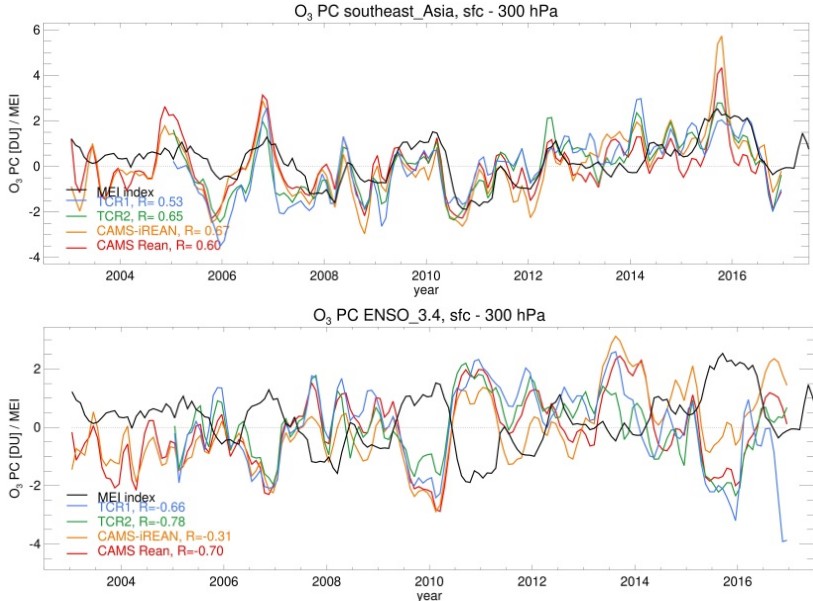

**Figure 13: Anomalies in O$_3$ partial columns (surface to 300 hPa) in four reanalyses, as compared to the MEI index for two regions: Southeast Asia (90° E - 120° E; 10° S - 20° N) and ENSO3.4 over the Eastern Pacific (120W-170W; 5S-5N). A 2-month smoothing has been applied to the reanalysis data, (as for the MEI index and TSI). Temporal correlations are given in the legends for comparison to MEI. Correlations are calculated on monthly data for the 2005-2016 time period.**

Figure 13 shows the anomalies in monthly mean ozone tropospheric columns (surface to 300 hPa) over two regions of the tropics. These anomalies are computed by subtracting the reanalysis-specific mean seasonal cycle based on the 2005-2016 time series. When comparing the anomalies with the Multivariate ENSO Index (MEI), (Wolter and Timlin, 1998), we find a significant correlation with R ranging between 0.6 (CAMS-REAN) and 0.65 (TCR-2) for the Southeast Asia region. A strong anti-correlation for the eastern Pacific region is found with R between -0.70 (CAMS-REAN) and -0.78 (TCR-2). The CAMS iREAN shows a lower correlation for this region, possibly associated with the jump in offset around the beginning of 2013, whose magnitude is significant in comparison to the signal. The high correlation over the Southeast Asia region is associated with enhanced fire emissions, and associated ozone production, during El Nino conditions over Indonesia (Inness et al., 2015), together with suppressed convection, while the anti-correlation over the Eastern Pacific is related to enhanced convection (Ziemke and Chandra, 2003).

Figures with anomalies in the monthly mean tropospheric ozone columns, together with their standard deviations are provided in Figure S4 in the Supplementary material. Table 10 presents an evaluation of the correspondence in this IAV between the four reanalyses. This is quantified as the correlation in the tropospheric ozone column anomalies for the four reanalyses. For southeast Asia CAMS-REAN is highly correlated to CAMS-iREAN (R=0.93), and likewise TCR-1 with TCR-2 (R=0.90). Lower, but still clear correlations are obtained particularly between the CAMS reanalyses and TCR-2 (R>0.82). Likewise for the ENSO_3.4 region CAMS-REAN is well correlated to CAMS-iREAN and TCR-2, but poorer





correlation is found between CAMS-iREAN and the TCR-reanalyses (R<0.57). Over the entire tropical region, correlations between the various reanalyses are relatively low (e.g. R between CAMS-REAN and TCR-2 is 0.29), caused by the small
signal, suggesting little robust information. Still, correlation between TCR-1 and TCR-2 is remarkably larger (R=0.73) than between CAMS-REAN and CAMS-iREAN (R=0.17). Generally smaller standard deviations in the monthly anomalies are found in the updated reanalyses.

We focus here on correlations between the ozone anomalies from the updated reanalyses (CAMS-REAN and TCR-2). Over the Arctic, and also the Eastern US, these are R=0.60 and R=0.63, respectively, giving some confidence in the robustness of
this IAV signal. Over Eastern Asia and Europe, these correlations decrease to 0.52 and 0.42. Over the Antarctic little correlation is remaining (R=0.33), implying that indeed any IAV from the reanalyses should be considered with care. Different reanalyses do not provide a consistent signal. Occasionally (e.g. over Antarctic) better correlations between reanalyses of the same family is found, but for instance over Europe and the Arctic the correlation between CAMS-REAN and TCR-2 is still better.

**Table 10: correlation coefficient R of the interannual variability in tropospheric O$_3$ columns between the four reanalyses, as computed for the 2005-2016 monthly mean time series in tropospheric O$_3$ columns from the surface to 300 hPa for different regions.**

| **Southeast Asia** | CAMS-REAN | CAMS-iREAN | TCR-2 | TCR-1 |
|---|---|---|---|---|
| CAMS-REAN | 1.00 | 0.93 | 0.82 | 0.73 |
| CAMS-iREAN | | 1.00 | 0.83 | 0.76 |
| TCR-2 | | | 1.00 | 0.90 |
| TCR-1 | | | | 1.00 |
| **ENSO_3.4** | CAMS-REAN | CAMS-iREAN | TCR-2 | TCR-1 |
| CAMS-REAN | 1.00 | 0.83 | 0.78 | 0.73 |
| CAMS-iREAN | | 1.00 | 0.57 | 0.48 |
| TCR-2 | | | 1.00 | 0.77 |
| TCR-1 | | | | 1.00 |
| **Arctic** | CAMS-REAN | CAMS-iREAN | TCR-2 | TCR-1 |
| CAMS-REAN | 1.00 | 0.50 | 0.60 | 0.35 |
| CAMS-iREAN | | 1.00 | 0.48 | 0.26 |
| TCR-2 | | | 1.00 | 0.46 |
| TCR-1 | | | | 1.00 |
| **Europe** | CAMS-REAN | CAMS-iREAN | TCR-2 | TCR-1 |
| CAMS-REAN | 1.00 | 0.33 | 0.42 | 0.07 |
| CAMS-iREAN | | 1.00 | 0.47 | 0.24 |



| TCR-2 | | | 1.00 | 0.40 |
|---|---|---|---|---|
| TCR-1 | | | | 1.00 |
| **EasternUS** | CAMS-REAN | CAMS-iREAN | TCR-2 | TCR-1 |
| CAMS-REAN | 1.00 | 0.41 | 0.63 | 0.45 |
| CAMS-iREAN | | 1.00 | 0.54 | 0.46 |
| TCR-2 | | | 1.00 | 0.64 |
| TCR-1 | | | | 1.00 |
| **EasternAsia** | CAMS-REAN | CAMS-iREAN | TCR-2 | TCR-1 |
| CAMS-REAN | 1.00 | 0.52 | 0.52 | 0.29 |
| CAMS-iREAN | | 1.00 | 0.69 | 0.55 |
| TCR-2 | | | 1.00 | 0.59 |
| TCR-1 | | | | 1.00 |
| **Tropics** | CAMS-REAN | CAMS-iREAN | TCR-2 | TCR-1 |
| CAMS-REAN | 1.00 | 0.17 | 0.29 | 0.01 |
| CAMS-iREAN | | 1.00 | 0.55 | 0.45 |
| TCR-2 | | | 1.00 | 0.73 |
| TCR-1 | | | | 1.00 |
| **Antarctic** | CAMS-REAN | CAMS-iREAN | TCR-2 | TCR-1 |
| CAMS-REAN | 1.00 | 0.65 | 0.33 | 0.39 |
| CAMS-iREAN | | 1.00 | 0.16 | 0.46 |
| TCR-2 | | | 1.00 | 0.56 |
| TCR-1 | | | | 1.00 |

## 8. Global spatial consistency between reanalyses

Figure 14 shows the multi-annual mean together with an evaluation of its multi-model standard deviation, at different

altitude levels. The standard deviation is computed from the multi-annual means of the four reanalyses, and provides a quantification of general agreement between reanalyses. The standard deviation at 850 and 650 hPa is relatively large over South America, Central Africa and Northern Australia, with values exceeding 6 ppb in the lower and middle troposphere. These results suggest that the representation of biomass burning emissions and its impacts on ozone production are largely different among the systems. Also large uncertainties in biogenic emissions likely contribute. In TCR, the optimization of

$NO_x$ emissions can have strong impacts on the lower and middle tropospheric ozone, in contrast to the CAMS configuration which applies prescribed anthropogenic and biogenic emissions, combined with the daily varying biomass burning





emissions. In addition, different representations of convective transport over the continents can lead to diversity in the vertical profile of ozone among the systems.

At 350 hPa, the multi-model standard deviation is large over Central Africa, South America and over the Arctic and
Antarctic, which could reflect different representations of deep convection along with biomass burning emissions at low latitudes, and polar vortex, stratospheric ozone intrusions and chemistry treatment at high latitudes among the systems.

The absolute differences between the two most recent reanalyses, TCR-2 and CAMS-REAN, are also shown. Apart from the regions mentioned above, differences are significant around Alaska and Siberia, regions with tropospheric ozone influenced by biomass burning events and where observational constraints at such high latitudes are more limited. Such larger
discrepancies once again highlight the importance of the forecast model performance in the reanalysis system as discussed in Miyazaki et al. (2019b), especially when direct observational constraints on tropospheric ozone are insufficient.

Frequency distributions of the annual mean ozone concentrations in the four reanalyses at three altitude levels are given in Figure 15 and summarize the general differences discussed above. In the lower and mid-troposphere the CAMS reanalyses show a larger frequency of $O_3$ values below 30 ppb (850 hPa) and 45 ppb (650 hPa) compared to particularly TCR-1, but
also TCR-2. This is associated to lower ozone in the CAMS reanalyses over the tropical regions. At 350 hPa the CAMS reanalyses and TCR-2 agree to a large extent in their frequency distribution. Only TCR-1 shows overall a larger occurrence of $O_3$ values in the range 70-100 ppb compared to the other reanalyses at the expense of primarily lower $O_3$ values. This is associated to positive model bias in this altitude range (see also Table 7).


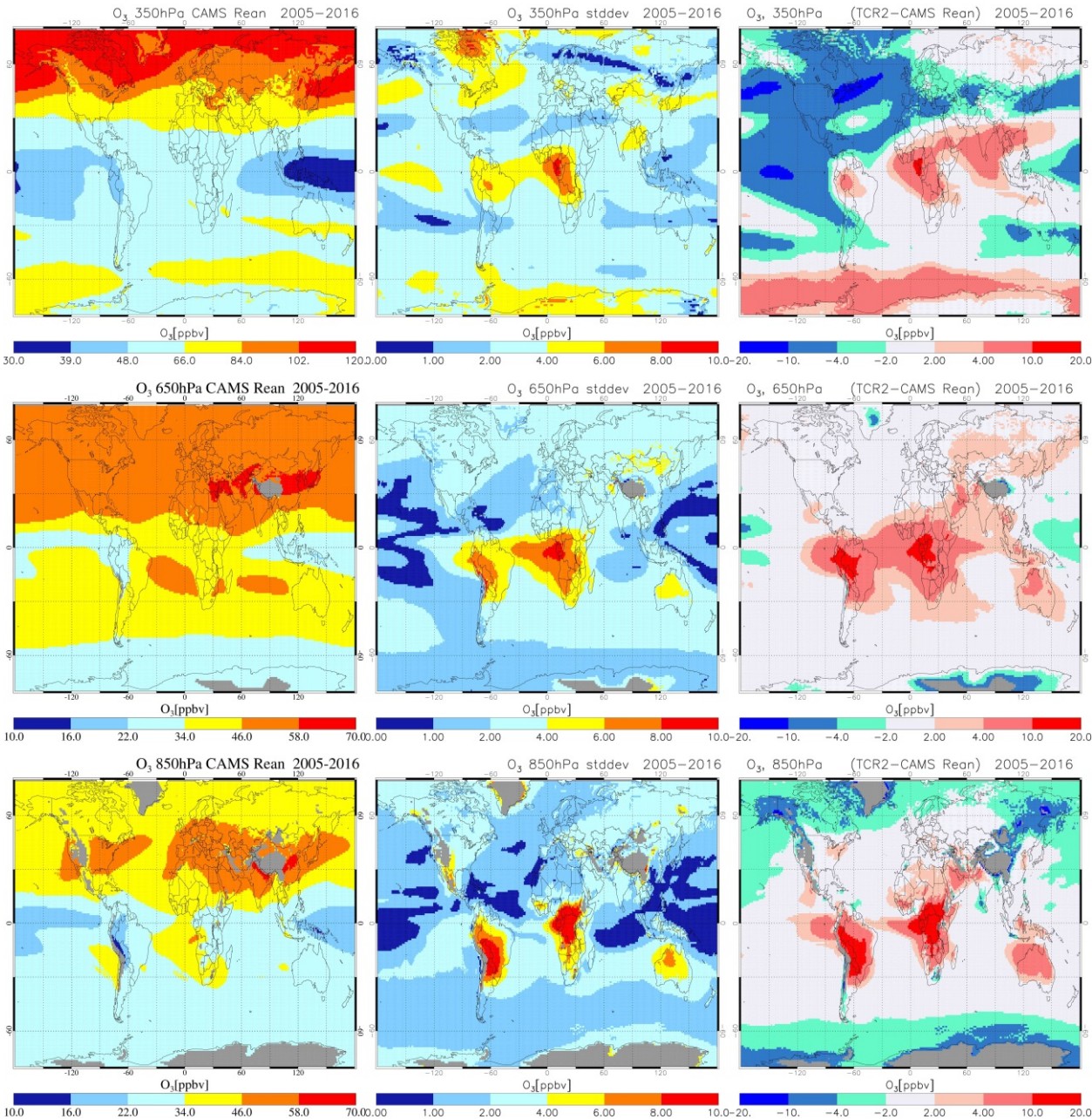

**Figure 14: Left: multi-annual mean O$_3$ at 350, 650 and 850 hPa for the CAMS Reanalysis at 650 hPa over 2005-2016. Middle:**
**755** **standard deviation in the multiannual means for the four reanalyses. Right: absolute difference between TCR-2 and CAMS**
**Reanalysis, all in units ppb.**





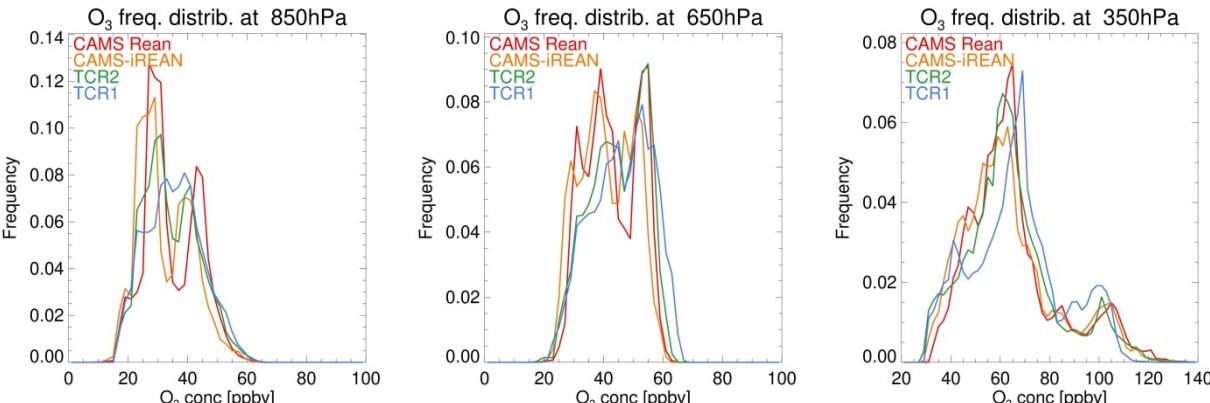

**Figure 15: Area-normalized frequency distributions of multi-annual mean O₃ mixing ratios at 350 (left), 650 (middle) and 850 hPa (right) for the four reanalyses.**


## 9. Conclusions and discussion

Four tropospheric ozone reanalyses have been compared in this paper, namely CAMS-iRean, CAMS-Rean, TCR-1, and TCR-2. A range of independent observations was used to validate the quality of the chemical reanalyses at various spatial

and temporal scales. These reanalyses aim to capture individual large-scale events, such as heat waves or wildfires, and at the same time aim to provide a globally consistent climatology of present-day composition. This implies stringent requirements on their temporal consistency. The changing constellation of satellite observations, their often limited sensitivity to tropospheric profiles and in particular the boundary layer, imply a significant dependency on the global chemistry model, its transport scheme, and its emissions, and makes the generation of any long-term chemical reanalysis

challenging. This calls for a detailed evaluation of the capability of the current reanalyses of tropospheric ozone.

Consistent with Inness et al., (2019), our evaluation also shows substantial improvement of CAMS-Rean over CAMS-iRean in the free troposphere, as quantified by lower RMS errors and higher correlations to ozone sonde observations, and better temporal consistency in multi-annual time series of tropospheric ozone columns. At the surface the CAMS-REAN is also generally better than CAMS-iREAN, assessed through evaluations of monthly mean surface concentrations against TOAR

observations, although similar performance of both CAMS reanalyses was seen for hourly to sub-seasonal variability assessed with EMEP observations over Europe for the year 2006, and in a few regions CAMS-iREAN showed a better diurnal cycle representation. The improved performance in the free troposphere can be attributed to a mixture of various upgrades, including revisions in the chemical data assimilation configuration, the chemistry mechanism, meteorological driver, model resolution, biogenic emissions.


Significant temporal changes in the quality of the ozone reanalysis in CAMS-iREAN across 2013 have been attributed to changes in the observing system, particularly a switch of MLS version 2 to version 3.4  In the CAMS system the MLS ozone profile measurements play a crucial role in constraining the partial column of ozone in the stratosphere. But as ozone total column observations are assimilated too, any changes in the MLS observations also affect the tropospheric ozone column in the CAMS reanalyses. In both CAMS reanalyses a change to the vertical resolution of the assimilated SBUV/2 data during

2013 had a negative impact on tropospheric ozone, particularly in polar regions.  Inness et al. (2019) had noticed such a change in performance, but had not yet identified the responsible observational dataset.

Compared with TCR-1, TCR-2 shows better agreements with independent observations throughout the troposphere, including at the surface. The improvements can be attributed to the use of more recent satellite retrievals and to an improved model performance, mainly associated with the increased model resolution. In spite of the good agreement with ozonesonde

measurements in the free troposphere, the surface ozone reanalysis exhibits large positive biases over Europe and the United States. Also, the lack of the TES measurements led to a degradation of the reanalysis performance after 2010 for many regions in the lower and middle troposphere, while none of total column measurements of ozone was assimilated in the TCR systems. In the TCR reanalysis, the chemical concentrations and precursor's emissions were simultaneously optimized through EnKF data assimilation, which was important in providing information on precursors' emissions variations

(Miyazaki et al., 2014; 2017; 2019a; Kiang et al., 2018) and in improving the vertical profiles of ozone.

Whereas free tropospheric ozone reanalyses agree well with independent observations, towards the surface this depends more on the model performance and emissions, and larger biases have been found in surface ozone analysis for many parts over the globe. A large spread at high latitudes could also be associated with the limited tropospheric ozone measurements. Recently developed retrievals with high sensitivity to the lower troposphere (e.g. Deeter et al., 2013; Fu et al., 2018; Cuesta

et al., 2018) would be helpful in improving the analysis of the lower troposphere. Meanwhile, the analysis ensemble spread from EnKF can be regarded as the uncertainty information about the analysis mean fields, indicating the need for additional observational constraints, whereas the 4-D Var system could be used to test the contributions from individual retrieval products.

We have demonstrated that the recent chemical reanalyses of CAMS-REAN and TCR-2 agree well with each other and with

the independent observations in the majority of cases. This highlights the usefulness of the current chemical reanalyses in a variety of studies. Meanwhile, our comparisons suggest that the model performance can still lead to discrepancies in the ozone reanalysis quality among the systems. For instance, differences in the representation of convective transport over the continents and those in the precursor's emissions, as well as differences in the chemical scheme, lead to substantial differences in the vertical profile of ozone and ozone production, as discussed in Miyazaki et al. (2019b). The relatively

coarse horizontal resolution of any of the global models could also cause significant model errors at urban sites. A coarse vertical resolution additionally has larger impacts on the quality of tropospheric ozone around the UTLS. Thus, although the reanalysis dataset provides comprehensive information about interannual variability in tropospheric ozone, both the data



assimilation settings and the model performance are critical in improving the tropospheric ozone analysis and obtaining consistent data assimilation analysis, especially for the lower troposphere.

Discontinuities in the availability and coverage of the assimilated measurements are also shown to affect the quality of the reanalysis, particularly in terms of temporal consistency, both in the CAMS and TCR-reanalyses. This is important for assessing interannual variability, and the usability of such reanalysis products for model evaluation. The influence of data discontinuities must be considered and where possible removed when studying interannual variability and trends using products from reanalyses. To improve the temporal consistency, a careful assessment of changes in the assimilation

configuration, most prominently associated with ozone column and profile assimilation is needed, including a detailed assessment of biases between various retrieval products.

The assimilation of multi-species data influences the representation of the entire chemical system, the influence of persistent model errors in complex tropospheric chemistry continues to be a concern. Also changes and biases in assimilation of precursor trace gases, such as $NO_2$, could influence temporal consistency in reanalyses of tropospheric ozone. Validation of

various trace gases from the chemical reanalysis products can be used to better identify potential sources of error in the reanalysis ozone fields. Furthermore, increasing the observational constraints together with the optimization of model parameters, such as the chemical mechanism, deposition, and mixing processes, could lead to more consistent data assimilation fields, hence further improving long-term reanalyses.

**Data availability**

The CAMS reanalyses data are freely available from https://atmosphere.copernicus.eu/ (last access: 18 October 2019). The TCR-1 reanalysis is available from https://ebcrpa.jamstec.go.jp/~miyazaki/tcr/ , the TCR-2 reanalysis is available from https://ebcrpa.jamstec.go.jp/tcr2/index.html .

**Author contributions**

VH and KM designed the study and wrote large parts of the manuscript. VH performed the evaluations and analyses. JF and
AI provided the CAMS-Reanalysis data, KM and TS provided the TCR-Reanalysis data. MGS provided the TOAR data, and contributed to its interpretation. All co-authors contributed to the writing and the analyses.

**Acknowledgements**

We acknowledge the use of data products from the NASA AURA, EOS Terra, and Aqua satellite missions. Part of the research was carried out at the Jet Propulsion Laboratory, California Institute of Technology, under a contract with the
National Aeronautics and Space Administration. We also acknowledge the free use of tropospheric $NO_2$ column data from





the SCIAMACHY, GOME-2, and OMI sensors from http://www.qa4ecv.eu and. This work was supported through JSPS KAKENHI grant number 18H01285 and the Environment Research and Technology Development Fund (2-1803) of the Ministry of the Environment, Japan, and by the Post-K computer project Priority Issue 4 - Advancement of meteorological and global environmental predictions utilizing observational Big Data. The Earth Simulator was used for simulations as
"Strategic Project with Special Support" of Japan Agency Marine-Earth Science and Technology. Part of this work was performed at the Jet Propulsion Laboratory, California Institute of Technology, under a contract with the National Aeronautics and Space Administration. The Copernicus Atmosphere Monitoring Service is operated by the European Centre for Medium- Range Weather Forecasts on behalf of the European Commission as part of the Copernicus Programme (http://copernicus.eu, last access: 15 March 2019). We thank all research and agency teams who provided data to WOUDCC,
SHADOZ, TOAR, and EMEP.

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
