# Peer review of "An inter-comparison of tropospheric ozone reanalysis products from CAMS, CAMS-Interim, TCR-1 and TCR-2"

_Geoscientific Model Development, 2019_

## Referee Comment (RC1) · Anonymous Referee #1 · 11 Dec 2019

The manuscript '*An inter-comparison of tropospheric ozone reanalysis products from CAMS, CAMS-Interim, TCR-1 and TCR-2*' presents a description and extensive evaluation of tropospheric ozone from four recent global chemical reanalyses: CAMS-iRean, CAMS-Rean, TCR1 and TCR2. The study performs very detailed comparisons between the reanalyses and independent observations of surface, profile and column ozone and assesses the relative performance of the reanalyses. This includes some very nice analyses of specific aspects of the reanalyses such as the representation of the diurnal ozone cycle. I am really impressed by the amount of work that went into this study and I applaud the authors for their very thorough and well-organized analysis. This type of paper is not easy to write but it is very important for the scientific community, especially as the understanding of the importance of chemical reanalyses is growing.

The paper is well thought out, well-written and well-organized. It is certainly worth of prompt publication. I have only several minor suggestions for edits and some technical corrections.

**Minor general comments**

1. Tables and the discussion of reanalyses' performance: it would be really good to have the RMSE values shown in percent in addition to absolute values (ppbv, DU). Having absolute values alone makes it difficult to judge if the RMSE is large or not. I appreciate that sometimes, particularly when the mean ozone is low, large percent values may be misleading but that shouldn't be an issue if both, absolute and relative RMSE is shown. In my specific comments I point to some places where having percentages would be particularly useful but it would really be best to have them in all the tables.
2. It would be helpful to have a schematic figure, similar to Davis et al. 2017 Fig. 1, showing the ozone observations assimilated by each reanalysis, indicating whether a bias correction was applied or not, and, as an added benefit, showing time periods of the reanalyses.
3. The authors often use the word 'model' as synonymous with 'reanalysis', e.g. L273, L307, L316, L318, and in many other places. I suggest limiting the use of this term to the instances where you are really talking about a model (e.g. 'model levels' or 'chemistry model', etc.)
4. Section 5 contains detailed discussions of reanalyses comparisons against multiple data sets. It's easy to lose the big picture in all these details. It would be really helpful to include 2-3 sentence summary highlighting the key results at the end of each subsection as it is already done in Subsection 5.3.

**Specific comments and technical corrections**

L35 climate-change → climate change

L34-36. This sentence conflates two different things: (1) the importance of ozone forcing for climate and (2) a lack of impact of improved ozone representation on long-term weather forecasts. I suggest splitting it into two sentences.

L38. This deserves more references than just the two that are provided.

L120-121 'to evaluate their fitness for purpose for the various types of application described above'. This sounds a little awkward. Please, consider rephrasing.

L158. Was there any kind of bias correction applied to these ozone data, as in CAMS-REAN? Maybe I missed that information. As I stated in my general comments a figure summarizing all these data types and how they're used in each reanalysis would be useful. This information could also be added to tables 2, 3 and 4.

Table 2. For the profile data types it would be helpful to include the vertical ranges or at least the lowest levels assimilated.

L200. Why couldn't they be filtered out?

L231 & L247.Livesey et al., 2011 is not in the reference list. If this is the MLS version 4.2 data quality and description document then its latest version is from 2018 (https://mls.jpl.nasa.gov/data/v4-2_data_quality_document.pdf). Is the MLS data quality screening based on some earlier guidelines? Note that v4.2 didn't exist in 2011.

L245-251. TES should also be mentioned here for completeness.

Table 4. According to the table TCR-1 uses MLS v3.3. It's version 4.2 in the text (L230).

L273. Data has been collocated → data **have** been collocated

L299 'any of the reanalysis model resolutions is considered too coarse' please correct the grammar

L310. What's the frequency of EMEP data?

L325. I think 'multiannual' is one word. At least, please be consistent; 'multi-annual' it's hyphenated a few lines below.

Figure 2 appears to be repeated or at least I can't discern any difference between the top two and bottom two rows. In addition, please, explain in the caption what 'Season: AYR' means ('all year'?) or remove it from the legend.

L325-342. What about the large discrepancy between the sondes and all the reanalysis near the surface at NH subtropics and, to a lesser extent, the tropics?

L341-346. The comparison with ACCMIP would be easier to see if the biases shown here were given as percentages in addition to absolute values.

L370. Could you briefly justify the use of an 'ozonopause' rather than more commonly used lapse rate or dynamical definitions of the tropopause? In addition, because of the high vertical resolution of ozonesondes they're likely to attain 150-ppbv threshold at very different (and somewhat random) altitudes than the reanalyses. How does that impact these comparisons?

L373. SH midlatitudes also look messy, especially TCR reanalyses. The absolute RMSE may be less than at high latitudes but relative to the mean column it looks quite high. Here and elsewhere it might be helpful to provide percent values for the mean biases and RMSE.

L390. 'These figures'. It's **one** figure (multiple panels)!

Figure 4. The caption says that ozonesondes are shown in black but since what's shown is biases w.r.t. the sondes the latter are not really shown at all, are they? I suggest deleting that sentence. Also, please state that numbers of observations are shown as gray dashed lines, even if it's obvious from the previous figure. As a side note, I'm not against multi-panel figures but I don't think I've seen one with 21 panels before.

L401. Why is MIPAS relevant to the troposphere? Is that an indirect impact of assimilating total ozone with stratospheric ozone constrained by MIPAS data? The same question applies to line 451-453 (Antarctic ozone).

L430-433. Any idea what happens around 2010-2011 that causes this improvement over Japan?

L445-446. The CAMS reanalyses show some large departures before 2005, especially at 382 hPa. Can you comment on that?

L507. But it's not exactly the same period, is it? Figure 6 shows aggregated data from 2005 to 2012 and S3 is extended through 2014.

Figure 8. The caption says 'left' and 'right'. It should be 'top' and 'bottom'. Alternatively, the panels could be labelled.

L652. 'Figure 11'. I think it should be '12'.

L727. Here and elsewhere, please provide percentages in addition to absolute values. How large (small) is 6 ppbv in this case?

L806. Could you expand on this? It would be very helpful to include a paragraph with specific recommendations for the users: What kind of studies are these reanalyses good for? Which reanalyses are recommended for a particular type of study and which ones are less reliable? Are there any types of problems for which these reanalyses are not useful?

This is partially addressed in the second to last paragraph where the authors delineate some issues related to trend and long-term variability studies using reanalyses but I think this type of discussion could be expanded to other areas.

L 810. Do you really mean 'any' models or is it 'many' models?

**References**

Davis, S. M., Hegglin, M. I., Fujiwara, M., Dragani, R., Harada, Y., Kobayashi, C. et al. (2017). Assessment of upper tropospheric and stratospheric water vapor and ozone in reanalyses as part of S-RIP,. *Atmos. Chem. Phys.*, 17, 12743-12778, https://doi.org/10.5194/acp-17-12743-2017

---

## Referee Comment (RC2) · Anonymous Referee #2 · 23 Dec 2019

This paper intercompare four tropospheric ozone reanalyses against independent observations. Each reanalysis and the independent observations are relatively well described. The intercomparison is done between 2003 and 2017 over a large number of diagnostics covering different situation of tropospheric ozone chemistry. There are nevertheless many shortcomings in this manuscript. First, the four reanalyses are not independent (two – CAMS-iREAN and TCR-1 – are the ancestor of the two letters – CAMS-REAN and TCR-2) which is confusing. Moreover, TCR-1 seems to have changed since its published paper (Miyazaki et al., 2015) which is even more confusing. There is a lot of discussion on the impact of change in the observing system during the reanalyses but these are not clearly shown. Finally, the overall presentation is poor

– figures and text – which make the paper difficult to be recommended for publications after minor revision. Here below are my detailed comments on the paper where I provide direction for improving the manuscript.

**Major comments.**

There are several aspects of the study that should be revised before the paper be accepted in GMD which are listed below:

1. The paper uses four reanalyses which are by far not independent. CAMS-REAN has been built above CAMS-iREAN in order to solve some of its shortcomings. This is the same for TCR-2 vs TCR-1. For me, the authors need to refocus the study by comparing only CAMS-REAN and TCR-2. If they want to compare CAMS-REAN and CAMS-iREAN, this should be done in a separate section. For TCRs, such a section is necessary since no publication have done a dedicated comparison as it is the case for CAMS in Inness et al. (2019).

2. There is a large confusion between TRC-1 (Miyazaki et al., 2015, available here https://ebcrpa.jamstec.go.jp/ miyazaki/tcr) and the version used in this paper. First, two different names should be used for these two different products. TCR-1 being already used, I suggest TCR-M (for MIROC) or anything that would clarify the confusion. But TCR-M seems closer to TCR-2 than TCR-1, except for the model spatial resolution. Moreover, on the TCR-1 webpage, it seems that surface NOx has been updated from Miyazaki et al. (2015) so it is difficult to know what is really TRC-M. In the revised paper, and in the section comparing TCRs reanalyses as suggested above, the authors should compare TCR-2 and TCR-1, not TCR-2 and TCR-M.

3. The paper lack a dedicated section on the changes in the observing systems and its impact on the reanalyses which is largely commented throughout the paper. How does the time series of the Obervation-minus-Forecast statistics affected by

these changes? Or the $\chi^2$-test, or the spread of the ensemble for EnKF systems, or the size of the analysis increments, or the number of relevant observations, or the comparison with a control run... This is essential for the users to know what they could expect – and what they can't – from these products. Regarding the use of the assimilated observations, the paper discuss ozone reanalyses in the polar region where TCRs are poorly constrained (no TES observations poleward $72°$). What is not said in the paper is that CAMS reanalyses are probably not well constrained as well in the winter poles since the assimilated ozone column are from UV sensors which are blind during the polar night. In the revised manuscript, I suggest removing all the discussion related to the polar regions (thus removing these regions also from the figures).

4. The figures need to be improved. The resolution of all the figures are too small. Many readers, like me, will try to zoom into them in the PDF document, which is not possible with their current resolution. Please, increase them. For the line plots, add a grid in the background of the figure. In general, the fonts are too small, they must be increased, as well as the line width. The legends are not always complete, please, describe everything shown in the figure. E.g. in Fig. 4, what is the dashed line referring to the left y-axis (which I cannot read due to the small size of the fonts)? You must also write what is shown when biases are plotted: obs-reanalyses or reanalyses-obs. If normalized differences, what is the norm? In Fig. 5, the colour levels in the bias are not very well chosen because it appears that all of the reanalyses seems to be highly biased. Why not using a constant colorbar with large steps showing only relevant differences? To extract major signal from the time series, I am suggesting plotting moving average allowing to detect the major differences between the observations and the reanalyses. Also, their readability will be improved by plotting the values of CAMS-REAN and TCR-2 only.

5. Many aspects of the conclusions and in the abstract are not shown in the paper,

e.g. the impact of the change in the observing system or the differences between the forecast models. On the other hand, the performance of the reanalyses in different tropospheric layer, conditions and seasons – which what this paper discusses – is almost ignored. In the conclusions it should make clear of what are the findings of this paper and what are subjects for future research.

6. The writing lack of clarity. For example, I do not understand the first sentence of the introduction. A careful reread of the paper is necessary to improve its readability. See some example in the specific comments.

**Other general comments**

1. Tables 5-9 provides a summary of the performances of each reanalysis compared to independent observations. This information is important and the values in the tables are mentioned throughout the paper. I have two major concerns with these tables. First, extracting the comparison between the reanalyses is difficult and I suggest replacing the tables by bar-plots. Second, I suggest replacing the RMS with the standard deviation of the difference. The RMS combines a measure of the bias and the variability of the difference. Since the bias is already provided, the standard deviation will tell us by how much the differences are distributed around the bias. For these figures, TCR-1 and CAMS-iREAN could be compared with their updates versions.

2. Also regarding differences, how are them calculated: obs-rean or the opposite? When normalized, what is the norm?

3. In Figure 3, the authors define the tropopause in each product as the altitude where ozone exceeds 150 ppbv which means that the altitude of the tropopause change from a product to the other. I suggest taking a surface pressure as defining the upper level of the free troposphere, e.g. 200 or 300 hPa. By using 300

hPa, they will be able to remove Fig. 12, which I suggest. Also, why showing the number of stations and not the number of soundings?

4. Regarding the use of the observations and in addition to my major comment above, the Tables 2 and 3 need to be revised.

   (a) As far as I know, there is only one CCI product for SCIAMACHY/GOME-2 TC and MIPAS profiles. I thus recommend to remove "(BIRA)" and "(KIT)".

   (b) What version of SCIAMACHY CCI is used? Same for MIPAS CCI, and GOME profiles? (I understand that NRT products have version changing during the time but this should not be the case for scientific – or offline – products.)

   (c) Also, does CAMS-iREAN and CAMS-REAN both assimilated MIPAS ESA NRT and CCI profiles? Which seems to use twice the profiles of the same instruments? Please, clarify. I am also surprised to see that CAMS use MIPAS NRT, a product older than 15 years and which was reprocessed by ESA several times (the ESA offline v7 is now the latest validated version).

   (d) You also mention MLS V3.4 which does not exist (at least for the offline products) – this is it either V3.3 or V4.2 (or shortly V3 or V4).

   (e) I would also add the reference to each dataset in an additional column.

   (f) The MLS version used in TCR-1 and TCR-2 are not clear. Version 4 is mentioned in the text while Table 4 mention version 3. Please clarify. Also use the appropriate MLS data quality document when referencing a version.

5. The terminology of "error statistics" is misused in the paper. It is generally applied to the error statistics in the DA system (i.e. B and R matrix and model error if any). In the case of this study, it is applied to the differences between the reanalyses and the observations so I would use the "observation-minus-analysis" statistics instead.

6. The authors use the inter-annual variability (IAV) and elsewhere deseasonalized anomaly, which seems to reflect to the same quantity. Could they clarify and use only one of those terminology?

7. I prefer the acronyms CIRA and CAMSRA, it is much easier when speaking than CAMS-iREAN and CAMS-REAN.

8. Many acronyms are undefined and should be.

**Specific comments**

L13-16: "Global tropospheric ozone reanalyses constructed using different state-of-the-art satellite data assimilation systems, prepared as part of the Copernicus Atmosphere Monitoring Service (CAMS-iRean and CAMS-Rean) as well as two fully independent Tropospheric Chemistry Reanalyses (TCR-1 and TCR-2), have been inter-compared and evaluated for the past decade." This is not true. CAMS-iREAN and TRC-1 are not constructed using state-of-the-art satellite data assimilation systems since these systems have been updated for CAMS-REAN and TCR-2.

L18-20: "The improved performance can be attributed to a mixture of various upgrades..." This is not shown in the paper.

L21-23: "Meanwhile, significant temporal changes in the reanalysis quality in all the systems can be attributed to discontinuities in the observing systems." Idem, this is not shown in the paper.

L22-24: "To improve the temporal consistency, a careful assessment of changes in the assimilation configuration, such as a detailed assessment of biases between various retrieval products, is needed." Which is what this paper should have been shown.

L24-26: "Even though the assimilation of multi-species data influences the representation of the trace gases in all the systems and also the precursors' emissions in the TCR reanalyses, the influence of persistent model errors remains a concern, especially for

the lower troposphere." Again, this is not shown in the paper.

L31-32: "The global distribution of present-day tropospheric ozone..." I don't understand this sentence, please, rephrase.

L41: "...tropospheric ozone, but are generally..." => "...tropospheric ozone, which is generally..."

L45: "Tropospheric ozone is reasonably well monitored..." You are talking about surface ozone in this sentence so I would write "Surface ozone is reasonably..."

L50-52: This list of satellite dataset is incomplete (missing are e.g. OMPS and TROPOMI for the most recent instruments) so I would write "These observations are complemented with (combined) satellite observations from, e.g., GOME-2, ..."

L62-64: "Simultaneously international modelling initiatives..." I don't understand this sentence, please, clarify.

L77: "...individual measurements suffer..." Do you mean "...individual measurements which suffer..."?

L81: "...particular constellations of pollution..." What do you mean by "constellations"?

L85: "However, all of these applications presume that the reanalysis is sufficiently accurate,..." What matter is that reanalysis is well characterized more than accurate.

L118-119: CAMS-REAN and CAMS-iREAN acronyms are undefined.

L126: "NOx" => "NOx"

L129: "...changing constellation of ..." => "... the change in the observing system..."

Table 1: What are the output frequency of each product. Are the output snapshots or time averages?

L156: => "The meteorological model version is CY40R2."

L157: "In terms of ozone, observations from the following set of satellite instruments have been assimilated:..."

L159: "Limb observations are instrumental to discriminate..." => "Profiles from limb instruments (MIPAS and MLS) are used to discriminate..." Could you explain how does limb profiles are used to discriminate the tropospheric and stratospheric contribution of the total column observations?

L161-163: See my general comment above regarding MLS V3.4.

L211: Remove reference to Watanabe et al. since it is already provided 2 lines above.

L341: "In the TCR systems,..." Move this info in Sect. 2.3.

Figure 2: What is the difference between the part in page 14 and 15? It seems to be the same.

L376: "...both model and observations..." Which model? Do you mean the reanalyses? If yes, replace by "... the reanalyses and the ozonesondes..."

L377: same as above "modelled" or "analysed"?

L379-381: Is the poor correlation between reanalyses and observations due to the missing total column observations during the polar night? Since, as far as I know, none of the total column assimilated data are taken by emissions instruments thus failing to measure during the night?

L397-399: "During 2003 and 2004 both CAMS reanalyses..." Why? This is not related to GOME data since CAMS-REAN does not assimilate GOME.

L399: "...GOME observations..." => "...GOME nadir profiles...".

L400-403: Why does CAMS assimilate MIPAS NRT and not the offline reprocessed products delivered by ESA?

L412-413: "Also both the observations and reanalyses indicate an upward trend of

tropospheric ozone in the UTLS..." I don't see this from figure 4. Could you clarify?

L431-433: "From 2011 onwards the correspondence with observations improves remarkably, despite the lack of TES measurements in TCR from June 2011 onwards." Why?

L434-444: I do not agree with most of what is written. • "In the tropics, ..." This is not true for CAMS-iREAN which generally underestimate the ozonesondes. • "... both CAMS reanalyses show a strong peak ..." In fact, TCRs also show a peak. • "...overestimation of up to 20 ppb." None biases are going up to -20 ppb. I would rather say -15 ppb. Do not omit the sign of the bias in the comparison. • "This spike appears much weaker in TCR..." Does the reason not due to the fact that TCR also optimize surface emissions allowing the reduce the bias with observations? • But the authors does not discuss the fact that CAMS-iREAN seems to have the best agreement with ozonesondes during the whole period and they should comment on the reason for this.

L449: "332 hPa" => "382 hPa"?

L467-474: I see other reasons for the seasonal variations in the bias time series than those mentioned in this §. For CAMS products, their troposphere is not constrained by any data during the polar night since all of the assimilated nadir instruments are measuring UV sun-scattered light. For TCR, TES ozone data are only available at latitude lower than +/-72°. Could the author comment on that?

L473: "332 hPa" => "351 hPa"

Sect. 5.2: "Figure 6 presents the temporal variability..." Well, figure 6 is a scatter plot without any time axis (on the x-, y- or any colorbar) so I would change this sentence. Moreover, all the discussion in this §related to seasonal differences are not supported by Fig. 6. I understand that Fig. S3 could support this discussion but as being part of the supplement, it cannot be used for new discussion.

L521: Here and at several other placed "R=0.89 – 0.96"? Do you mean "R between 0.89 and 0.96" or "R[0.89,0.96]"? Or something else?

L541: "We compute the interannual…" Do you mean the deseasonnalized anomaly for each region? See also the general comments.

L652: Do you mean Fig. 12? So this is almost Fig. 3 without observations. Is it really the annual mean? It seems more to be a time series of monthly mean?

L669-670: The change in behaviour is clear above the SH polar latitude but less clear in SH midlatitudes.

Figure 13: • Is it as Fig. 7 but for PC surface-300 hPa in south-east Asia and ENSO? • "A 2-month smoothing". Do you mean a running mean or moving average? • What is TSI?

L742: "…annual mean…" For which year?

Figure 15 is very interesting but I would add the ozone sonde values in order to assess the quality of the reanalyses against the best estimation of the truth (i.e. the sondes).

L767: "The changing constellation…" I would rather say "The changes in the observing system…"

L770: "This calls for a detailed evaluation of the capability of the current reanalyses of tropospheric ozone." Do you mean this is something to do in the future? Please, clarify.

L793-795: "In the TCR reanalysis, the chemical concentrations and precursor's emissions were simultaneously optimized through EnKF data assimilation, which was important in providing information on precursors' emissions variations (Miyazaki et al., 2014; 2017; 2019a; Kiang et al., 2018) and in improving the vertical profiles of ozone." Well, this is not shown in the paper so I would remove this comment from the conclusions.

L800-803: "Meanwhile, the analysis ensemble spread …" Well, again, the TCRs ensemble spread are not shown in the paper. Also, what do you mean with "4D-var could be used . . ." Altogether, I don't understand the message in this sentence.

L413: The acronym UTLS must be defined.

L819: " . . . a careful assessment of changes in the assimilation configuration. . ." Which what this paper should have done.

L822: "The assimilation of multi-species data influence. . ." This has not been addressed in the paper.

---

## Referee Comment (RC3) · Anonymous Referee #3 · 2 Jan 2020

This paper inter-compared tropospheric ozone reanalysis products from CAMS, CAMS-Interim, TCR-1, and TCR-2. This study is of scientific importance and the research is well conducted. The presentation is generally clear with logic flow and convincing discussions. The paper provides an enhanced understanding of issues related to trospehric ozone reanalysis products. I only have some minor issues for the authors to consider when revising their paper.

Minor issues:

1. In the abstract and conclusions, it is useful to summary where and when the reanalysis products perform strongest and weakest, in term of relative difference with ozonesonde data.
2. Figures and tables need more annotation.
   Fig. 1, What are the boxed areas?
   Table 5-7, the area definitions can be provided in Fig. 1.
   Fig. 4. Relative difference is more meaningful.
   Fig. 11, what is the time zone for this figure? How are the model errors removed?
3. The definition of the tropopause needs some discussion and references.
4. The paper appears lengthy. Please shorten the paper and move less significant contents to Supplements.

---

## Author Comment (AC1) · 7 Feb 2020

**Response to the first reviewer**

We thank the referee for his/her positive review and for the provision of useful comments and suggestions. Below we answer them to our best ability. The reviewer comments are in italic. Our responses are in regular font, and changes to the manuscript are given in bold.

*The manuscript 'An inter-comparison of tropospheric ozone reanalysis products from CAMS, CAMS-Interim, TCR-1 and TCR-2' presents a description and extensive evaluation of tropospheric ozone from four recent global chemical reanalyses: CAMS-iRean, CAMS-Rean, TCR1 and TCR2. The study performs very detailed comparisons between the reanalyses and independent observations of surface, profile and column ozone and assesses the relative performance of the reanalyses. This includes some very nice analyses of specific aspects of the reanalyses such as the representation of the diurnal ozone cycle. I am really impressed by the amount of work that went into this study and I applaud the authors for their very thorough and well-organized analysis. This type of paper is not easy to write but it is very important for the scientific community, especially as the understanding of the importance of chemical reanalyses is growing.*
*The paper is well thought out, well-written and well-organized. It is certainly worth of prompt publication. I have only several minor suggestions for edits and some technical corrections.*

*Minor general comments*

*1. Tables and the discussion of reanalyses' performance: it would be really good to have the RMSE values shown in percent in addition to absolute values (ppbv, DU). Having absolute values alone makes it difficult to judge if the RMSE is large or not. I appreciate that sometimes, particularly when the mean ozone is low, large percent values may be misleading but that shouldn't be an issue if both, absolute and relative RMSE is shown. In my specific comments I point to some places where having percentages would be particularly useful but it would really be best to have them in all the tables.*

We now add barplots with normalized mean bias and normalized standard deviation (instead of RMSE, as per a comment from the second reviewer) in the supplementary material.

*2. It would be helpful to have a schematic figure, similar to Davis et al. 2017 Fig. 1, showing the ozone observations assimilated by each reanalysis, indicating whether a bias correction was applied or not, and, as an added benefit, showing time periods of the reanalyses.*

We have checked the Fig 1. in Davis et al. (2017), but find it unpractical to introduce a similar figure for our purposes, as this implies considerable overlap with Tables 2-4. Also not only information on the satellite instrument is important, but also the version specification, which implies that the figure cannot replace the existing tables. We now introduce a separate section in the manuscript to discuss any changes in the observing systems.

*3. The authors often use the word 'model' as synonymous with 'reanalysis', e.g. L273, L307, L316, L318, and in many other places. I suggest limiting the use of this term to the instances where you are really talking about a model (e.g. 'model levels' or*

*'chemistry model', etc.)*

The reviewer is fully correct. We have checked the manuscript, and replaced 'model' with 'analysis' or similar, where appropriate.

*4. Section 5 contains detailed discussions of reanalyses comparisons against multiple data sets. It's easy to lose the big picture in all these details. It would be really helpful to include 2-3 sentence summary highlighting the key results at the end of each subsection as it is already done in Subsection 5.3.*

We now introduce a summary section at the end of Sec. 4.3, concluding the evaluation against ozone sondes. Likewise we now introduce summary statements at the end of Sec 5.2 an Sec 6:

End of sec. 4.3:
**In conclusion, evaluation against ozone sondes has revealed the following:**
**- The updated reanalyses show on average improved performance compared to the predecessor versions, but with some notable exceptions, such as an increased positive bias over the Antarctic in CAMS-Rean versus CAMS-iRean. Over the Antarctic the TCR-2 strongly improved upon TCR-1, despite the lack of direct observational constraints.**
**- For individual regions or conditions CAMS Reanalysis and TCR-2 show different performance, but averaged for all regions of similar quality. Best performance, in terms of mean bias, standard deviation and correlation, for the updated reanalyses is obtained for the Western Europe, Eastern US and SH mid latitude regions (both normalized mean bias and standard deviation below 8% at 850 and 650 hPa). Relatively worst performance is found for the Antarctic region, with normalized standard deviation up to 18%. This is likely associated to the fewer observational constraints in the polar regions compared to the other regions.**
**- In terms of temporal consistency, the CAMS Reanalyses show degraded performance over the polar regions during 2003 and 2004, due to lower quality MIPAS and SCIAMACHY data usage. CAMS-iREAN also shows a change in performance statistics in the polar regions from 2014 onwards, associated to a changes in the MLS retrieval product versions. Furthermore, both CAMS-Rean and CAMS-iRean are affected by the change in the SBUV/2 product versions in 2013.**
**With the reduced data-availability from TES from 2010 onwards the TCR tropospheric ozone products show changes in their performances. Remarkably, TCR-1 and TCR-2 show overall slight improvements from 2010 onwards. This is marked by reduced positive biases in the lower troposphere over NH-mid-latitude regions and may be attributed to biases in the TES retrieval product, combined with changes in the OMI product, see also Sec. 2.5. Additional Observing System Experiments (OSEs) are needed to identify the relative roles of individual assimilated measurements on the changes in reanalysis bias.**
**end of sec 5.2:**
**In summary, CAMS-Rean shows the best ability to capture the regional mean surface ozone and its variability, while particularly TCR-2 (and to lesser extent also TCR-1) shows positive biases and reduced correlations. Particularly good performance is seen over the western US (R=0.95, MB=-0.2), while over east, and particularly southeast, Asia the performance is poorest.**

end of sec.6 :
**In summary, all reanalyses capture the synoptic to diurnal variability, as illustrated by the assessment of the heatwave event in July 2006. Still there are considerable differences in**

**performance, depending on the reanalysis, region and season. While CAMS-iRean and CAMS-Rean perform mostly similar, for TCR-2 a considerable improvement was found compared to TCR-1. Overall better temporal correlations are obtained for the summer period compared to winter, and also for Western Europe compared to the Mediterranean region. Further improvements can be obtained by a better description of surface processes, including emissions and deposition, together with higher spatial resolution modelling.**

*Specific comments and technical corrections*

*L35 climate-change à climate change*

*Changed.*

*L34-36. This sentence conflates two different things: (1) the importance of ozone forcing for climate and (2) a lack of impact of improved ozone representation on long-term weather forecasts. I suggest splitting it into two sentences.*

Thanks for this suggestion. We now write:
**Owing to its radiative effects, tropospheric ozone is an important driver in climate change (Checa-Garcia et al., 2018). Also it may affect long-range weather forecasts, even if in evaluations no improvement has been detected so far (Cheung et al., 2014).**

*L38. This deserves more references than just the two that are provided.*

This is correct. We now have added references to Monks et al. (2015), Huang et al., (2017), and Hsu and Prather (2009).

*L120-121 'to evaluate their fitness for purpose for the various types of application described above'. This sounds a little awkward. Please, consider rephrasing.*

We now write:
**To assess the quality of these reanalysis products, with attention for the various potential types of application described above, this study evaluates tropospheric ozone…**

*L158. Was there any kind of bias correction applied to these ozone data, as in CAMS-REAN? Maybe I missed that information. As I stated in my general comments a figure summarizing all these data types and how they're used in each reanalysis would be useful. This information could also be added to tables 2, 3 and 4.*

We now explicitly mention the bias correction settings in any of the reanalyses. The settings for CAMS-iRean and CAMS-Rean are identical (variational bias correction for OMI, SCIAMACHY, GOME-2, and anchoring for SBUV/2, MLS and MIPAS), while in TCR-1 and TCR-2 all observations were used without bias correction.

*Table 2. For the profile data types it would be helpful to include the vertical ranges or at*

*least the lowest levels assimilated.*

Profile data from MIPAS and MLS instruments in the ranges 0.1 -150 hPa (MIPAS) and 0.1 - 147hPa (MLS) is used. For SBUV and GOME (ERS-2) the vertical resolution is very low, implying that they can effectively be considered as total column retrievals. We now include such a comment in the manuscript.

*L200. Why couldn't they be filtered out?*

These OMI row anomalies could not be filtered out because at the time this information was not available in the BUFR data which are used as input to the IFS data assimilation system. This had unfortunately not been noticed before running the reanalysis. This information will be taken into account in any future reanalysis. We now write:

**-Different behaviour of OMI data between 2009 and 2012, associated to a deterioration in the OMI row anomalies (Schenkeveld et al., 2017) which unfortunately have not be filtered out in the CAMS assimilation procedure;**

*L231 & L247.Livesey et al., 2011 is not in the reference list. If this is the MLS version 4.2 data quality and description document then its latest version is from 2018 (https://mls.jpl.nasa.gov/data/v4-2_data_quality_document.pdf). Is the MLS data quality screening based on some earlier guidelines? Note that v4.2 didn't exist in 2011.*

The reviewer is correct: Version 4.2 was actually used in both TCR-1 and TCR-2. The v3.3 MLS data was used in a predecessor of TCR-1, not assessed in our manuscript. MLS data quality screening is also based on the v4.2 guidelines. We have updated the manuscript, and reference on this.

*L245-251. TES should also be mentioned here for completeness.*

We now include such a sentence, thank you for this suggestion.

*Table 4. According to the table TCR-1 uses MLS v3.3. It's version 4.2 in the text (L230).*

The reviewer is correct: it should have been version 4.2, as was already mentioned in the text. This is now also updated in the table.

*L273. Data has been collocated à data **have** been collocated*

Updated, thank you.

*L299 'any of the reanalysis model resolutions is considered too coarse' please correct the grammar*

changed into:
**...because none of the reanalysis model resolutions is considered sufficient to resolve ...**

*L310. What's the frequency of EMEP data?*

EMEP provides hourly observations. For our evaluation we use a reference three-hourly time frequency. We now clarify these time frequency aspects specifically in Sec 3.3

*L325. I think 'multiannual' is one word. At least, please be consistent; 'multi-annual' it's hyphenated a few lines below.*

We now consistently write '**multiannual**'

*Figure 2 appears to be repeated or at least I can't discern any difference between the top two and bottom two rows. In addition, please, explain in the caption what 'Season: AYR' means ('all year'?) or remove it from the legend.*

The reviewer is correct about the duplication, we apologize for this. The reference to 'Season: AYR' (referring to full multiannual averaging as compared to seasonal averaging) is removed from the figures.

*L325-342. What about the large discrepancy between the sondes and all the reanalysis near the surface at NH subtropics and, to a lesser extent, the tropics?*

The near-surface discrepancy for the NH-subtropical region can mostly be attributed to positive biases in any of the reanalyses against the Hong Kong (114.2° E, 22.3° N) sonde observations, see also Figure R1 below. $O_3$ at the Hilo (155° W, 19.4° N) and Naha (127.7° E, 26.2° N) stations perform much better at these low altitudes. Likewise, for the tropical region a large bias could be attributed to the Kuala Lumpur, Malaysia station (101.3°E, 2.7°N). But due to the sparseness of the observations in these regions it remains difficult to derive general conclusions.

[Figure]

*Figure R1. Evaluation of multi-annual mean ozone from all reanalyses sampled at the Hong Kong (left) and Kuala Lumpur (right) stations.*

In the manuscript we now write:
**In the NH subtropics and the tropics regions the reanalyses show some larger deviation against sonde observations at lower altitudes, which was traced to comparatively large biases at the Hong Kong and Kuala Lumpur stations. Note that in these regions the ozonesonde network is sparse, while the spatial and temporal variability of ozone is large, which limits our understanding of the generalized reanalysis performance (Miyazaki and Bowman, 2017).**

*L341-346. The comparison with ACCMIP would be easier to see if the biases shown here*

*were given as percentages in addition to absolute values.*

We now also report on normalized biases (and standard deviation) in new figures in the Supplementary Material. We include a statement on the maximum normalized (absolute) mean bias being below 10%.

*L370. Could you briefly justify the use of an 'ozonopause' rather than more commonly used lapse rate or dynamical definitions of the tropopause? In addition, because of the high vertical resolution of ozonesondes they're likely to attain 150-ppbv threshold at very different (and somewhat random) altitudes than the reanalyses. How does that impact these comparisons?*

In line with the comment from reviewer #2 we now use a more clear definition of the tropospheric column. We now compute this as the partial column from the surface to 300 hPa. Indeed, this helps to intercompare the reanalyses, as alternatively the altitude of the tropopause level changes between reanalysis.

At the start of Sec 4.2 we now write:
**Collocated partial columns from the surface up to 300 hPa, hereafter for brevity referred to as 'tropospheric columns', have been compared to partial columns derived from the sonde observations.**
All figures and reporting on error statistics has been updated accordingly.
*L373. SH midlatitudes also look messy, especially TCR reanalyses. The absolute RMSE may be less than at high latitudes but relative to the mean column it looks quite high. Here and elsewhere it might be helpful to provide percent values for the mean biases and RMSE.*

Following the reviewer's recommendation we now also compute the normalized biases. They indeed indicate difficulties over the SH mid latitudes, although smaller compared to the high latitudes. We now write:

**Over the SH mid latitudes the reanalyses show similar features as over the Antarctic, with normalized mean biases within -1DU (-5%, CAMS-iREAN) and 1.5 DU (+10%, TCR-1). The normalized standard deviations over the SH mid latitudes are within 7%, marking a considerably better ability to capture temporal variability than over the Antarctic.**

*L390. 'These figures'. It's **one** figure (multiple panels)!*

Changed to '**the panels in this figure'**, thank you.

*Figure 4. The caption says that ozonesondes are shown in black but since what's shown is biases w.r.t. the sondes the latter are not really shown at all, are they? I suggest deleting that sentence. Also, please state that numbers of observations are shown as gray dashed lines, even if it's obvious from the previous figure. As a side note, I'm not against multipanel figures but I don't think I've seen one with 21 panels before.*

The reviewer is correct: the sentence is deleted now, and explanation of the gray dashed line is included instead.

*L401. Why is MIPAS relevant to the troposphere? Is that an indirect impact of assimilating total ozone with stratospheric ozone constrained by MIPAS data? The same question applies to line 451-453 (Antarctic ozone).*

The reviewer is correct. To explain this better, in Sec 2.1 the following sentence is included:

**Profile observations from limb instruments (MIPAS and MLS) are used to constrain the stratospheric contribution of the total column. In combination with the assimilated total column retrievals this implies that also the tropospheric part is constrained (Inness et al., 2013).**

And in sec. 4.3, when discussing the impact of MLS:

**Combined with total column retrievals, assimilation of such stratospheric profiles has been shown to also affect the tropospheric contribution (Inness et al., 2013).**

*L430-433. Any idea what happens around 2010-2011 that causes this improvement over Japan?*

It is very difficult to attribute the change in bias statistics over Japan around 2010 for the four reanalyses. Aspects that play a role are following:
- The ozone observations at 650 hPa show relatively large annual mean values, during 2010 and 2012, see Figure S1 in the (original) supplementary material. This may be associated to the increased NOx emissions from China in the preceding decade (e.g. Verstraeten et al., 2015), which show a maximum during 2011 – 2014 (van der A et al., 2017). Note that in the CAMS reanalyses $NO_x$ emissions are not optimized in the data-assimilation procedure, although $NO_2$ tropospheric columns have been assimilated. Instead an annual trend is assumed in the MACCity based emissions.
- The TCR-based reanalyses show a significant change in their characteristics after 2010 due to a reduction in TES retrievals, which stopped completely after June 2011.
- Both the TCR and CAMS reanalyses are affected by the row anomaly issue in the OMI $O_3$ (relevant to CAMS-REAN and CAMS-iREAN, particularly during 2009-2012, Inness et al., 2017) and $NO_2$ retrieval products (relevant to all reanalyses).

It is unfortunately beyond the scope of this work to assess the partial contributions of these effects. To provide more clarity, in the manuscript we now write:

**The changes in performance statistics for all reanalyses likely have multiple causes. This includes trends in the observed ozone (Verstraeten et al., 2015), associated to changes in Chinese precursor $NO_x$ emissions (e.g. van der A et al., 2017). Also changes in the observing system are important to consider, particularly the reduction of assimilated TES measurements in TCR from 2010 onwards, and the row anomaly issues affecting assimilated OMI $O_3$ and $NO_2$, see also Sec. 2.5.**

*L445-446. The CAMS reanalyses show some large departures before 2005, especially at 382 hPa. Can you comment on that?*

We attribute this to similar causes as identified for the NH polar region, namely the use of early SCIAMACHY and NRT MIPAS O3 retrievals, which are of poorer quality than the OMI MLS observations which have been used from August 2004 onwards, and reprocessed MIPAS data used from January 2005 onwards. We now write accordingly:

**Furthermore, CAMS-iRean and CAMS-Rean suffer from relatively large negative biases before 2005, particularly at 382 hPa. This is attributed to similar causes as have been discussed for the Arctic region.**

*L507. But it's not exactly the same period, is it? Figure 6 shows aggregated data from 2005 to 2012 and S3 is extended through 2014.*

That is correct, therefore we provide the exact time range in all the table and figure legends. We now also specify this additionally in this particular sentence.

*Figure 8. The caption says 'left' and 'right'. It should be 'top' and 'bottom'. Alternatively, the panels could be labelled.*

We now change to 'top'/'bottom', thank you.

*L652. 'Figure 11'. I think it should be '12'.*

The reviewer is correct, this is now changed, thank you.

*L727. Here and elsewhere, please provide percentages in addition to absolute values. How large (small) is 6 ppbv in this case?*

We now include such assessment here. We write:
**Normalized to local mean $O_3$ from the CAMS Reanalysis, the standard deviation values at 850 hPa reach 20% over Australia and up to 50% over South America and Central Africa. At 650 hPa these maximum ratios decrease to approx. 10% (Australia) and 20% (South America and Central Africa).**

*L806. Could you expand on this? It would be very helpful to include a paragraph with specific recommendations for the users: What kind of studies are these reanalyses good for? Which reanalyses are recommended for a particular type of study and which ones are less reliable? Are there any types of problems for which these reanalyses are not useful? This is partially addressed in the second to last paragraph where the authors delineate some issues related to trend and long-term variability studies using reanalyses but I think this type of discussion could be expanded to other areas.*

The reviewer is correct that such suggestions could be useful. We now include the following sentences:

**The well-characterized, small mean bias in tropospheric columns in these reanalyses suggest that they can be used to provide a climatology of present-day tropospheric ozone. This may serve as a reference for the present-day contribution of tropospheric ozone to the radiation budget, or may provide a climatology for a-priori ozone profiles as required for satellite retrieval products (e.g., Fu et al., 2018). The ability of the CAMS Reanalysis to capture the variability of (near-)surface ozone on multiple time scales, and for many regions over the globe, indicates it is fit for use as boundary conditions for hindcasts of regional air quality models.**

*L 810. Do you really mean 'any' models or is it 'many' models?*

We refer to the model configurations discussed in our evaluation. We now rewrite to:
**The relatively coarse horizontal resolution in any of the global reanalysis configurations could also cause significant errors at urban sites.**

*References*

*Davis, S. M., Hegglin, M. I., Fujiwara, M., Dragani, R., Harada, Y., Kobayashi, C. et al. (2017). Assessment of upper tropospheric and stratospheric water vapor and ozone in reanalyses as part of S-RIP,. Atmos. Chem. Phys., 17, 12743-12778, https://doi.org/10.5194/acp-17-12743-2017*

Verstraeten, W. W., Neu, J. L., Williams, J. E., Bowman, K. W., Worden, J. R., and Boersma, K. F.: Rapid increases in tropospheric ozone production and export from China, Nat. Geosci., 8, 690–695, doi:10.1038/ngeo2493, 2015.

van der A, R. J., Mijling, B., Ding, J., Koukouli, M. E., Liu, F., Li, Q., Mao, H., and Theys, N.: Cleaning up the air: effectiveness of air quality policy for SO2 and NOx emissions in China, Atmos. Chem. Phys., 17, 1775–1789, https://doi.org/10.5194/acp-17-1775-2017, 2017.

---

## Author Comment (AC2) · 7 Feb 2020

**Response to the second reviewer**

We thank the referee for his/her efforts to provide this critical review, which contain many useful comments and suggestions. Below we answer them to our best ability. This has substantially helped to improve the manuscript. The reviewer comments are in italic. Our responses are in regular font, and changes to the manuscript are given in bold.

*This paper intercompare four tropospheric ozone reanalyses against independent observations. Each reanalysis and the independent observations are relatively well described. The intercomparison is done between 2003 and 2017 over a large number of diagnostics covering different situation of tropospheric ozone chemistry. There are nevertheless many shortcomings in this manuscript. First, the four reanalyses are not independent (two – CAMS-iREAN and TCR-1 – are the ancestor of the two letters – CAMS-REAN and TCR-2) which is confusing. Moreover, TCR-1 seems to have changed since its published paper (Miyazaki et al., 2015) which is even more confusing. There is a lot of discussion on the impact of change in the observing system during the reanalyses but these are not clearly shown. Finally, the overall presentation is poor – figures and text – which make the paper difficult to be recommended for publications after minor revision. Here below are my detailed comments on the paper where I provide direction for improving the manuscript.*

We thank the reviewer for this summary of his/her main concerns. We address them below responding to the major comments. As consequence of this review, we have substantially revised the manuscript, which can hopefully be appreciated by the reviewer.

***Major comments.***
*There are several aspects of the study that should be revised before the paper be accepted in GMD which are listed below:*
*1. The paper uses four reanalyses which are by far not independent. CAMS-REAN has been built above CAMS-iREAN in order to solve some of its shortcomings. This is the same for TCR-2 vs TCR-1. For me, the authors need to refocus the study by comparing only CAMS-REAN and TCR-2. If they want to compare CAMS-REAN and CAMS-iREAN, this should be done in a separate section. For TCRs, such a section is necessary since no publication have done a dedicated comparison as it is the case for CAMS in Inness et al. (2019).*

We acknowledge that the four reanalyses are not equally independent, which is clearly reflected in the naming of the products. We also agree that the newer reanalyses can overall be considered as improvements with respect to the predecessor versions, as we also conclude in the manuscript.
The reviewer is correct that Inness et al. (2019) has presented some evaluations of tropospheric ozone, intercomparing the CAMS reanalysis with the CAMS Interim Reanalysis. Nevertheless, Inness et al. (2019) covers much more aspects of the composition reanalysis, at the expense of level of detail of the evaluation of tropospheric ozone. Therefore we believe that providing this evaluation is still useful.

Furthermore, we believe it is fully meaningful to compare the reanalysis performance between the different versions of chemical reanalyses produced using similar frameworks (TCR-1 vs TCR-2 and CAMS-iRean vs CAMS-Rean). This allows us to demonstrate the impact of updating

the data assimilation configurations on the performance of the reanalyses. It also provides information whether the recent reanalyses have got closer, in any of the aspects analyzed in this manuscript. These can be expected to provide important information on future developments of chemical reanalysis. As seen in the manuscript, strong statements were already made on the CAMS-Rean and TCR-2 comparisons.

To clarify this aspect, we now write in the revised manuscript, in the introduction:

**Even though these four reanalysis products are not equally independent, each of their configurations show substantial differences which are bound to impact the performance of the reanalysis products.** This intercomparison aims to reveal to what extend the reanalysis products agree, depending on region and time periods.

*2. There is a large confusion between TRC-1 (Miyazaki et al., 2015, available here https://ebcrpa.jamstec.go.jp/ miyazaki/tcr) and the version used in this paper. First, two different names should be used for these two different products. TCR-1 being already used, I suggest TCR-M (for MIROC) or anything that would clarify the confusion. But TCR-M seems closer to TCR-2 than TCR-1, except for the model spatial resolution. Moreover, on the TCR-1 webpage, it seems that surface NOx has been updated from Miyazaki et al. (2015) so it is difficult to know what is really TRC-M. In the revised paper, and in the section comparing TCRs reanalyses as suggested above, the authors should compare TCR-2 and TCR-1, not TCR-2 and TCR-M.*

Thank you for these suggestions. We agree that there have been some confusions. To solve the problem, (1) the TCR-1 website (https://ebcrpa.jamstec.go.jp/~miyazaki/tcr) has been updated. Now the original TCR-1 data using CHASER model (Miyazaki et al., 2015), as well as the updated version, as used in this manuscript, using the MIROC-Chem model (Miyazaki et al., 2017; Miyazaki and Bowman, 2017) are both provided on the TCR-1 website. So, now any reader can access both versions. Because the data assimilation settings are similar except for the forecast model, both versions are considered to be kinds of TCR-1. More detailed statements about these TCR-1 products are given in the revised manuscript to avoid any confusions. At the start of sec. 2.3 where we now write:

**A revised version of the TCR-1 data is used in this study. A major update from the original TCR-1 system (Miyazaki et al., 2015) to the system used here (Miyazaki et al., 2017; Miyazaki and Bowman, 2017) is the replacement of the forecast model from CHASER (Sudo et al., 2002) to MIROC-Chem (Watanabe et al., 2011), which caused substantial changes in the a priori field and thus the data assimilation results of various species.**

*3. The paper lack a dedicated section on the changes in the observing systems and its impact on the reanalyses which is largely commented throughout the paper. How does the time series of the Obervation-minus-Forecast statistics affected by these changes? Or the $\chi^2$-test, or the spread of the ensemble for EnKF systems, or the size of the analysis increments, or the number of relevant observations, or the comparison with a control run... This is essential for the users to know what they could expect – and what they can't – from these products.*

In response, changes in the observing systems indeed appear crucial to explain the behavior of the time series. The use of various satellite data streams is already mentioned in the manuscript, particularly Tables 2, 3 and 4. For detailed information of the assimilation statistics the reader is referred to Inness et al (2019) and Miyazaki et al. (2015), which we do not intend

to repeat here. Nevertheless, we now provide a new, dedicated section to discuss issues associated to the temporal consistency of the observing systems (sec 2.5), where we summarize the main issues with respect to the CAMS and TCR reanalyses. This now also includes references to the first guess and analysis departures relevant to the CAMS reanalyses, and reference to $\chi^2$ analysis relevant to TCR.

Furthermore, in the evaluation section we are now more specific as to which change we refer to, where 'changes of the observing system' are mentioned as a cause of artifacts.

*Regarding the use of the assimilated observations, the paper discuss ozone reanalyses in the polar region where TCRs are poorly constrained (no TES observations poleward 72 deg). What is not said in the paper is that CAMS reanalyses are probably not well constrained as well in the winter poles since the assimilated ozone column are from UV sensors which are blind during the polar night. In the revised manuscript, I suggest removing all the discussion related to the polar regions (thus removing these regions also from the figures).*

The CAMS reanalyses do not use $O_3$ total columns observations at solar elevation below 6° (Inness et al., 2019), which indeed implies that the CAMS reanalyses are not directly constrained during polar winters. Limb observations are used over a wider range of conditions, putting some constraints on tropospheric ozone as well. Therefore, as also suggested by the reviewer in a specific comment below, we move our comment on the TCR to Sec 2.3, and additionally we now include a comment in Sec 2.1 specifically on the CAMS systems:

**Note that no total columns are assimilated for solar elevations less than 6°, hence excluding polar winters.**

Nevertheless, we do not agree with the reviewer that any evaluation during polar conditions should be removed. Figure 4 of the original manuscript (time series of biases) in fact show that the tropospheric ozone during conditions where direct observations are absent are still influenced from satellite observations, as the biases are actually affected by changes in the observing system (e.g. the use of early SCIAMACHY and MIPAS retrievals during 2003). Also we believe it is worth evaluating the quality of the reanalyses for such conditions for any potential users. Although not perfect, the evaluation statistics still shows mostly acceptable values (with exception of TCR-1 over the Antarctic, and CAMS reanalyses before 2005), which could make this a useful product within its uncertainties.

*4. The figures need to be improved. The resolution of all the figures are too small. Many readers, like me, will try to zoom into them in the PDF document, which is not possible with their current resolution. Please, increase them. For the line plots, add a grid in the background of the figure. In general, the fonts are too small, they must be increased, as well as the line width. The legends are not always complete, please, describe everything shown in the figure. E.g. in Fig. 4, what is the dashed line referring to the left y-axis (which I cannot read due to the small size of the fonts)? You must also write what is shown when biases are plotted: obs-reanalyses or reanalyses-obs. If normalized differences, what is the norm? In Fig. 5, the colour levels in the bias are not very well chosen because it appears that all of the reanalyses seems to be highly biased. Why not using a constant colorbar with large steps showing only relevant differences? To extract major signal from the time series, I am suggesting plotting moving*

*average allowing to detect the major differences between the observations and the reanalyses. Also, their readability will be improved by plotting the values of CAMS-REAN and TCR-2 only.*

We apologize for the quality of the figures in the manuscript published in GMDD, which was indeed generally not sufficient. We will ensure figures with better quality for the revised manuscript.

Likewise to Figure 3, the gray dashed line in Figure 4 refers to the number of stations that contribute to the statistics (right vertical axis). This is now included in the legend.

Biases are always defined as 'reanalysis-observation', which is the most obvious for this type of validation activity. A corresponding sentence has been introduced in the manuscript at the start of Sec. 4.1, as well as in label of the new Figure 3.

Normalization is done with respect to observations, as now included in the legend of new Figure 7. The color levels were chosen non-linear on purpose, as we believe the order of magnitude in bias values is the most relevant information, particularly in this type of figures showing bias on a global scale. Nevertheless, we simplified and optimized the color scale such that the relevant information is more easily visible from the figures. The legends in Figure 9 in the revised manuscript have been increased.

*5. Many aspects of the conclusions and in the abstract are not shown in the paper, e.g. the impact of the change in the observing system or the differences between the forecast models. On the other hand, the performance of the reanalyses in different tropospheric layer, conditions and seasons – which what this paper discusses – is almost ignored. In the conclusions it should make clear of what are the findings of this paper and what are subjects for future research.*

We agree with the reviewer that the abstract and conclusions can be improved to better reflect the findings of this work. In response, we have revisited the conclusions by reporting quantitatively on the biases in tropospheric columns, and on important changes in the observing systems throughout the reanalyses, affecting the long-term consistency:

**For instance, averaged over the NH mid latitude region the mean bias in tropospheric ozone columns (surface to 300 hPa) is -0.3 DU (corresponding to approx. 1% of observed tropospheric column) for CAMS-Rean, which was 0.8 DU (3%) in CAMS-iRean.**
**(…)**
**Similar to the CAMS reanalyses, for the NH mid latitudes the mean bias in tropospheric columns against ozone sondes improved from 1.8 DU (7%) in TCR-1 to 0.8 DU (3%) in TCR-2.**
**(..)**
**Also changes in the $NO_2$ observing system, including the OMI row anomaly after December 2009 and the limited temporal coverage of SCIAMACHY and GOME-2, are considered to affect long-term consistency. These results indicate the requirements for additional observational information and/or stronger inflation of the forecast error covariance for measuring the long-term analysis spread corresponding to actual analysis uncertainty.**

In the abstract we have added the following sentence, to identify the quality of the latest reanalysis products:
**For instance, for the NH mid latitudes the tropospheric ozone columns (surface to 300 hPa) from the updated reanalyses show mean biases to within 0.8 DU (3% relative to the observed column) with respect to the ozonesonde observations.**

*6. The writing lack of clarity. For example, I do not understand the first sentence of the introduction. A careful reread of the paper is necessary to improve its readability. See some example in the specific comments.*

We have improved the formulations throughout the manuscript, particularly at the sentences identified by the reviewer, and the conclusions section. Thank you for addressing this.

***Other general comments***
*1. Tables 5-9 provides a summary of the performances of each reanalysis compared to independent observations. This information is important and the values in the tables are mentioned throughout the paper. I have two major concerns with these tables. First, extracting the comparison between the reanalyses is difficult and I suggest replacing the tables by bar-plots. Second, I suggest replacing the RMS with the standard deviation of the difference. The RMS combines a measure of the bias and the variability of the difference. Since the bias is already provided, the standard deviation will tell us by how much the differences are distributed around the bias. For these figures, TCR-1 and CAMS-iREAN could be compared with their updates versions.*

These are good suggestions, thank you. We now compute the unbiased standard deviation, and provide the information in terms of bar-plots, see new Figures 3 and 5. We note that the information on the standard deviation now closely relates to the correlation analysis.

*2. Also regarding differences, how are them calculated: obs-rean or the opposite? When normalized, what is the norm?*

All biases are computed as 'rean-obs'. The normalization is always done with respect to the observations. We now include such comm
Biases are always defined as 'reanalysis-observation'. A corresponding sentence has been introduced in the manuscript at the start of Sec. 4.1, as well as in label of Figure 3. Normalization is done with respect to observations, as now included in the legend of new Figure 7, and at the start of Sec 4.1:

Corresponding mean biases […] are given in **Figure 3, where the bias is defined as the reanalysis-observation, throughout this work. The normalized values, as scaled with the mean of the observations, are given in Figure S1 in the Supplementary Material.**

*3. In Figure 3, the authors define the tropopause in each product as the altitude where ozone exceeds 150 ppbv which means that the altitude of the tropopause change from a product to the other. I suggest taking a surface pressure as defining the upper level of the free troposphere, e.g. 200 or 300 hPa. By using 300 hPa, they will be able to remove Fig. 12, which I suggest.*

The definition of the top altitude defining the troposphere indeed deserves some further consideration. The argument for choosing the 150 ppbv level is that in this way the tropospheric columns, as predicted by the reanalyses, and as observed from the ozone soundings, can most clearly be intercompared. But this indeed does not correct for any discrepancies in the altitude

of the chemical tropopause level between the reanalyses, and hence the actual partial columns within a pressure range can give a different values. This is particularly relevant for conditions where the reanalysis shows a significant under-estimation of the tropopause altitude, which would not be penalized. Indeed, using this metric, as a most remarkable change the TCR-1 performance over the Antarctic now shows decreased performance with mean bias of 2.6 DU instead of 2.1 DU.

Therefore we agree now to evaluate the $O_3$ PC from surface to 300 hPa. Also in the time series plots (new Figure 4) the 300 hPa level is now used. Differences in performance quality for the other reanalyses, and for regions are overall similar, so this does not affect our conclusions.

The key difference of (old) Figure 12 with respect to Figure 4 is that in Figure 12 the tropospheric ozone is *not* sampled at the locations of the observations, but assessed for the whole latitude band. Particularly for the tropics, but also for the Antarctic region this makes a large difference, relevant for the interpretation, which is otherwise not highlighted. Nevertheless, considering the length of the manuscript, together with the limited additional value, we agree to move this figure to the Supplementary Material and only briefly refer to it.

*Also, why showing the number of stations and not the number of soundings?*

We choose to present the number of stations in the figure, as we believe this quantity is most suitable for representing any changes in the evaluation configuration relevant to explain potential jumps in the reanalysis performance. Changes in the number of actual observations for different month would not reflect this, but would instead give a better indication of the robustness of the evaluation. Please note that in Figure 1 the number of observations per station that is contributing to the statistics has been indicated already.

*4. Regarding the use of the observations and in addition to my major comment above, the Tables 2 and 3 need to be revised.*
*(a) As far as I know, there is only one CCI product for SCIAMACHY/GOME-2 TC and MIPAS profiles. I thus recommend to remove "(BIRA)" and "(KIT)".*

The reviewer is correct, we now remove this in Table 2.

*(b) What version of SCIAMACHY CCI is used? Same for MIPAS CCI, and GOME profiles?*
*(I understand that NRT products have version changing during the time but this should not be the case for scientific – or offline – products.)*

The ERS GOME profiles used in CAMS-iRean are a version provided by the Rutherford Appleton Laboratory (RAL) that was also used previously in ERA-40, Munro et al. (1998). The MIPAS, GOME-2 and SCIAMACHY CCI data were obtained from http://cci.esa.int/ozone. To be more precise, the CAMS reanalyses used the HARMOZ_MIPAS/fv0004, TC_GOME2-A/B fv0100 and fc0300, and TC_SCIAMACHY/fv0300 data.
We now specify these version numbers in Tables 2 and 3.

*(c) Also, does CAMS-iREAN and CAMS-REAN both assimilated MIPAS ESA NRT and CCI profiles? Which seems to use twice the profiles of the same instruments? Please, clarify. I am also surprised to see that CAMS use MIPAS NRT, a product older than 15 years and which was reprocessed by ESA several times (the ESA offline v7 is now the latest validated version).*

The MIPAS NRT data were only assimilated for the period between January 2003 and February 2004, because no reprocessed CCI MIPAS data were available from the HARMOZ_MIPAS/fv0004 product for dates before 2005. For future reanalyses this dataset should be revisited to resolve this inconsistency.

*(d) You also mention MLS V3.4 which does not exist (at least for the offline products) – this is it either V3.3 or V4.2 (or shortly V3 or V4).*

We should clarify that the CAMS-interim reanalysis was using the V3.4 from January 2013 onwards, i.e. not the offline product. Note that V3.4 is documented in https://mls.jpl.nasa.gov/data/v3_data_quality_document.pdf . We now add this link in the manuscript.

*(e) I would also add the reference to each dataset in an additional column.*

We acknowledge that including references helps traceability, and also gives proper credit to the retrieval providers, if not given yet in the text. We now include full references in the tables.

*(f) The MLS version used in TCR-1 and TCR-2 are not clear. Version 4 is mentioned in the text while Table 4 mention version 3. Please clarify. Also use the appropriate MLS data quality document when referencing a version.*

The reviewer is correct: this should have been version 4.2 both for TCR-1 and TCR-2. This is now updated. We now also refer to Livesey et al. (2018) rather than Livesey et al. (2011).

*5. The terminology of "error statistics" is misused in the paper. It is generally applied to the error statistics in the DA system (i.e. B and R matrix and model error if any). In the case of this study, it is applied to the differences between the reanalyses and the observations so I would use the "observation-minus-analysis" statistics instead.*

Thank you for this comment. Our use of the wording 'error-statistics' is meant rather general, but may indeed be confusing in this context. We believe "observation-minus-analysis statististics" is also not appropriate, as this generally refers to the error statistics of any reanalysis against observations that are actually assimilated. Instead, we now change 'error-statistics' into 'reanalysis performance statistics'

*6. The authors use the inter-annual variability (IAV) and elsewhere deseasonalized anomaly, which seems to reflect to the same quantity. Could they clarify and use only one of those terminology?*

In our manuscript we analyze the inter-annual variability (IAV) of monthly mean variables. For this purpose we compute and assess the deseasonalized anomaly, by subtracting the multi-year average monthly mean concentrations from their instantaneous values, similar to what is for instance presented in Davis et al., (2017). To prevent confusion we now make a more strict difference in our referencing to IAV (which refer to variability in the absolute values), and anomalies with respect to the mean value.

*7. I prefer the acronyms CIRA and CAMSRA, it is much easier when speaking than CAMS-iREAN and CAMS-REAN.*

We agree that the definition of these acronyms is a little subjective, and CIRA and CAMSRA may be easier to read and pronounce. Nevertheless, the use of CAMS-iRean and CAMS-Rean was chosen to stress its common assimilation framework, in analogy to TCR-1 and TCR-2. Therefore we choose to stick to these acronyms. There have been some inconsistencies between use of capitals or not, this is now also resolved.

*8. Many acronyms are undefined and should be*

We went through the manuscript and now consistently defined acronyms at first appearance.

***Specific comments***
*L13-16: "Global tropospheric ozone reanalyses constructed using different state-ofthe-art satellite data assimilation systems, prepared as part of the Copernicus Atmosphere Monitoring Service (CAMS-iRean and CAMS-Rean) as well as two fully independent Tropospheric Chemistry Reanalyses (TCR-1 and TCR-2), have been intercompared and evaluated for the past decade." This is not true. CAMS-iREAN and TRC-1 are not constructed using state-of-the-art satellite data assimilation systems since these systems have been updated for CAMS-REAN and TCR-2.*

We do not agree with the reviewer on this point, arguing that the data assimilation systems used either for CAMS and TCR have not fundamentally changed between the predecessor and their latest versions. The reviewer is correct that the resulting reanalyses, which depend on more aspects than the data-assimilation system (forward model configuration, model resolution, etc) cannot equally be referred to as 'state-of-the-art', but we also do not claim that. The second sentence in the abstract ("the updated reanalyses generally show substantially improved agreements..") indeed clarifies that the latest versions should be considered 'state-of-the-art'.

*L18-20: "The improved performance can be attributed to a mixture of various upgrades..." This is not shown in the paper.*

The reviewer is correct that we are not able to pinpoint exactly the cause of the improved performance, as that requires dedicated sensitivity experiments. Nevertheless, the improvements seen for the updated reanalyses must be a consequence of their different configuration, both in data-assimilation and forecast model, as specified in particular in Sec. 2 Therefore we now rewrite this statement as:

"The improved performance can **likely** be attributed to..."

*L21-23: "Meanwhile, significant temporal changes in the reanalysis quality in all the systems can be attributed to discontinuities in the observing systems." Idem, this is not shown in the paper.*

We now provide a specific section (Sec. 2.5) where we summarize the changes in time in the observing system, and also throughout the various evaluations we refer to specific changes. Therefore we consider this to be shown by our evaluations.

*L22-24: "To improve the temporal consistency, a careful assessment of changes in the assimilation configuration, such as a detailed assessment of biases between various retrieval products, is needed." Which is what this paper should have been shown.*

Here we do not fully agree with the reviewer. This paper is meant as an a-posteriori evaluation of the reanalysis products, and it is beyond the scope of this work to analyze biases between retrieval products. This has is in part been addressed in Inness et al (2019), see their Sec. 3.2, and Figure 6, as well as Figures S1-S3 in their Supplementary Material. Nevertheless, the posteriori evaluation shown in our work indicates various other jumps which cannot be explained from changes in foreward model configuration, and hence implies biases between retrieval products. Likewise for TCR, changes in performance are detected which have already been briefly addressed in Miyazaki et al (2015), and hence do not need analysis here. The recommendation written in our abstract addresses the identified issue of biases between retrieval products, which needs to be addressed in future reanalysis configurations to obtain an improved consistency over time in tropospheric ozone reanalyses.

*L24-26: "Even though the assimilation of multi-species data influences the representation of the trace gases in all the systems and also the precursors' emissions in the TCR reanalyses, the influence of persistent model errors remains a concern, especially for the lower troposphere." Again, this is not shown in the paper.*

The reviewer is correct that we do not assess the impact of model errors in the scope of this work, but only make various references to its potential impact. Therefore we agree to remove this sentence from the abstract. We still believe there is sufficient evidence that part of the discrepancies seen in the observations are due to biases in model parameterizations, which would justify the last sentence of the abstract, discussing potentials for improvement.

*L31-32: "The global distribution of present-day tropospheric ozone..." I don't understand this sentence, please, rephrase.*

Thank you for your fair comment. We have rewritten, and thereby simplified, the formulation of this sentence into:

**Both human activity and natural processes influence the global distribution of present-day tropospheric ozone, together with its interannual variability and trends.**

*L41: "...tropospheric ozone, but are generally..." => "...tropospheric ozone, which is generally..."*

Changed

*L45: "Tropospheric ozone is reasonably well monitored..." You are talking about surface ozone in this sentence so I would write "Surface ozone is reasonably..."*

Changed.

*L50-52: This list of satellite dataset is incomplete (missing are e.g. OMPS and TROPOMI for the most recent instruments) so I would write "These observations are*

*complemented with (combined) satellite observations from, e.g., GOME-2, ...”*

We changed this into:
(…) **satellite observations from instruments such as** (…)

*L62-64: “Simultaneously international modelling initiatives...” I don't understand this sentence, please, clarify.*

This sentence is meant to address some of the main coordination and collaboration frameworks that have emphasis on various aspects which rely more heavily on modeling, both in air quality and climate change context. To clarify better we rephrased this sentence to :

**Additional coordination with the emphasis on modelling activities related to tropospheric ozone have been established, for instance (…)**

*L77: “...individual measurements suffer...” Do you mean “...individual measurements which suffer...”?*

No, here we refer to the impact of representativity of individual observations for drawing general conclusions, i.e. undersampling, or sampling bias. We clarify this better by writing:

**This was shown useful as evaluations using individual measurements are subject to significant sampling biases**

*L81: “...particular constellations of pollution...” What do you mean by “constellations”?*

We simply mean 'pollution events', as directly clarified in the consecutive sentence. The reviewer is correct that the wording is a bit awkward. We have rewritten this to:

**… and to analyse particular pollution  events such as those associated with heat waves…**

*L85: “However, all of these applications presume that the reanalysis is sufficiently accurate,...” What matter is that reanalysis is well characterized more than accurate.*

Strictly speaking the reviewer is correct. When well characterized, users of respective reanalyses can take such information into account in their applications. On the other hand, if the characterization of biases is complex, because of changes in time and space, then the use of any such product is still hampered. Therefore, we argue that in practice a specification of the accuracy of the reanalysis may then be more desirable. We rewrite this into:

**However, all of these applications presume that the reanalysis is sufficiently accurate, or, to the least, well characterized. Despite the range of observations assimilated into the respective systems, this is not necessarily ensured.**

*L118-119: CAMS-REAN and CAMS-iREAN acronyms are undefined.*

Now defined slightly above:

**… the 'CAMS Interim Reanalysis' (hereafter 'CAMS-iRean') (…) and recently the 'CAMS Reanalysis' ('CAMS-Rean')**

*L126: "NOx" => "NOx"*
Changed

*L129: "...changing constellation of ..." => "... the change in the observing system..."*

Changed

*Table 1: What are the output frequency of each product. Are the output snapshots or time averages?*

The basic output frequency in the CAMS products is three-hourly for the 3D-fields evaluated here, as already specified at the end of Sec. 2.2. The TCR products adopt two-hourly output. This is already specified at the end of Sec. 2.4. We think this should do.

*L156: => "The meteorological model version is CY40R2."*

Thank you. Changed to:

**The meteorological model is IFS CY40R2.**

*L157: "In terms of ozone, observations from the following set of satellite instruments have been assimilated:..."*

Changed, thank you.

*L159: "Limb observations are instrumental to discriminate..." => "Profiles from limb instruments (MIPAS and MLS) are used to discriminate..." Could you explain how does limb profiles are used to discriminate the tropospheric and stratospheric contribution of the total column observations?*

By assimilating both total and stratospheric columns, the tropospheric columns are indirectly constrained as the residue of both elements. We now change the manuscript on this aspect writing:

**Profile observations from limb instruments (MIPAS and MLS) are used to constrain the stratospheric contribution of the total column. In combination with the assimilated total column retrievals this implies that also the tropospheric part is constrained (Inness et al., 2013).**

*L161-163: See my general comment above regarding MLS V3.4.*

We clarify that this indeed refers to the version V3.4, see also above.

*L211: Remove reference to Watanabe et al. since it is already provided 2 lines above.*

Done, thank you

*L341: "In the TCR systems,..." Move this info in Sect. 2.3.*

Sentense has been moved to Sect 2.3.

*Figure 2: What is the difference between the part in page 14 and 15? It seems to be the same.*

This was indeed an duplication of plots, we apologize for this.

*L376: "...both model and observations..." Which model? Do you mean the reanalyses? If yes, replace by "... the reanalyses and the ozonesondes..."*

The reviewer is correct. Nevertheless the complete sentence is now removed as this statement is no longer correct when analyzing the partial columns from surface to 300hPa instead.

*L377: same as above "modelled" or "analysed"?*

We have updated this. Also elsewhere throughout the document we have revisited the use of *'model'* and *'modeled'*, and changed to 'analysis'/'analyzed' where appropriate.

*L379-381: Is the poor correlation between reanalyses and observations due to the missing total column observations during the polar night? Since, as far as I know, none of the total column assimilated data are taken by emissions instruments thus failing to measure during the night?*

The reviewer is correct that no total column (in CAMS), and also no TES profile retrievals (in TCR) are assimilated during polar nights. We discuss these aspects in more detail as part of Sec 4.3, see also the reviewer comments on this issue below (as well as in our response to his/her main comments).

*L397-399: "During 2003 and 2004 both CAMS reanalyses..." Why? This is not related to GOME data since CAMS-REAN does not assimilate GOME.*

The 2003-2004 discrepancy compared to other years, particularly at the 350 hPa level, was attributed to the use of early SCIAMACHY and NRT MIPAS $O_3$ retrievals, which are of poorer quality than the observations used lateron. The GOME issue was mostly related to the differences between the two CAMS reanalyses in 2003 at altitudes below 650 hPa. The manuscript was not fully clear on this. To clarify better, we rewrote this section:

**During 2003 and 2004 both CAMS reanalyses show anomalously low springtime ozone, different to the rest of the time period, particularly at ~350 hPa. The different reanalysis performance statistics 2003 over the Arctic compared to later years is attributed to the use of early SCIAMACHY and NRT MIPAS $O_3$ retrievals, which are of poorer quality than the OMI MLS observations which have been used from August 2004 onwards, and reprocessed MIPAS data used from January 2005 onwards. CAMS-iRean also shows a large offset compared to observations and CAMS-Rean in 2003, particularly at altitudes below 650 hPa. This was attributed to the assimilation of GOME nadir profiles in CAMS-iRean, which has been omitted in CAMS-Rean (Inness et al., 2019).**

*L399: "...GOME observations..." => "...GOME nadir profiles...".*

Changed, thank you.

*L400-403: Why does CAMS assimilate MIPAS NRT and not the offline reprocessed products delivered by ESA?*

The MIPAS NRT data were only assimilated for the period between January 2003 and February 2004, because no reprocessed CCI MIPAS data were available from the HARMOZ_MIPAS/fv0004 product for dates before 2005 from [http://cci.esa.int/ozone](http://cci.esa.int/ozone). As already commented above, in future reanalyses this dataset should be harmonized to resolve this inconsistency, which is indeed an important issue. This is now also addressed specifically in the conclusion where we now write:

**Discontinuities in the availability, coverage and *product version* of the assimilated measurements are also shown to affect the quality of the reanalysis, particularly in terms of temporal consistency, both in the CAMS and TCR-reanalyses.**

*L412-413: "Also both the observations and reanalyses indicate an upward trend of tropospheric ozone in the UTLS..." I don't see this from figure 4. Could you clarify?*

This indeed cannot be seen from Figure 4, but is visible from the corresponding Figure S1 in the Supplementary material, presenting the $O_3$ monthly mean values over the given regions and altitude ranges. The NH polar region at 378 hPa shows a clear sign of an upward trend, both in observations and reanalyses. We now make explicit reference to this figure in the manuscript, which was missing indeed.

*L431-433: "From 2011 onwards the correspondence with observations improves remarkably, despite the lack of TES measurements in TCR from June 2011 onwards."*
*Why?*

[Figure]

*Figure R2: Absolute value (left) and mean bias (right) of $O_3$ at ~652 hPa against sonde observations over Japan.*

The changes in bias characterization of the reanalysis is obvious from Figure R2, but the reason for this is not well understood. Not only the absolute values show an upward trend over the 2003-2016 time period (Figure R2, left), which seems absent in the reanalyses, but also there are changes in the observing system. We now write:

**The changes in performance statistics for all reanalyses likely have multiple causes. This includes trends in the observed ozone (Verstraeten et al., 2015), associated to changes in Chinese precursor NO$_x$ emissions (e.g. van der A et al., 2017). Also changes in the observing system are important to consider, particularly the reduction of assimilated TES measurements in TCR from 2010 onwards, and the row anomaly issues affecting assimilated OMI O$_3$ and NO$_2$, see also Sec. 2.5.**

*L434-444: I do not agree with most of what is written.*
*"In the tropics, ..." This is not true for CAMS-iREAN which generally underestimate the ozonesondes.*
*"... both CAMS reanalyses show a strong peak ..." In fact, TCRs also show a peak.*
*"...overestimation of up to 20 ppb." None biases are going up to -20 ppb. I would rather say -15 ppb. Do not omit the sign of the bias in the comparison.*
*"This spike appears much weaker in TCR..." Does the reason not due to the fact that TCR also optimize surface emissions allowing the reduce the bias with observations?*

*But the authors does not discuss the fact that CAMS-iREAN seems to have the best agreement with ozonesondes during the whole period and they should comment on the reason for this.*

We thank the reviewer for closely checking our analysis. We have updated the comment on the mean bias before 2012. Also the exceptional peak in 2015 was only visible at the ~850 hPa altitude, only for CAMS reanalyses, and to much lesser extent at ~650 hPa. We confirm that the sign of the bias (reanalysis-observation) is positive, and reaches 20 ppb. As the reviewer suggests, the discussion why TCR behaves differently than CAMS, with on average more acceptable O$_3$ values, is possibly not only due to the sampling issue, but can also be associated to better optimized NO$_x$ emissions compared to those from GFAS, as used in CAMS.
The CAMS-iRean is not superior to CAMS-Rean at the 650 and 350 hPa altitude range; it is unfortunately not clear what is the reason for the better performance before 2012 at the 850 altitude range, although a likely explanation appears the change in MLS version used in CAMS-iRean from 1 January 2013 onwards.
In summary, following his/her comments, we change this section into:

**In the tropics, all reanalyses except CAMS-iRean overestimate ozone at 850 hPa before 2012, with positive biases in the range 2.5-3 ppb. The different performance for CAMS-iRean from 2012 onwards is probably associated to the use of another version of the MLS retrieval product. Interestingly, both CAMS reanalyses show a strong peak in ozone at 850 hPa during the second half of 2015 (see corresponding Figure S1 in the Supplementary material), but with a zonally averaged overestimation of up to 20 ppb. This is associated to the strong El Niño conditions, and this particular spike was attributed to an over-estimate of ozone observed at the Kuala Lumpur station for October 2015. Here exactly the grid box affected by the extreme fire emissions in Indonesia for this period (Huijnen et al., 2016), as prescribed by the daily GFAS product, has been sampled. This peak appears much weaker in TCR. Possible explanations are lower optimized NO$_x$ and CO emissions in TCR compared to those used in CAMS, resulting in**

**weaker ozone production, together with a coarser model resolution. At 650 hPa, the TCR reanalyses overestimate ozone almost throughout the reanalysis period (by 3.1–3.8 ppb on average), whereas the CAMS-Rean shows closer agreement with the observations (mean bias = 0.5 ppb, RMSE = 3.2 ppb). At ~350 hPa, the TCR-2 shows improved agreement compared with the earlier TCR-1, as confirmed by improved mean bias (from 4.3 to 0.6 ppb) and RMSE (from 6.6 to 5.7 ppb) although the temporal correlation remains relatively low.**

*L449: "332 hPa" => "382 hPa"?*

Changed, thank you

*L467-474: I see other reasons for the seasonal variations in the bias time series than those mentioned in this §. For CAMS products, their troposphere is not constrained by any data during the polar night since all of the assimilated nadir instruments are measuring UV sun-scattered light. For TCR, TES ozone data are only available at latitude lower than +/-72°. Could the author comment on that?*

The reviewer is correct that there are no constraints on total $O_3$ column in the CAMS reanalyses during polar winter, neither tropospheric $O_3$ profiles from TES in the TCR reanalyses over the poles. Indeed the seasonal variations in the availability of satellite observations, in particular for the CAMS reanalyses, is bound to contribute to the seasonal cycle in their biases.
Likewise, if TES observations would have been available for this region then the bias in TCR-1 would probably have been much smaller. Nevertheless, as shown for the TCR-2 reanalysis, also a meaningful product with a mean bias (stddev) of within 2 (4.5) ppb at 650 hPa can be provided by optimizing the data-assimilation system, even if direct satellite observations are not available.
We revise the manuscript accordingly as follows:

**The seasonal cycle in the biases can largely be attributed to the lack of $O_3$ total column observations during polar night, combined with a seasonal variation in model forecast biases.** The TCR reanalyses largely underestimate ozone during austral summer and autumn in the lower troposphere. At 351 hPa, TCR-1 substantially overestimates ozone throughout the year because of large model biases and the lack of observational constraints. **This large positive bias was resolved in TCR-2 by improving the modelling framework.**

*L473: "332 hPa" => "351 hPa"*

Changed, thank you

*Sect. 5.2: "Figure 6 presents the temporal variability..." Well, figure 6 is a scatter plot without any time axis (on the x-, y- or any colorbar) so I would change this sentence. Moreover, all the discussion in this §related to seasonal differences are not supported by Fig. 6. I understand that Fig. S3 could support this discussion but as being part of the supplement, it cannot be used for new discussion.*

The reviewer is correct. We changed the formulation to better connect the discussion to the presented figures, and omit statements that largely rely on results presented in the Supplementary material. We have rewritten this section as follows:

**Figure 6 presents scatter plots of monthly mean ozone from the reanalyses against those from the TOAR surface observations for various regions. The corresponding time series are given in Figure S3 in the Supplementary Material. As is clear from Figure S3, the main driver of the variation in magnitude of ozone concentrations in the reanalyses and observations in Figure 6 is the seasonal cycle. Over the Arctic, the general pattern in the seasonal variations is captured for all reanalyses (R between 0.58 and 0.72), although they all underestimate the increased ozone values during boreal spring.**

**Over Europe and the US, the CAMS reanalyses show the closest agreement with the observations (MB between  -2.4 and 1.5 ppb, R>0.8). Furthermore, CAMS-REAN shows reduced negative biases for observed low ozone values compared with the CAMS-iREAN, which is in boreal winter and spring . The TCR reanalyses exhibit large positive biases over Europe and the US regions (MB between 6.7 and 17 ppb), with significantly lower biases in TCR-2. Over East Asia, all the reanalyses show positive biases in the range of 2.7 ppb (CAMS-REAN) to 10.5 ppb (TCR-1) and fail to reproduce the minimum concentrations in autumn. Still the temporal correlations are similar to most other regions (R between 0.79 and 0.83), associated with the stable seasonal cycle in both the reanalyses and observations. Over Southeast Asia, positive biases exist throughout the period, which are largest in TCR-1. For this region the TCR-reanalyses show lower temporal correlations (R between 0.39 and 0.49) compared to the CAMS reanalyses (R=0.68). Significant changes in the surface ozone biases are found in the TCR reanalyses over the SH mid latitudes, with reduced values after 2010.**

**The CAMS reanalyses capture well the temporal variability over the SH mid latitudes and Antarctic (R between 0.89 and 0.96), while CAMS-REAN shows a positive bias for observed high ozone values. This is associated to model biases austral winter (JJA), particularly during 2005-2013, Figure S3. The TCR reanalyses show a significant negative bias throughout the year except for observed low ozone values (during Austral summer) which results in lower temporal correlations (R~0.68).**

*L521: Here and at several other placed "R=0.89 – 0.96"? Do you mean "R between 0.89 and 0.96" or "R˘CO˝ [0.89,0.96]"? Or something else?*

We refer to values between a minimum and maximum. We clarify this now by writing explicitly

**R between 0.39 and 0.49** (etc)

*L541: "We compute the interannual..." Do you mean the deseasonnalized anomaly for each region? See also the general comments.*

As described above, we now make a more strict difference in our referencing to IAV, and to deseasonalized anomalies with respect to the mean value. Particularly, at the start of Sec. 5.3 we now write:

**We assess the interannual variability (IAV) by computing the deseasonalized anomaly of surface ozone concentrations. For this, the 2005-2012 multi-annual monthly, regional mean surface ozone is subtracted from its corresponding instantaneous monthly, regional mean value, (…)**

*L652: Do you mean Fig. 12? So this is almost Fig. 3 without observations. Is it really the annual mean? It seems more to be a time series of monthly mean?*

The reviewer is fully correct that this should have been reference to Fig. 12, and refers to monthly means rather than annual mean. This figure is analogue to Figure 3, but with the main difference that it much better reflects the average zonal mean, as it is not sampled for station locations. This figure has now been moved to the Supplementary Material, together with most of the contents of this section.

*L669-670: The change in behaviour is clear above the SH polar latitude but less clear in SH midlatitudes.*

The reviewer is correct, thank you. It should have written:
**Particularly at the SH high-latitudes, but to lesser extend also at the SH mid-latitudes, there is a remarkable change in behaviour after 2013 in all reanalyses except TCR-1**
But, following reviewer #3 we choose to remove this section from the main manuscript, in view of duplication and length. The figure is retained in the Supplementary Material.

*Figure 13:  Is it as Fig. 7 but for PC surface-300 hPa in south-east Asia and ENSO?*
 *"A 2-month smoothing". Do you mean a running mean or moving average?*
*What is TSI?*

Indeed a similar procedure has been followed to create Figure 13 as was done for Figure 7. For better clarity we now refer to 'deseasonalized anomalies'. The reference to 'TSI' was spurious, and has now been removed. Discussion of this figure has been moved to the end of the next section.

*L742: "...annual mean..." For which year?*

This actually refers to the multi-annual mean analogous to what is presented in Figure 14.

*Figure 15 is very interesting but I would add the ozone sonde values in order to assess the quality of the reanalyses against the best estimation of the truth (i.e. the sondes).*

This is a good suggestion. We now also compute the frequency distribution sampled for instantaneous sonde observations at three pressure levels. This indeed gives a quantitative impression of (differences in) reanalysis performances, as quantified by the total absolute difference between the frequency distributions of the reanalyses and observations. Nevertheless, an important drawback is that by sampling the analyses at the location and time of the observations the global representativity, which was central to this section is largely lost.

Therefore we choose to provide this evaluation as part of the supplementary material, figure S6. In the manuscript we now write:

**A corresponding evaluation of the frequency distributions, but sampled at individual ozone sonde observations, is given in Figure S7 in the Supplementary material. Because of the different sampling approach the shape of the frequency distributions is different than was seen in Figure 15. Evaluation of the absolute differences *d* between analyzed and observed frequency distributions indicates that at 850 hPa the performance between the four reanalyses is very similar (*d* between 0.17 and 0.19), while at 650 hPa CAMS-Rean is superior (*d*=0.13). CAMS-iRean shows an under-estimate of the frequency of high ozone values (larger than ~55 ppbv) at 850 and 650 hPa, explaining the worst performance at 650 hPa (*d*=0.20). At 350 hPa the differences in performance are largest, with best correspondence to observations for CAMS-iRean (*d*=0.11), and worst for TCR-1 (*d*=0.43).**

To aid the interpretation, Figure 15 is now presented in terms of bars.

*L767: "The changing constellation..." I would rather say "The changes in the observing system..."*

We change this, thank you for your suggestion.

*L770: "This calls for a detailed evaluation of the capability of the current reanalyses of tropospheric ozone." Do you mean this is something to do in the future? Please, clarify.*

Here we refer to our study. We change the sentence into:

**This calls gives rise for a detailed evaluation of the capability of the current reanalyses of tropospheric ozone, as presented here.**

*L793-795: "In the TCR reanalysis, the chemical concentrations and precursor's emissions were simultaneously optimized through EnKF data assimilation, which was important in providing information on precursors' emissions variations (Miyazaki et al., 2014; 2017; 2019a; Kiang et al., 2018) and in improving the vertical profiles of ozone."*
*Well, this is not shown in the paper so I would remove this comment from the conclusions.*

We agree with the reviewer that this is not shown in this manuscript, and remove the sentence.

*L800-803: "Meanwhile, the analysis ensemble spread ..." Well, again, the TCRs ensemble spread are not shown in the paper. Also, what do you mean with "4D-var could be used ..." Altogether, I don't understand the message in this sentence.*

These sentences contain recommendations for further improvements, and are therefore not shown in the manuscript. To clarify better, we change the sentence to:

**Furthermore, in future studies the analysis ensemble spread from EnKF can be regarded as uncertainty information about the analysis mean fields, indicating the need for**

**additional observational constraints. Likewise, in the 4-D Var system the contributions from individual retrieval products can be tested.**

*L413: The acronym UTLS must be defined.*

We do this now at first appearance (sec 4.3)

*L819: " ... a careful assessment of changes in the assimilation configuration..." Which what this paper should have done.*

Here we do not agree with the reviewer. Our manuscript provides an a-posteriori evaluation of the reanalysis products, and as such provides various indications where changes in the tropospheric ozone reanalyses are linked to changes in the observing system. Our evaluations should be taken into account when designing an updated observing system and details regarding the data assimilation configuration in future reanalyses. To clarify better, we rewrite this section into:

**We have shown that discontinuities in the availability, coverage and product version of the assimilated measurements affect the quality of any of the reanalyses, particularly in terms of temporal consistency. This is particularly important for assessing interannual variability. The influence of data discontinuities must be considered and where possible removed when studying interannual variability and trends using products from these reanalyses. To improve the temporal consistency in future reanalyses, a careful assessment of changes in the assimilation configuration, most prominently associated with ozone column and profile assimilation is needed, including a detailed assessment of biases between various retrieval products.**

*L822: "The assimilation of multi-species data influence..." This has not been addressed in the paper.*

Analogous to our response above, our manuscript is not intended to assess in detail the impact of individual contributions of the data assimilation configurations on the quality of resulting reanalyses, such as multi-species assimilation, or issues regarding the CTM's. The reviewer is correct that this has not been analyzed in our manuscript, as this would require dedicated sensitivity experiments. Therefore we agree with the reviewer that we should be more accurate in our formulation. We now write:

**The assimilation of multi-species data in both the CAMS and TCR configurations influences the representation of the entire chemical system, while the influence of persistent model errors in complex tropospheric chemistry continues to be a concern. Therefore, further improvements to long-term reanalyses of tropospheric ozone can be achieved by improving the observational constraints, together with a further optimization of model parameters, such as the chemical mechanism, emission, deposition, and mixing processes.**

**References**

Davis, S. M., Hegglin, M. I., Fujiwara, M., Dragani, R., Harada, Y., Kobayashi, C., Long, C., Manney, G. L., Nash, E. R., Potter, G. L., Tegtmeier, S., Wang, T., Wargan, K., and Wright, J. S.: Assessment of upper tropospheric and stratospheric water vapor and ozone in reanalyses as part of S-RIP, Atmos. Chem. Phys., 17, 12743–12778, https://doi.org/10.5194/acp-17-12743-2017, 2017.

Munro, R., R. Siddans, W. J. Reburn, and B. J. Kerridge, Direct measurements of tropospheric ozone distributions from space, Nature, 392, 168–171, 1998.

---

## Author Comment (AC3) · 7 Feb 2020

**Response to the third reviewer**

We thank the referee for his/her short, but nevertheless useful, positive review, which contain various useful comments and suggestions. Here we answer them to our best ability. The reviewer comments are in italic. Our responses are in regular font, and changes to the manuscript are given in bold.

*This paper inter-compared tropospheric ozone reanalysis products from CAMS, CAMS-Interim, TCR-1, and TCR-2. This study is of scientific importance and the research is well conducted. The presentation is generally clear with logic flow and convincing discussions. The paper provides an enhanced understanding of issues related to tropospehric ozone reanalysis products. I only have some minor issues for the authors to consider when revising their paper. Minor issues:*
*1. In the abstract and conclusions, it is useful to summary where and when the reanalysis products perform strongest and weakest, in term of relative difference with ozonesonde data.*

In response, we now specify in the abstract a sentence on the evaluation against ozone sondes:

**For instance, for the NH mid latitudes the tropospheric ozone columns (surface to 300 hPa) from the updated reanalyses show mean biases to within 0.8 DU (3% relative to the observed column) with respect to the ozonesonde observations.**

Also in the conclusions we describe the main strengths of the reanalyses, and suggest potential application areas:

**The well-characterized, small mean bias in tropospheric columns in these reanalyses suggest that they can be used to provide a climatology of present-day tropospheric ozone. This may serve as a reference for the present-day contribution of tropospheric ozone to the radiation budget, or may provide a climatology for a-priori ozone profiles as required for satellite retrieval products (e.g., Fu et al., 2018). The ability of the CAMS Reanalysis to capture the variability of (near-)surface ozone on multiple time scales, and for many regions over the globe, indicates it is fit for use as boundary conditions for hindcasts of regional air quality models.**

*2. Figures and tables need more annotation.*

In response, we have extended the descriptions of (new) Figures 1, 4, 6, 7, 15.

*Fig. 1, What are the boxed areas?*

We now specify the regions used in the analyses in the legend of Figure 1, together with specification of the other regions.

*Table 5-7, the area definitions can be provided in Fig. 1.*

This is a good suggestion, thank you. We now provide this information, see above.

*Fig. 4. Relative difference is more meaningful.*

We prefer to stick to the absolute values here, to remain close to the physical quantity. Nevertheless, in our revisions we now report relative differences much more frequently, e.g. by adding bar-plots presenting the relative biases and standard deviations in the Supplementary material, and referring to this in our analyses, as well as in the abstract and conclusions.

*Fig. 11, what is the time zone for this figure? How are the model errors removed?*

For the diurnal cycle we use UTC, we now include this in the x-axis label in the Figure. The model bias was removed by subtracting the seasonal mean analysis bias with respect to the corresponding observations. We now write this explicitly.

***3.** The definition of the tropopause needs some discussion and references.*

As was also commented by the other reviewers we have updated our analysis of tropospheric columns. This now refers to subcolumns from the surface to 300hPa. In this way we circumvent any potential ambiguity regarding the definition of the tropopause, and make the reanalysis products better comparable.

***4.** The paper appears lengthy. Please shorten the paper and move less significant contents to Supplements.*

The reviewer is correct that the manuscript benefits from a more stringent priority in presenting material, thank you for your comment. In response, we decided to move most of Sec. 7 into the supplementary material. Only an assessment of the correlation with the ENSO is retained, as well as the concluding sentences which describe the consistency in time series between the various renalyses.